# Pluripotent stem cell-derived model of the post-implantation human embryo

Bailey A. T. Weatherbee[1,7], Carlos W. Gantner[1,7], Lisa K. Iwamoto-Stohl[1], Riza M. Daza[2], Nobuhiko Hamazaki[2], Jay Shendure[2,3,4,5] & Magdalena Zernicka-Goetz[1,5,6 ✉]

The human embryo undergoes morphogenetic transformations following implantation into the uterus, but our knowledge of this crucial stage is limited by the inability to observe the embryo in vivo. Models of the embryo derived from stem cells are important tools for interrogating developmental events and tissue–tissue crosstalk during these stages[1]. Here we establish a model of the human post-implantation embryo, a human embryoid, comprising embryonic and extraembryonic tissues. We combine two types of extraembryonic-like cell generated by overexpression of transcription factors with wild-type embryonic stem cells and promote their self-organization into structures that mimic several aspects of the post-implantation human embryo. These self-organized aggregates contain a pluripotent epiblast-like domain surrounded by extraembryonic-like tissues. Our functional studies demonstrate that the epiblast-like domain robustly differentiates into amnion, extraembryonic mesenchyme and primordial germ cell-like cells in response to bone morphogenetic protein cues. In addition, we identify an inhibitory role for SOX17 in the specification of anterior hypoblast-like cells[2]. Modulation of the subpopulations in the hypoblast-like compartment demonstrates that extraembryonic-like cells influence epiblast-like domain differentiation, highlighting functional tissue–tissue crosstalk. In conclusion, we present a modular, tractable, integrated[3] model of the human embryo that will enable us to probe key questions of human post-implantation development, a critical window during which substantial numbers of pregnancies fail.

Human reproduction is remarkably inefficient, with an estimated 60% of pregnancies failing to progress during the first two weeks following fertilization[4]. The human blastocyst consists of the outermost trophectoderm (a precursor of the placenta) and the inner cell mass, which gives rise to both the embryonic epiblast and the yolk sac precursor (the hypoblast). Between 7 and 8 days post-fertilization (dpf), the blastocyst implants into the endometrium and the epiblast polarizes and transitions from the naive state of pluripotency to the primed state[1]. A central amniotic cavity forms within the epiblast, separating the dorsal amniotic epithelium and the ventral epiblast, which gives rise to the embryo proper[1]. A subset of cells in the hypoblast maintains expression of NODAL, bone morphogenetic protein (BMP) and WNT inhibitors, safeguarding the future anterior epiblast from posteriorizing signals during primitive streak formation[2]. An additional extraembryonic tissue, the extraembryonic mesenchyme, is located between the tissues derived from the inner cell mass and the trophoblast—however, the origin of these cells remains unclear[5].

Recent work in mouse embryos established conditions amenable to human embryo in vitro culture through implantation, opening up this developmental black box[6–8]. However, mechanistic work in the human embryo remains challenging. Thus, models of the embryo derived from stem cells will serve as an important complementary tool for understanding this crucial period of human development. Several groups have reported the generation of blastocyst-like structures derived from human embryonic stem (ES) cells[9–13], but these develop poorly to post-implantation stages. Other models, including gastruloids, 2D micropatterns and spheroids, can model aspects of post-implantation development[14–18] but are derived entirely from human ES cells, lack targeted extraembryonic tissues and do not recapitulate embryo morphology.

The derivation of lineage-specific cell lines offers scope for modelling these tissues in vitro. However, generating a modular, integrated model system that includes both embryonic and extraembryonic tissues has proved challenging. This may be owing to the opposing signalling pathway modulators required in human ES cell culture, hypoblast-like cell differentiation and trophoblast-like cell differentiation. Moreover, although tissue–tissue crosstalk is an advantage of integrated model systems, the generation of embryo-like structures in medium containing exogenous factors may compromise tissue-driven self-organization. Given these limitations and the ability of tri-lineage models to mimic mouse development[19–21], we used the approach of overexpressing transcription factors that can drive downstream activation of extraembryonic-like gene programmes without the need for exogenous factors.

[1]Department of Physiology, Development and Neuroscience, University of Cambridge, Cambridge, UK. [2]Department of Genome Sciences, University of Washington School of Medicine, Seattle, WA, USA. [3]Brotman Baty Institute for Precision Medicine, Seattle, WA, USA. [4]Howard Hughes Medical Institute, Seattle, WA, USA. [5]Allen Discovery Center for Cell Lineage Tracing, Seattle, WA, USA. [6]Division of Biology and Biological Engineering, California Institute of Technology, Pasadena, CA, USA. [7]These authors contributed equally: Bailey A. T. Weatherbee, Carlos W. Gantner. ✉e-mail: mz205@cam.ac.uk

Aggregates of cells induced to overexpress extraembryonic factors and wild-type human ES cells are capable of self-organization into embryo-like structures, which mimic several hallmarks of post-implantation development, including lumenogenesis, amniogenesis, primordial germ cell formation and specification of the anterior hypoblast. Notably, our inducible human embryoids are modular, do not rely on exogenous signalling factors, and are amenable to genetic perturbation.

## Induction of extraembryonic lineages

We first identified factors that can similarly upregulate extraembryonic gene programmes in human ES cells. We integrated published single-cell RNA-sequencing data for human embryos cultured up to gastrulation[2,22–26] (Extended Data Fig. 1a–e) and used the computational tool SCENIC to score the predicted activity of transcription factors enriched in the epiblast, trophoblast or hypoblast (Extended Data Fig. 1f). As expected, we found that *SOX2*, *NANOG* and *POU5F1* (which encodes OCT4) showed high predicted activity in the epiblast. Transcription factors including *GATA4*, *GATA6*, *SOX17* and *FOXA2* were particularly active in the hypoblast and *GATA3*, *NR2F2*, *GATA2* and *TFAP2C* (which encodes AP2γ) showed enriched activity in the trophoblast (Extended Data Fig. 1f,g). Overexpression of *GATA6* or *SOX17* has been shown to drive endodermal gene programmes from primed human ES cells[27,28]; therefore, we selected these genes as candidates to program human ES cells to become hypoblast-like. Similarly, *GATA3* and *TFAP2C* have been reported to share high chromatin co-occupancy during the differentiation of human ES cells to trophoblast stem cells[29]. This, together with their high predicted activity in the trophoblast, led us to select *GATA3* and *TFAP2C* as candidates to drive human ES cells to become trophoblast-like. We generated and validated human ES cells with doxycycline-inducible individual or combined transgenes for the transcription factors of interest (Fig. 1a,b and Extended Data Fig. 1h).

The pluripotent state, in part, dictates differentiation potential from ES cells[30–32]. Therefore, to assess the capacity of the selected candidate transcription factors to drive human ES cells toward extraembryonic-like expression profiles, we overexpressed them—singly and in combination—in cells across the naive-to-primed pluripotency spectrum. We cultured cells using three established starting conditions: (1) PXGL, supporting pre-implantation-like cells; (2) RSeT, which generates intermediate peri-implantation-like cells; and (3) conventional mTeSR1 conditions to maintain post-implantation-like cells[33]. We observed significant differences in extraembryonic gene induction using both individual or combined transgenes and starting from different pluripotency states, at both the protein and mRNA levels (Extended Data Fig. 2). In hypoblast-like induction, GATA6 overexpression did not drive SOX17 expression from RSeT or PXGL conditions, but SOX17 overexpression resulted in robust GATA6 upregulation across starting pluripotency state conditions (Extended Data Fig. 2a,c,d). FOXA2 expression was consistently upregulated after combined GATA6 and SOX17 induction from the primed state and RSeT conditions, but not from PXGL conditions (Extended Data Fig. 2c,d). These data indicate that although GATA6 and SOX17 can indeed drive endodermal gene programmes, the regulation of specific downstream targets differs depending on the initial pluripotency state.

The AP2γ transgene appeared particularly effective at upregulating GATA2 and CK7 expression when driving trophoblast-like gene programmes. However, induction of AP2γ alone resulted in cell death and loss of transgene expression in primed cells, but not in RSeT or PXGL cells (Extended Data Fig. 2e,f). Combined induction of GATA6 and SOX17 or GATA3 and AP2γ resulted in consistent downregulation of pluripotency markers, including NANOG, SOX2 and OCT4 (Extended Data Fig. 2).

We hypothesized that RSeT human ES cells are the best starting cell type to generate our model of the human post-implantation embryo because they: (1) represent a peri-implantation stage of development[31]; (2) express low levels of amnion-specific genes after induction of GATA3 and AP2γ compared with primed cells (Extended Data Fig. 2b,e,f); and (3) are known to more readily differentiate to peri- and post-implantation yolk sac-like endoderm cells compared with PXGL cells[30,31]. For these reasons, and owing to the synergistic action of dual induction of candidate transcription factors, we used inducible GATA6-SOX17 and inducible GATA3-AP2γ RSeT human ES cells for induction of hypoblast-like and trophoblast-like cells, respectively, in subsequent experiments. Dual induction of GATA6 and SOX17 from RSeT cells in basal medium induced endodermal gene expression equivalent to directed differentiation protocols in yolk sac-like cell differentiation conditions (Extended Data Fig. 3a,b). Dual induction of GATA3 and AP2γ from RSeT cells in basal medium induced trophoblast gene expression, albeit at varying levels compared with directed trophoblast differentiation protocols (Extended Data Fig. 3c,d).

To further characterize RSeT human ES cells induced to express GATA6 and SOX17 or GATA3 and AP2γ, we carried out single-cell 10X multiome sequencing and assessed transcriptomic and chromatin accessibility simultaneously. Cells clustered on the basis of sample origin (Fig. 1c, Extended Data Fig. 4a and Supplementary Table 1). The application of a logistic regression framework showed that the wild-type, inducible GATA6-SOX17 and inducible GATA3-AP2γ RSeT human ES cells showed the highest similarities to the epiblast, hypoblast and cytotrophoblast of the post-implantation embryo, respectively (Fig. 1d). Further, compared to in vitro blastoid[11] and directed differentiation models[34], RSeT human ES cells showed similarity to the pluripotent population, inducible GATA6-SOX17 cells were similar to blastoid-derived hypoblast, and inducible GATA3-AP2γ cells showed similarity to post-implantation-like trophoblast stem cells, but not to blastoid-derived trophectoderm-like cells (Extended Data Fig. 4b). Analysis of differentially expressed genes and differentially accessible motifs revealed similar embryonic and extraembryonic dynamics (Fig. 1e, Extended Data Fig. 4c and Supplementary Table 2). Specifically, we detected enriched expression and motif accessibility scores of pluripotency and epiblast markers in RSeT human ES cells; of hypoblast markers in inducible GATA6-SOX17 cells; and of trophoblast markers in the GATA3-AP2γ inducible cells (Fig. 1e). Together, these data demonstrate that transcription factor-mediated induction of extraembryonic cell fate from RSeT human ES cells drives hypoblast- or trophoblast-like gene programmes without the need for exogenous factors, albeit heterogeneously and with some deficiencies in marker gene expression (Extended Data Fig. 4c). The ability of inducible cells to propagate in culture as stable cell lines was not tested. When aggregated with 8-cell-stage mouse embryos, the human cell contribution shifted towards SOX17-positive primitive endoderm and GATA3-positive trophectoderm for the inducible GATA6-SOX17 and GATA3-AP2γ cells, respectively, relative to wild-type controls (Extended Data Fig. 4d–g). If this relative shift towards extraembryonic identity was sufficient to allow for self-organization, it would overcome the challenge presented for successful co-culture of embryonic and extraembryonic-like cells by their conflicting culture media requirements. Indeed, 1:1:1 co-culture of wild-type RSeT human ES cells with inducible GATA6-SOX17 and GATA3-AP2γ RSeT human ES cells demonstrated good survival and mixed identity (Fig. 1f).

## Assembly of a 3D post-implantation model

As all three RSeT human ES cell-derived cell types—wild-type, inducible GATA6-SOX17 and inducible GATA3-AP2γ—can be cocultured in N2B27 medium, we induced expression of the selected transcription factors with doxycycline for three days and subsequently aggregated cell mixtures in Aggrewell dishes (Fig. 2a). Cells aggregated within 24 h and by 48 h post-aggregation, we observed clear distinctions between inner and outer cellular domains by brightfield microscopy (Fig. 2a). At 48 h post-aggregation, the medium was changed to post-implantation human embryo (hIVC1) medium[2,26]. Incubation with

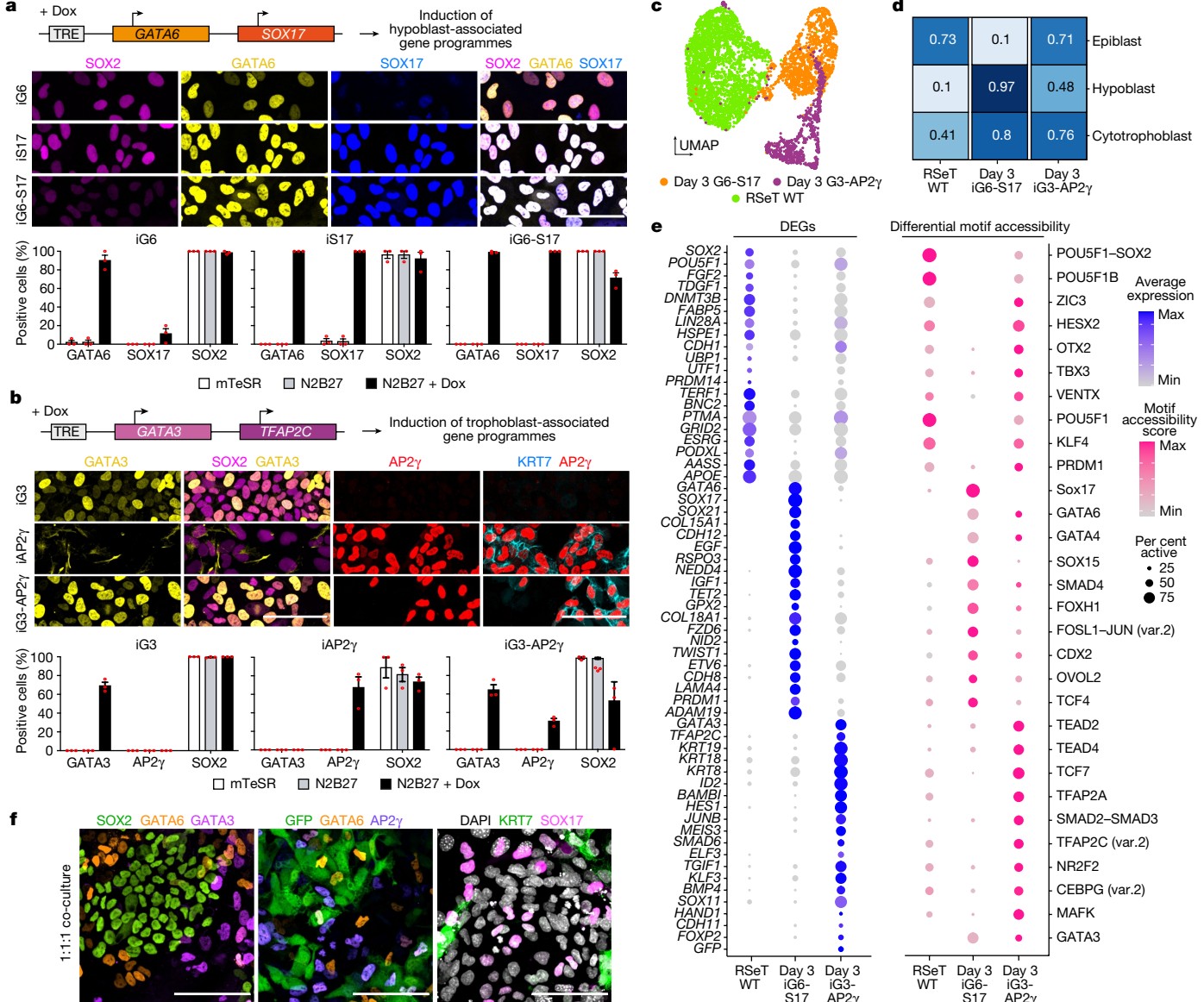

**Fig. 1 | Validation of extraembryonic-like induction. a**, Generation of human ES cells with inducible expression of GATA6 (iG6), SOX17 (iS17) or both (iG6-S17) and validation by immunofluorescence after 24 h of doxycycline (Dox) addition in basal N2B27 medium. iG6: $n = 551$; iS17: $n = 550$; iG6-S17: $n = 707$ cells from 3 independent experiments. TRE, tetracycline response element. **b**, Generation of human ES cells with inducible expression of GATA3 (iG3), AP2γ (iAP2γ) or both (iG3-AP2γ) and validation after 24 h of doxycycline addition in basal N2B27 medium. iG3: $n = 1,456$; iAP2γ: $n = 1,456$; iG3-AP2γ: $n = 782$ cells from 3 independent experiments. **a,b**, Data are mean ± s.e.m. **c**, Uniform manifold approximation and projection (UMAP)-based dimensional reduction of sequenced wild-type (RSeT WT), inducible GATA6-SOX17 (day 3 iG6-S17) and inducible GATA3-AP2γ

(day 3 iG3-AP2γ) RSeT human ES cells after 3 days of doxycycline induction. **d**, Logistic regression analysis (from ref. 47) and comparison of cells to human post-implantation embryo populations. Human embryo data from Molè et al.[2] were used as training data and the cell line data were used as test data. **e**, Selected differentially expressed genes (DEGs) from RNA sequencing (blue; left) and predicted differential motif accessibility from ATAC-seq scored by chromVAR (pink; right; var.2, secondary motif variation in JASPAR database) for wild-type, GATA6-SOX17 and GATA3-AP2γ RSeT human ES cells after three days of doxycycline induction. *TDGF1* is also known as *CRIPTO*. **f**, Validation of co-culture of wild-type, induced GATA6-SOX17 and induced GATA3-AP2γ RSeT human ES cells in 2D. $n = 3$ independent experiments. Scale bars, 100 μm.

doxycycline continued throughout the whole period of culture, and proliferation was consistent across experiments (Fig. 2a,b). Four days post-aggregation, the cell aggregates had self-organized into structures with a SOX2-positive, epiblast-like domain containing a central lumen, an outer single layer of GATA3-positive putative trophoblast-like cells, and an intermediate putative hypoblast-like domain of GATA6-positive cells between inner lumenized domain and outer layer (Fig. 2c and Extended Data Fig. 5a).

Similar to models of post-implantation mouse embryos, aggregates did not transit through a blastocyst-like morphology before forming post-implantation-like structures[20,21]. The efficiency of inducible

human embryoid formation (defined as aggregates containing an organized SOX2-positive domain, surrounded by concentric layers of GATA6-positive and GATA3-positive cells) was approximately 23% (Fig. 2d). By contrast, when using primed mTeSR1 or naive PXGL human ES cells as the starting pluripotency state for constitutive wild-type, inducible GATA6-SOX17 and GATA3-AP2γ cells, the efficiency of organized, multi-lineage structure formation was less than 5% (Fig. 2d,e). Organized embryo-like structures exhibited an organization reminiscent of the human embryo at 8 to 9 dpf (Fig. 2f).

Our inducible human embryoids expressed several other lineage markers in an organized manner, including N-cadherin, SOX17, and

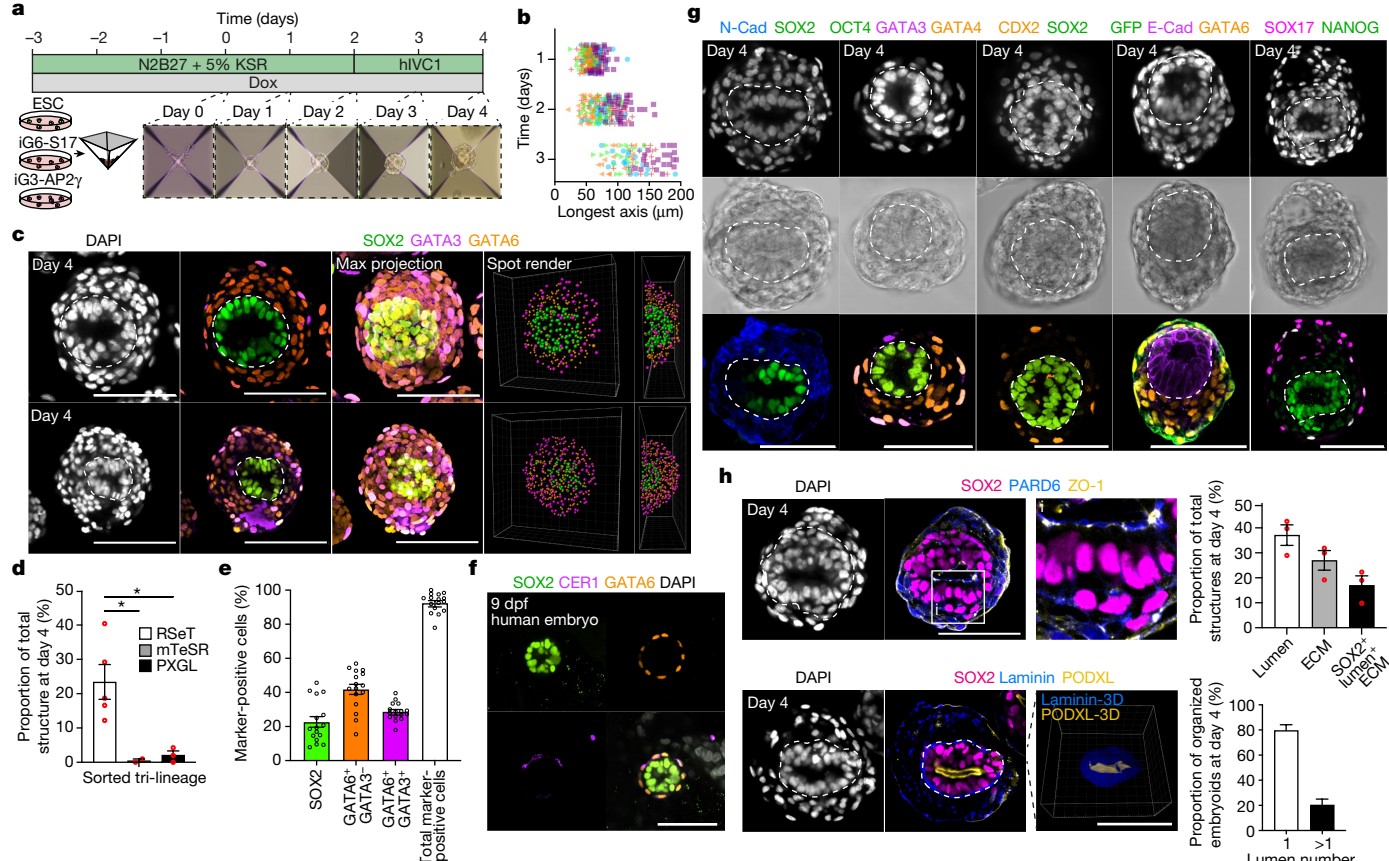

**Fig. 2 | Generation of inducible post-implantation human embryoids.**
**a**, Overview of the protocol used to generate inducible human embryoids by combining wild-type RSeT human ES cells with inducible GATA6-SOX17 and GATA3-AP2γ cells. Extraembryonic-like cells were induced for 3 days before aggregation at day 0. KSR, knockout serum replacement. **b**, Size of cell aggregates between days 1 and 3 post-aggregation. Day 1: $n = 175$, day 2: $n = 171$ and day 3: $n = 91$ structures from 5 independent experiments. All individual embryoid lengths are plotted. Each symbol (orange cross, orange triangle, green triangle, blue circle, purple square) represents an independent experiment. **c**, At 96 h post-aggregation, structures demonstrate clear self-organization. **d**, Quantification of embryoid formation across starting pluripotency states. RSeT: $n = 952$, mTeSR: $n = 30$ and PXGL: $n = 207$ structures from 5 independent experiments. One-way ANOVA with Holm–Šídák multiple comparisons test. RSeT versus mTeSR: $P = 0.0347$; RSeT versus PXGL, $P = 0.0283$. Unmarked pairwise comparisons are not significant ($P > 0.05$). **e**, Proportions of cell types in correctly organized embryoids. $n = 16$ embryoids from 3 independent experiments. **f**, Representative image of an in vitro cultured human embryo at 9 dpf showing a clear lumenized SOX2 domain surrounded by a layer of GATA6-positive cells. A subset of GATA6-positive cells expresses the anterior hypoblast marker CER1. Representative of three independent experiments. **g**, The hypoblast-like domain expresses N-cadherin (N-cad), SOX17 and GATA4, and the epiblast-like domain maintains expression of the pluripotency factors SOX2, OCT4 and NANOG. Cells derived from GATA3-AP2γ cells (which express GFP) show clear outer localization. Representative of two experiments each. E-cad, E-cadherin. **h**, Inducible human embryoids demonstrate clear apicobasal polarity. Right, quantification of inducible human organization. Top right, $n = 506$ structures from 3 independent experiments for lumen and extracellular matrix (ECM) efficiency. Bottom right, $n = 27$ embryoids from 2 independent experiments for lumen number. **g**,**h**, Inner domains of embryoids are surrounded by a dashed line. **d**,**e**,**h**, Data are mean ± s.e.m. Scale bars, 100 μm. *$P < 0.05$, **$P < 0.01$, ***$P < 0.001$, ****$P < 0.0001$. NS, not significant.

GATA4 in the putative hypoblast-like compartment (Fig. 2g). We also observed structures with SOX17 and/or GATA6 expression within outer GATA3-AP2γ-induced cells (marked by eGFP), which may reflect the reported tendency of peripheral cells to adopt endodermal identities in embryoid bodies[35,36]. The epiblast-like inner compartment expressed SOX2, NANOG, E-cadherin and maintained pluripotent and epithelial identity akin to the human embryo (Fig. 2g). Further, this inner domain exhibited apicobasal polarity with basal deposition of laminin and apical expression of PODLX, PARD6 and ZO-1 (Fig. 2h). These data demonstrate that embryo-like structures derived from RSeT human ES cells can self-organize in minimal media conditions.

## Differentiation within embryoids

To gain insight into whether our human embryo-like model develops gene expression and chromatin accessibility patterns that reflect the natural human embryo, we performed single-cell multiome RNA sequencing and assay for transposase-accessible chromatin with sequencing (ATAC-seq) at 4, 6 and 8 dpf (Fig. 3a). We selected individual structures for sequencing on the basis of their development of the three tissues: (1) an inner, epithelial domain; (2) an intermediate domain surrounding the central epithelium; and (3) an outer, GFP-positive cell layer (Extended Data Fig. 5b,c). To assign clusters without bias, we used scmap to project our dataset onto human[2,37] and cynomolgus macaque[38–40] (*Macaca fascicularis*) datasets spanning peri-implantation to gastrula stages (Fig. 3b and Extended Data Fig. 5d). This analysis allowed us to project gene-expression signatures from previously annotated cynomolgus macaque cell type clusters onto our model of the human embryo. Using multiome-based velocity inference, we found that the inferred differentiation time correlated well with the transition of structures from day 4 to day 8 post-aggregation. These data, in combination with canonical marker expression, enabled us to annotate the cell types in our human embryo-like structures (Fig. 3b, Extended Data Fig. 5e,f and Supplementary Table 3). We identified

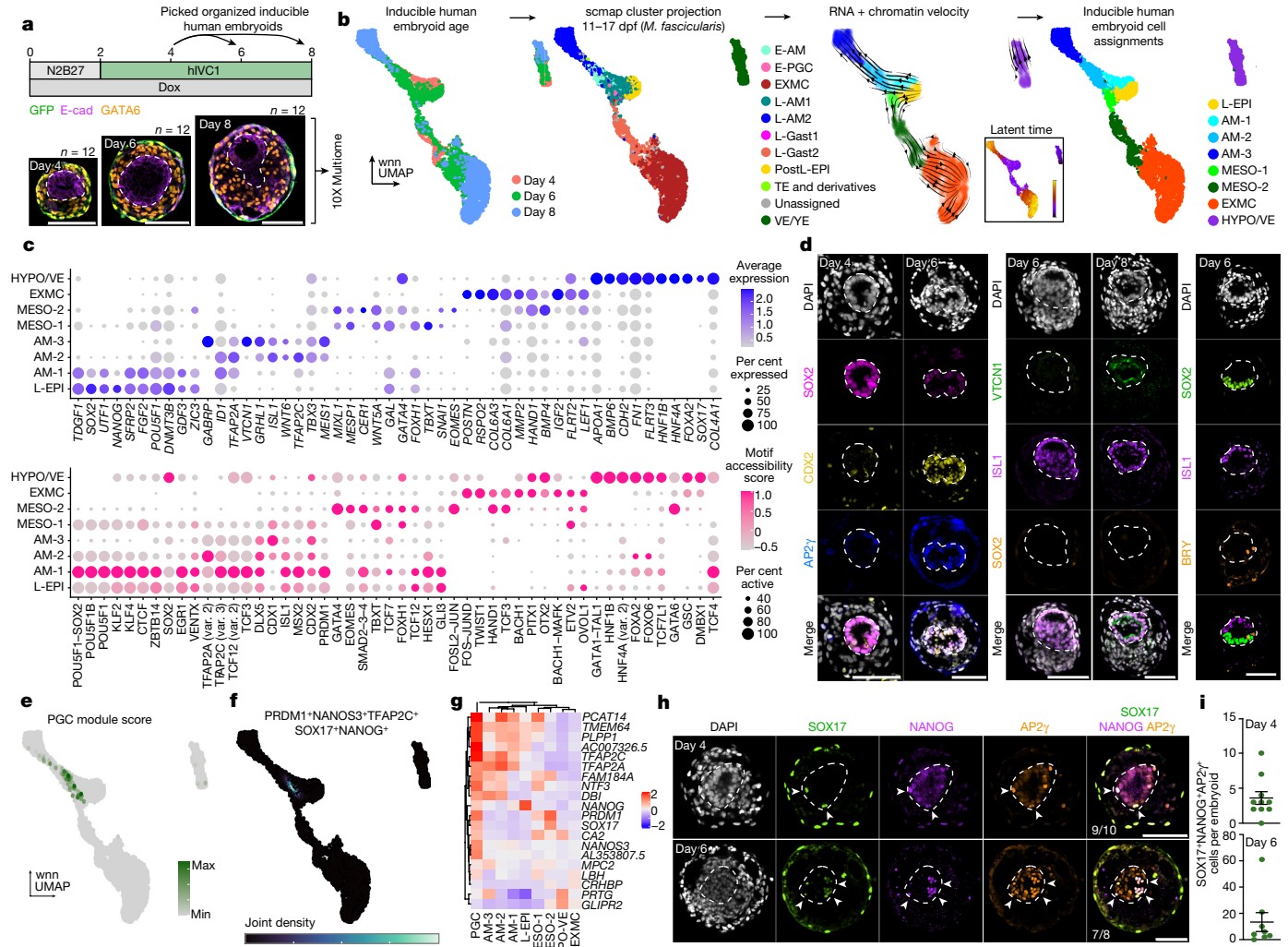

**Fig. 3 | Differentiation of extraembryonic mesenchyme, amnion and primordial germ cells. a**, Schematic of the extended culture protocol of inducible human embryoids and sampling for combined-single-cell RNA sequencing and single-cell ATAC-seq using the 10X platform. Twelve embryoids each at day 4, 6 and 8 post-aggregation were sequenced. **b**, Cells were annotated on the basis of transcriptional projection to multiple human and non-human primate (*M. fascicularis*) embryo datasets using scmap in conjunction with RNA and chromatin velocity. wnn, weighted nearest neighbour. E-PGC, early PGC; E-AM, early amnion; L-AM1, late amnion 1; L-AM2, late amnion 2; L-Gast1, late gastrulating epiblast 1; L-Gast2, late gastrulating epiblast 2; TE, trophectoderm; VE/YE, visceral/yolk sac endoderm. **c**, Selected differentially expressed genes in the RNA-sequencing data (top) and predicted differentially accessible motifs scored by chromVAR on the ATAC-seq data (bottom) across clusters. **d**, Inducible human embryoids downregulated SOX2

and upregulated CDX2 and ISL1 at day 6, and upregulated VTCN1 at day 8, indicative of robust amnion differentiation and maturation. In some, rare cases, dorsoventral and/or anterior–posterior axis patterning is observable. Representative of three experiments. **e**, Module scoring for primordial germ cell marker genes[44]. **f**, Nebulosa plot visualizing joint expression density of key primordial germ cell genes in inducible human embryoids. Max, maximum; min, minimum. **g**, Heat map of selected primordial germ cell gene expression across clusters. **h**, Immunofluorescence identification of SOX17⁺NANOG⁺AP2γ⁺ primordial germ cell-like cells in inducible human embryoids highlighted by arrowheads. **i**, Quantification of SOX17⁺NANOG⁺AP2γ⁺ cells at days 4 (*n* = 10 embryoids) and 6 (*n* = 10 embryoids). Two independent experiments. Data are mean ± s.e.m. Inner domains of embryoids are surrounded by a dashed line. Scale bars, 100 μm.

clusters resembling embryonic late-epiblast (L-EPI), amnion (AM-1, AM-2 and AM-3), mesoderm (MESO-1 and MESO-2), extraembryonic mesenchyme (EXMC) and hypoblast/visceral endoderm (HYPO/VE) (Fig. 3c and Extended Data Fig. 5e,f). These assigned clusters showed differences in their composition based on the day of sample collection, with the L-EPI cluster comprising only day 4 and day 6 structures, a progressive shift from AM-1 to AM-2 and AM-3 over time, and similarly from mesoderm to EXMC (Extended Data Fig. 5f).

Finally, we directly compared our clusters with previously annotated peri-implantation and peri-gastrulation cynomolgus macaque and human embryo datasets[2,37–40]. This analysis demonstrated similarity between inducible human embryoid clusters and primate embryos (Extended Data Fig. 6a,b). Similarly, inducible human embryoid clusters

aligned with in vitro human models derived from ES cells, including post-implantation amniotic sac embryoids[15], blastoids[11] and the recently identified extraembryonic mesenchyme-like cells generated during trophoblast stem cell-like directed differentiation[34] (Extended Data Fig. 6b,c) and we observed significant similarities between in vitro amnion, hypoblast and extraembryonic mesenchyme populations compared with embryo-like structures. We did not identify a distinct trophoblast-like cluster derived from the GFP-positive inducible GATA3-AP2γ cells, despite their presence as an outer layer within the inducible human embryoids (Fig. 3a and Extended Data Fig. 6d). Given the aberrant upregulation of endodermal markers after aggregation, inducible GATA3-AP2γ-derived cells were not likely to represent bona fide trophoblast. Nevertheless, inducible human embryoids

generated several cell types that did not robustly differentiate when human embryos were cultured in vitro to post-implantation stages, including amnion and extraembryonic mesenchyme. Indeed, immuno-fluorescence analysis demonstrated that the inner, SOX2-positive domain upregulated amnion markers, including CDX2 and ISL1 by day 6 post-aggregation. By day 8 post-aggregation, the inner domain expressed the mature amnion markers VTCN1 and HAND1[39,41] (Fig. 3d and Extended Data Fig. 7a,b), correlating to the transition between AM-1, AM-2 and AM-3. The GATA6-positive domain also expressed HAND1, supporting the presence of extraembryonic mesenchyme[34] (Extended Data Fig. 7a,b). A subset of GATA6-positive cells showed high co-expression of TBX20, further highlighting the presence of extraembryonic mesenchyme in this intermediate region[34,39] (Extended Data Fig. 7c,d). In the majority of our human embryoids, the entire epiblast-like domain differentiated towards an amnion fate. However, we also observed rare cases of embryo-like structures at days 6 to 8 post-aggregation that exhibited dorsoventral and/or anterior–posterior symmetry breaking, with regionalized ISL1, SOX2 and Brachyury (also known as TBXT) expression (Fig. 3d).

Recent reports have postulated that both amnion and primordial germ cells, the precursors to gametes, are at least partially generated from a bipotent progenitor[42,43]. We therefore explored whether such progenitors or their progeny were specified in our human embryo-like model. We first assigned a primordial germ cell module score on the basis of expression of genes identified in human primordial germ cell-like cells differentiated in vitro[44] and could identify cells with transcriptomes resembling those of primordial germ cell-like cells (Fig. 3e,f). We labelled cells in the 98th percentile of primordial germ cell-like cell gene-expression module scores as putative primordial germ cell-like cells (PGC). Primordial germ cell-like cells expressed *TFAP2A* (which encodes AP2α), a crucial marker of bipotent amnion and primordial germ cell-like cell progenitors[42,43]. In contrast to other cells in AM-1 and AM-2 clusters, primordial germ cell-like cells expressed the pluripotency marker *NANOG* and primordial germ cell markers *PRDM1* (also known as *BLIMP1*) and *NANOS3* (Fig. 3g). Immunofluorescence analysis of a canonical set of human primordial germ cell markers[42] confirmed that AP2γ, SOX17 and NANOG triple-positive primordial germ cell-like cells were observed by day 4 post-aggregation and increased in number by day 6 (Fig. 3h,i and Extended Data Fig. 7f). These data demonstrate that robust primordial germ cell-like cell specification is concomitant with amnion-like cell formation and occurs within the inner epiblast-like compartment, supporting the existence of a bipotent progenitor for these two lineages.

We next utilized Shef6-mKate wild-type ES cells to confirm the differentiation trajectories within embryoids. At day 4 post-aggregation, the majority of the wild-type cells contributed to the inner SOX2-positive epiblast-like domain, with some contribution to the GATA6-positive population (Extended Data Fig. 7g), in line with a small proportion of early extraembryonic mesenchyme differentiation in our sequencing analysis (Extended Data Fig. 5f). By day 6 post-aggregation, wild-type cells contributed to ISL1-positive amnion-like domain, as well as GATA6 and TBX20-positive extraembryonic mesenchyme-like cells (Extended Data Fig. 7h). Finally, we confirmed the derivation of AP2γ, SOX17 and NANOG triple-positive primordial germ cell-like cells from the mKate2-labelled wild-type population (Extended Data Fig. 7i). These data confirm that the epiblast-like domain differentiates toward several post-implantation lineages.

## BMP mediates epiblast differentiation

Amnion, primordial germ cells and extraembryonic mesenchyme are thought to differentiate in response to BMP signalling in the primate embryo[34,38]. To understand whether this is consistent in our human embryo-like model, we examined the expression of the downstream BMP response genes *ID1–ID4*. *ID1* and *ID4* were upregulated during amnion formation, whereas *ID2* and *ID3* were enriched in both amnion and extraembryonic mesenchyme trajectories, indicating that BMP signalling was probably active (Fig. 4a and Extended Data Fig. 8a). Additionally, SMAD5 motif accessibility was high in both trajectories whereas SMAD2–SMAD3–SMAD4 motif accessibility score, a downstream target of activin–NODAL signalling, was not (Fig. 4b and Extended Data Fig. 8b). In line with this observation, a high BMP and low NODAL signalling environment has been implicated recently in amnion differentiation of marmoset ES cells[41] and in human ES cells during extraembryonic mesenchyme differentiation[34], suggesting that similar dynamics may drive differentiation of these populations within inducible human embryo-like structures.

To further understand potential tissue–tissue crosstalk in our human embryo-like model, we used the computational tool CellPhoneDB to predict ligand–receptor pairing across clusters in our single-cell sequencing data (Extended Data Fig. 8c,d). This analysis uses the expression of curated receptor–ligand pairs across clusters to score potential tissue–tissue crosstalk. CellPhoneDB predicted that hypoblast cluster-derived BMP2–BMP6 and extraembryonic mesenchyme-secreted BMP4 were probable mediators of tissue–tissue crosstalk. By contrast, predicted NODAL signalling between tissues was low, further supporting the presence of a high-BMP, low NODAL signalling environment in the human embryo-like model. When CellPhoneDB was applied to the single-cell sequencing data of the three cell lines aggregated to generate the embryo-like model, the inducible GATA3-AP2γ cells were predicted to be the initial source of BMP (Extended Data Fig. 8d). Aggregation of inducible GATA6-SOX17 and wild-type RSeT human ES cells alone (that is, without inducible GATA3-AP2γ cells) or addition of the ALK1, 2, 3 or 6 (type I BMP receptors) inhibitor LDN193189 between days 0 and 2 blocked formation of organized structures (Extended Data Fig. 8e–g), demonstrating the necessity of inducible GATA3-AP2γ-cell secreted BMP during embryoid formation.

To verify the role of BMP signalling during differentiation of the epiblast-like domain, we examined phosphorylated (p)SMAD1.5 expression. pSMAD1.5 enrichment in the OCT4-positive epiblast-like domain at days 4 and 6 post-aggregation was indicative of active BMP signalling (Fig. 4c). By contrast, these cells had a low nuclear/cytoplasmic ratio of total SMAD2.3, reflecting low NODAL signalling within the epiblast-like domain (Fig. 4d) and in accord with the CellPhoneDB predictions. To functionally validate the role of BMP signalling in the differentiation of the inner domain, we treated the human embryo-like model with LDN193189 between 48 h and 96 h post-aggregation. Treated structures exhibited increased maintenance of SOX2 expression in the inner domain and reduced CDX2 and AP2α upregulation at day 4 and day 6 post-aggregation compared with untreated control or BMP4-treated structures. Supplementation with activin-A, an agonist of SMAD2.3 signalling, resulted in a similar phenotype, although to a lesser degree (Fig. 4e,f and Extended Data Fig. 8h). Additionally, LDN193189 treatment decreased the number of primordial germ cell-like cells, whereas BMP4 or activin-A treatment had minimal effect on the emergence of the this population (Fig. 4g,h). These data demonstrate that endogenous BMP and NODAL are key drivers of amnion and primordial germ cell-like cell differentiation from the epiblast-like domain within inducible embryoids.

## SOX17 inhibits anterior hypoblast

BMP signals are localized to the posterior of the embryo by the antagonistic action of the anterior hypoblast, which secretes inhibitors of BMP, WNT and NODAL, including CER1 and LEFTY1[2] (Fig. 2f). These markers of the anterior hypoblast are expressed in peri- and post-implantation human embryos cultured in vitro[2]. Neither *CER1* nor *LEFTY1* was meaningfully expressed in the HYPO/VE single-cell sequencing cluster (Fig. 5a). Reanalysis of our previously published 10X single-cell RNA sequencing data from in vitro cultured post-implantation human

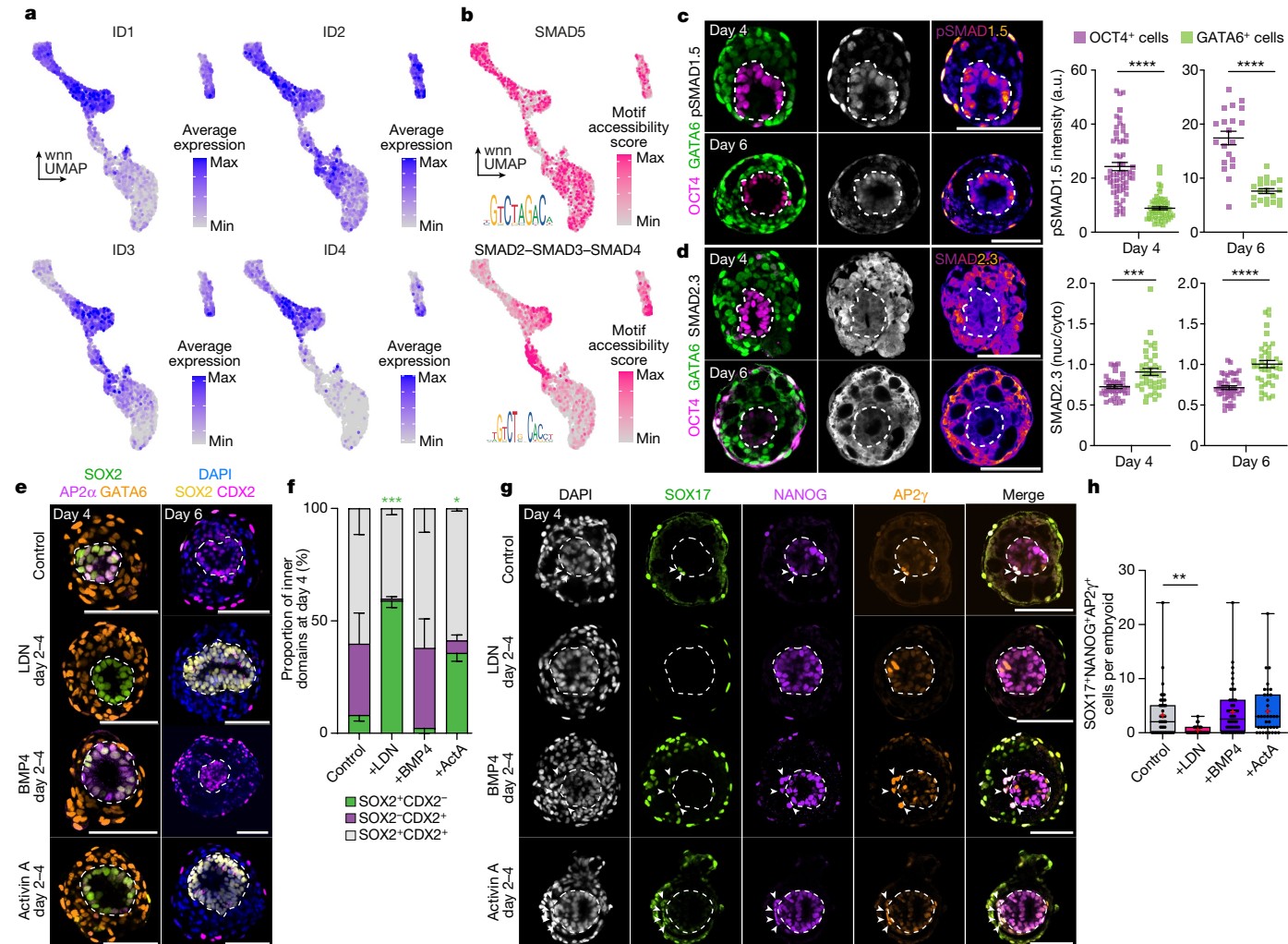

**Fig. 4 | BMP signalling drives amnion specification in inducible human embryoids. a**, Expression of ID1–4, downstream targets of BMP signalling, in embryoids. **b**, chromVAR-based motif accessibility scores of SMAD5 and SMAD2–SMAD3–SMAD4, effectors of BMP and NODAL signalling, respectively. **c**, Representative images and quantification of OCT4-positive and GATA6-positive cells from representative inducible human embryos at days 4 ($n = 60$ cells each; $P < 0.0001$) and 6 ($n = 40$ cells each; $P < 0.0001$). Three independent experiments. a.u., arbitrary units. **d**, Representative images and quantification of SMAD2.3 in OCT4-positive and GATA6-positive cells from representative inducible human embryos at days 4 ($n = 40$ cells each; $P = 0.0004$) and 6 ($n = 40$ cells each; $P < 0.0001$). Two independent experiments. Cyto, cytoplasm; nuc, nuleus. **c**,**d**, Two-sided Mann–Whitney test. **e**, Inhibition of BMP signalling blocks exit from pluripotency and upregulation of amnion markers AP2α and CDX2. **f**, The proportion of day 4 inner domains expressing SOX2 and CDX2 (control: $n = 147$, LDN193189 (LDN)-treated: $n = 126$, BMP4-treated: $n = 57$ and

activin-A (ActA)-treated: $n = 60$ embryoids from 5 independent experiments). For SOX2$^+$CDX2$^-$ domains, control versus LDN193189: $P = 0.0002$; control versus activin-A: $P = 0.0433$; control versus BMP4: $P = 0.1753$. Repeated measures two-way ANOVA with Holm–Šídák multiple comparisons test. **g**, Inhibition of BMP reduces the number of primordial germ cell-like cells in embryoids. **h**, The number of SOX17$^+$NANOG$^+$AP2γ$^+$ primordial germ cell-like cells at day 4 (control: $n = 45$, LDN193189-treated: $n = 30$, BMP4-treated: $n = 48$ and activin-A-treated: $n = 36$ embryoids from 6 independent experiments). In box plots, the centre line is the median, boxes encompass the 25th–75th centiles, whiskers extend to the minimum and maximum values and the plus sign denotes the mean. Control versus LDN193189: $P = 0.0011$; control versus BMP4: $P > 0.99$; control versus activin-A: $P = 0.98$. Unmarked comparisons to control are not significant. Kruskal–Wallis with Dunn's multiple comparisons test. **c**,**d**,**f**, Data are mean ± s.e.m. Inner domains of embryoids are surrounded by a dashed line. Scale bars, 100 μm.

embryos revealed that SOX17 regulon activity was significantly enriched in the *CER1*-negative hypoblast subcluster (Fig. 5b and supplementary data table 8 of ref. 2). Indeed, induction of SOX17 singly or in combination with GATA6 resulted in decreased capacity to upregulate *CER1* compared with GATA6 overexpression alone (Extended Data Fig. 1h). To test whether single induction of GATA6 changed the identity of hypoblast-like cell subpopulations in our human embryo-like model, we generated hypoblast-like cells with inducible GATA6 or SOX17 expression individually or in combination. An increased proportion of CER1-positive cells were observed within embryoids derived with inducible GATA6 cells alone compared with embryoids generated with inducible GATA6 plus SOX17 or SOX17 cells (Fig. 5c,d and Extended Data

Fig. 9a). To validate that SOX17 induction inhibits CER1 expression, we withdrew doxycycline at 1 or 3 days post-aggregation. Withdrawal at day 3—but not at day 1—promoted CER1 expression in embryoids (Fig. 5c and Extended Data Fig. 9a). Staining of pSMAD1.5 together with CER1 showed significantly decreased pSMAD1.5 expression in the inner-domain cells of structures with a CER1-positive cell population (Fig. 5e,f). By day 6 post-aggregation, CER1 expression decreased in all embryoids, regardless of the initial hypoblast induction regime (Extended Data Fig. 9b). However, transient CER1 expression in anterior hypoblast-like cells influenced the epiblast-like domain within embryoids. Embryoids generated with single GATA6 induction or doxycycline withdrawal at day 3 showed increased expression of the primitive streak

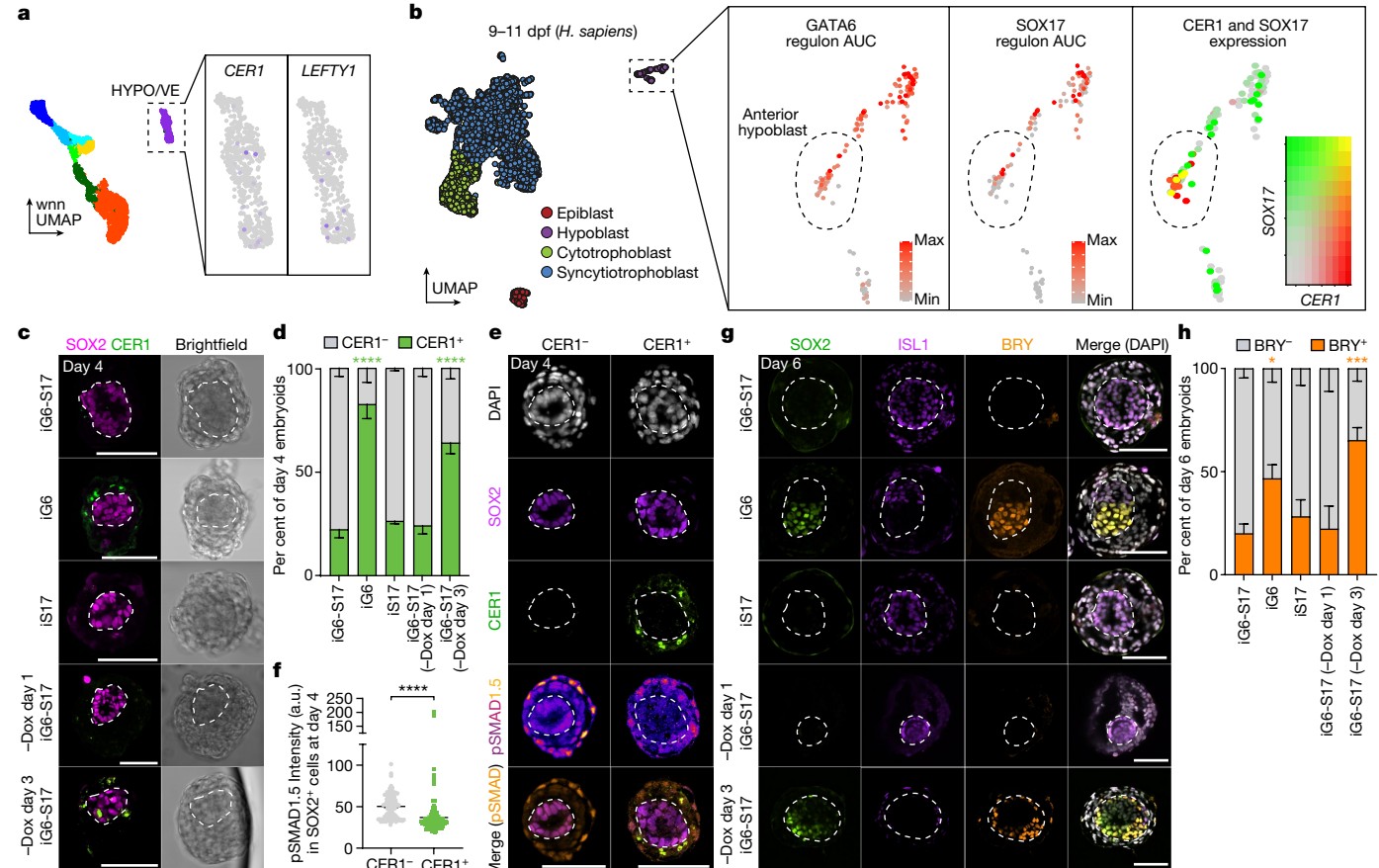

**Fig. 5 | SOX17 induction is antagonistic to specification of the anterior hypoblast. a**, Expression of *CER1* and *LEFTY1* in the HYPO/VE in embryoids. **b**, Analysis of GATA6 and SOX17 regulon activity scored by SCENIC, and *SOX17* and *CER1* co-expression in post-implantation human hypoblast (9–11 dpf). Data are from Molè et al.[2]. AUC, area under the curve. **c**, Representative examples of embryoids showing CER1-positive cells generated using induced GATA6 (iG6) but not induced GATA6-SOX17 (iG6-S17) hypoblast-like cells. CER1 expression is also observed if doxycycline is withdrawn at day 3 post-aggregation. **d**, The proportion of embryoids in **c** expressing CER1. iG6-S17: *n* = 78; iG6: *n* = 25; iS17: *n* = 26; iG6-S17 −Dox, day 1: *n* = 54; and iG6-S17 −Dox, day 3: *n* = 65 embryoids from 7 independent experiments. iG6-S17 versus iG6: *P* < 0.0001; iG6-S17 versus iG6-S17 −Dox, day 3: *P* < 0.0001; iG6-S17 versus iS17, *P* = 0.87; iG6-S17 versus iG6-S17 −Dox, day 1: *P* = 0.87. **e**, Representative images at day 4 demonstrate decreased pSMAD1.5 in the epiblast-like domain of structures with a CER1-positive cell population. **f**, Quantification of pSMAD1.5 levels in SOX2-positive cells in CER1-negative versus CER1-positive iG6-S17 embryoids at day 4. CER1⁻: *n* = 108 cells and CER1⁺: *n* = 123 cells from 8 embryoids each from 2 independent experiments. *P* < 0.0001. Two-way Mann–Whitney test. **g**, Representative images of Brachyury (BRY) expression in inducible human embryoids generated with iG6, iS17 or iG6-S17 cells (with doxycycline maintained, or removed at day 1 or day 3 post-aggregation). **h**, Quantification of Brachyury expression in **e**. iG6-S17: *n* = 34; iG6: *n* = 15; iS17: *n* = 16; iG6-S17 −Dox, day 1: *n* = 16; and iG6-S17 −Dox, day 3: *n* = 20 embryoids from 6 independent experiments. iG6-S17 versus iG6: *P* = 0.0225; iG6-S17 versus iG6-S17 −Dox, day 3: *P* = 0.0002; iG6-S17 versus iS17: *P* = 0.69; iG6-S17 versus iG6-S17 −Dox, day 1: *P* = 0.81. **d,f,h**, Data are mean ± s.e.m. **d,h**, Repeated measures two-way ANOVA with Holm–Šídák multiple comparisons test. Inner domains of embryoids are surrounded by a dashed line. Scale bars, 100 μm.

marker Brachyury at day 6 post-aggregation compared with structures with consistent GATA6 plus SOX17 or SOX17 induction (Fig. 5g,h and Extended Data Fig. 9c).

Together, these data demonstrate that functional differences in the gene regulatory network underlying differentiation of hypoblast subpopulations can be observed in embryoids and show an inhibitory role of prolonged SOX17 overexpression on CER1-positive anterior hypoblast identity. These experiments highlight the value of modular embryoid models for studying interactions between embryonic and extraembryonic tissues.

## Discussion

Here we have generated a multi-lineage stem cell-derived model of the human post-implantation embryo that undergoes epiblast-like domain lumenogenesis and differentiation that reflects developmentally relevant interactions between extraembryonic-like and embryonic-like tissues. Our stem cell-derived inducible model of the human embryo

generates amnion-like cells in response to BMP signalling that progressively mature. Similarly, primordial germ cell-like cells are readily differentiated in our stem cell model of the human embryo, and we present evidence that these cells are specified along the amnion differentiation trajectory, probably originating from a common AP2α-positive progenitor, as reported in other in vitro systems[42,43]. We also observe extraembryonic mesenchyme-like cells, which closely resemble those of the primate embryo. Our analyses suggest a trajectory from the late-epiblast-like population through a mesodermal intermediate, in line with a recently reported in vitro extraembryonic mesenchyme differentiation protocol[34,39], data from cynomolgus macaque[39] and historical observations in rhesus macaque and human embryos[45].

Unexpectedly, modulation of the transgenes used to drive hypoblast-like identity alters the balance of hypoblast contribution from CER1-negative to CER1-positive hypoblast-like cells, demonstrating that SOX17 overexpression blocks the formation of CER1-positive anterior hypoblast. NODAL signalling is required for the formation of CER1-positive mouse anterior visceral endoderm and SOX17 may

restrain excessive NODAL activity and antagonize its targets[46]. The low NODAL activity observed in inducible human embryoids, compounded by SOX17 overexpression may, in part, explain the low levels of anterior hypoblast formation in embryoids. Additionally, the low NODAL environment, combined with the lack of anterior hypoblast, may contribute to the differentiation of the epiblast-like population over time.

In our human embryo models with CER1-positive hypoblast, we observed a noticeable increase in primitive streak-like Brachyury expression despite the loss of the CER1-positive cells by day 6. We hypothesize that the transient presence of an anterior hypoblast-like population protects epiblast-like domain pluripotency for a longer period, allowing cells to exit pluripotency at a developmentally later, gastrulation-competent cell stage. These results contrast with embryoids lacking an anterior hypoblast-like population, which predominantly generate amnion. These data point to the potential existence of a distinct intermediate pluripotent state that is able to give rise to both amnion and extraembryonic mesenchyme but not germ layer derivatives.

The modular generation of integrated embryoids from their constitutive parts will be useful in interrogating the role of specific tissues and tissue-specific gene requirements. However, the use of transcription factor overexpression to generate extraembryonic tissues may also lead to deficiencies in differentiation. For example, although GATA3–AP2γ induction drives trophoblast-like gene programmes in 2D, upon aggregation in our human embryo-like model, this cell population aberrantly upregulates endodermal markers (including SOX17 and GATA6). Nevertheless, the GATA3–AP2γ inducible cells are required for successful organization of an embryo-like structure and are likely to act as a crucial source of BMP. The initial pluripotent state is a crucial factor in the induction of downstream gene regulatory networks, and we provide evidence that embryo models form efficiently using peri-implantation stage human ES cells, but not more naive human ES cells. However, given that induction of trophoblast gene networks appears more robust in naive human ES cells, the use of discordant pluripotent cells may generate embryo models that better recapitulate the embryo. Likewise, different combinations of transcription factors may be necessary for lineage specification from different starting states. Thus, further work interrogating the epigenetic landscape and binding sites of these factors may be useful to improve strategies to generate bona fide extraembryonic cells.

In summary, we present a modular model of human post-implantation development that includes both embryonic- and extraembryonic-like cells. Our post-implantation embryo models self-organize and, in rare cases, show axis formation. Further optimization is needed to permit the maintenance of all major lineages of the post-implantation embryo with their more complete differentiation potential and embryo-like morphology. As this model cannot implant, it does not have the capacity to develop towards fetal stages and does not mimic stages beyond primitive streak formation or contain all cell types of the gastrulation-stage embryo[3]. However, the construction of these integrated models of the post-implantation human embryo is an important step towards mechanistic studies of post-implantation development that are impossible to perform in the real human embryo.

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

## Methods

### Ethics statement

Work with human embryonic stem cells (Shef6) was carried out with approval from the UK Human Stem Cell Bank Steering Committee under approval SCSC21-38 and adheres to the regulations of the UK Code of Practice for Use of Human Stem Cell Lines. Human embryo work was regulated by the Human Fertility and Embryology Authority (HFEA) and carried out under license R0193. Ethical approval was obtained from the Human Biology Research Ethics Committee at the University of Cambridge (reference HBREC.2021.26). Patients undergoing IVF at CARE Fertility, Bourn Hall Fertility Clinic, Herts and Essex Fertility Clinic, and King's Fertility were given the option of continued storage, disposal, or donation of embryos to research (including project specific information) or training at the end of their treatment. Patients were offered counselling, received no financial benefit, and could withdraw their participation at any time until the embryo had been used for research. Informed research consent for donated embryos was obtained from both gamete providers. Only blastocysts that exhibited appropriate morphology (that is, expanded blastocoel cavity and healthy inner cell mass) were used for subsequent experimentation. Embryos were not cultured beyond 14 dpf or the first appearance of the primitive streak. Mice were kept in an animal house on 12:12 h light-dark cycle with ad libitum access to food and water. Experiments with mice are regulated by the Animals (Scientific Procedures) Act 1986 Amendment Regulations 2012 and conducted following ethical review by the University of Cambridge Animal Welfare and Ethical Review Body (AWERB). Experiments were approved by the Home Office under Licenses 70/8864 and PP3370287. CD1 and $F_1$ wild-type males aged 6 to 45 weeks and CD1 and $F_1$ wild-type females aged 6 to 18 weeks were used for this study. Animals were inspected daily and those showing health concerns were culled by cervical dislocation. All embryo and embryoid work was performed in the UK and adheres to the 2021 ISSCR guidelines[3].

### Human ES cell culture

Shef6 (from the UK Stem Cell Bank) or RUES2 (kindly provided by A. Brivanlou, The Rockefeller University) human ES cells were cultured on Matrigel-coated plates in mTESR medium (05825, STEMCELL Technologies) at 37 °C with 20% $O_2$ and 5% $CO_2$. Plates were coated with 1.6% growth factor-reduced Matrigel (356230, BD Biosciences) dissolved in DMEM/F12 (21331-020, Life Technologies) for 1 h at 37 °C. Human ES cells were passaged with TrypLE (12604013, Thermo Fisher Scientific). For the first 24 h after passaging, 10 µM ROCK inhibitor Y-27632 (72304, STEMCELL Technologies) was added. Medium was changed every 24 h. Cells were routinely tested for mycoplasma contamination by PCR (6601, Takara Bio), and have been authenticated by short tandem repeat analysis. To convert primed human ES cells to RSeT or PXGL culture conditions, cells were passaged to mitomycin-C inactivated CF-1 mouse embryonic fibroblasts ($3 \times 10^3$ cells per cm²; GSC-6101G, Amsbio) in media consisting of DMEM/F12 with 20% Knockout Serum Replacement (10828010, Thermo Fisher Scientific), 100 µM β-mercaptoethanol (31350-010, Thermo Fisher Scientific), 1× GlutaMAX (35050061, Thermo Fisher Scientific), 1× non-essential amino acids, 1× penicillin-streptomycin and 10 ng ml⁻¹ FGF2 (Department of Biochemistry, University of Cambridge) and 10 µM ROCK inhibitor Y-27632 (72304, STEMCELL Technologies). For RSeT cells, the medium was switched to RSeT medium after 24 h (05978, STEMCELL Technologies). Cells were maintained in RSeT and passaged as above every 4–5 days. For PXGL cells, conversion was performed as previously described[48]. In brief, cells were cultured in 5% $O_2$, 7% $CO_2$, and the medium was switched to chemical resetting medium 1 (cRM-1) consisting of N2B27 medium supplemented with 1 µM PD0325901 (University of Cambridge, Stem Cell Institute), 10 ng ml⁻¹ human recombinant LIF (300-05, PeproTech), and 1 mM valproic acid. N2B27 medium contains 1:1 DMEM/F12 and Neurobasal A (10888-0222, Thermo Fisher Scientific)

supplemented with 0.5× B27 (10889-038, Thermo Fisher Scientific), 0.5× N2 (made in-house), 100 µM β-mercaptoethanol, 1× GlutaMAX, and 1× penicillin-streptomycin. cRM-1 medium was changed every 48 h for 4 days, after which the medium was changed to PXGL. PXGL consists of N2B27 medium supplemented with 1 µM PD0325901, 10 ng ml⁻¹ human recombinant LIF, 2 µM Gö6983 (2285, Tocris) and 2 µM XAV939 (X3004, Merck). PXGL cells were passaged every 4–6 days using TrypLE (12604013, Thermo Fisher Scientific) for 3 min. 10 µM ROCK inhibitor Y-27632 and 1 µl cm⁻² Geltrex (A1413201, Thermo Fisher Scientific) was added at passage for 24 h. For yolk sac-like cell or trophoblast differentiation, RSeT cells were passaged onto Matrigel-coated IBIDI chamber slides. 24 h later, the medium was switched to ACL (100 ng ml⁻¹ activin-A (Qk001, QKINE), 3 µM CHIR99021 (72052, STEMCELL Technologies) and 10 ng ml⁻¹ human LIF) for hypoblast induction or PA (1 µM PD0325901 and 1 µM A83-01 (72022, STEMCELL Technologies)) with or without 500 nM lysophosphatidic acid (LPA) (3854, Tocris).

### Generation of inducible human ES cell lines

To generate Piggyback plasmids, full-length coding sequences were amplified from human cell line cDNA with AttB overhangs using Phusion High-Fidelity DNA polymerase (M0530S, New England Bio-Labs) according to the manufacturer's instructions. Amplicons were introduced to pDONR221 entry plasmids using BP clonase (11789100, Thermo Fisher Scientific), and subsequently to destination plasmids using LR clonase (11791020, Thermo Fisher Scientific) according to the manufacturer's instructions. Human ES cells were electroporated with GATA6-3×Flag-TetOn-Zeo (entry plasmid 72922, Addgene) and/or SOX17-TetOn-Hygro or GATA3-EGFP-TetO-Hygro (a gift from M. Drukker) and/or TFAP2C-TetOn-G418 in addition to PB-CAG-rTTA3-Bsd or PB-CAG-rTTA3-Zeo and pBase plasmid expressing PiggyBac Transposase using the Neon transfection system with the following settings: 1,200 V, 20 ms, 2 pulses. Two days after transfection, antibiotics were applied at a 0.25× dosage and increased to final concentrations of 100 µg ml⁻¹ zeocin (ant-zn-1, Invitrogen), 20 µg ml⁻¹ blasticidin (A113903, Thermo Fisher Scientific), 50 µg ml⁻¹ G418 (10131035, Thermo Fisher Scientific) or 50 µg ml⁻¹ Hygromycin B (10687010, Thermo Fisher Scientific). Shef6-mKate2 human ES cells were a gift from M. Shahbazi. Clones were generated by manually picking single colonies under a dissecting microscope. Transgene activation was triggered by the addition of 1 µg ml⁻¹ doxycycline hyclate (D9891, Sigma). To select clones for downstream experiments, isolated colonies that survived manual picking were induced for 72 h and cell pellets were collected for quantitative PCR or stained for immunofluorescent analysis (note: this was performed in primed human ES cells). Transgene expression and another key lineage marker were assessed for changes in expression compared to uninduced controls. Clones with robust transgene upregulation and downstream upregulation of an uninduced lineage marker were selected for subsequent experimentation (1–2 clones per transgenic line). Note that AP2γ-inducible cells did not reset in PXGL naive conditions.

### Quantitative PCR with reverse transcription analysis

Cell pellets were collected and RNA was extracted using the Qiagen RNeasy kit following the manufacturer's instructions. The reverse transcriptase reaction was performed with 1 µg RNA with random primers (C1181, Promega), dNTPs (N0447S, New England BioLabs), RNAse inhibitor (M0314L, New England Biolabs, and M-MuLV reverse transcriptase (M0253L, New England Biolabs). Quantitative PCR with reverse transcription was performed using Power SYBR Green PCR Master Mix (4368708, Thermo Fisher Scientific) on a Step One Plus Real-Time PCR machine (Applied Biosystems). The following program was used: 10 min at 95 °C followed by 40 cycles of 15 s at 95 °C and 1 min at 60 °C. Single melt curves were observed for all primers used in this study. Oligonucleotides used in this study are provided in Supplementary Table 4.

## Generation of human pluripotent stem cell–mouse embryo chimeras

For human cell–mouse embryo chimeras, oviducts and uterine horns were recovered and flushed with M2 medium (made in-house) supplemented with 4 mg ml$^{-1}$ BSA (A9418, Sigma) on E2.5. Recovered pre-compacted 8-cell stage embryos were then subjected to zona pellucida removal by treatment with acidic Tyrode's Solution. Human cells (wild-type, 3-day induced GATA6-SOX17 cells, and 3-day induced GATA3-AP2γ RSeT cells) were prepared by dissociating cells with TrypLE and washing them as described above. Cells were resuspended in either RSeT or N2B27 medium supplemented with 5% KSR and 1 μg ml$^{-1}$ doxycycline. The resulting small clumps of cells were aggregated with the 8-cell stage mouse embryos in indentations in this medium for 24 h, before being transferred to KSOM embryo culture medium (made in-house) with or without 1 μg ml$^{-1}$ doxycycline for a further 24 h until E4.5. For negative controls, embryos were cultured in these conditions without the addition of human cells. Chimeric blastocysts were then fixed for analysis by immunofluorescence. The contribution of human nuclear antigen-positive cells to either SOX2, SOX17, and/or GATA3 populations were quantified in those embryos that successfully developed to the late blastocyst stage.

## Generation of inducible human embryoids

To generate the 3D stem cell-derived model of the post-implantation embryo, RSeT cells between 2 and 6 passages post-conversion to RSeT media were passaged as normal; the following day (day −3) the medium for extraembryonic-like cells (induced GATA6, induced GATA6-SOX17 or induced GATA3-AP2γ) was changed to N2B27 medium with 5% knockout serum replacement and 1 μg ml$^{-1}$ doxycycline. This medium was refreshed every 24 h for 3 days. On day 0 (the day of aggregation), an Aggrewell dish (34415, STEMCELL Technologies) was prepared by pre-coating with anti-adherence solution (07010, STEMCELL Technologies) and centrifuging at 2,000$g$ for 5 min. Wells were washed twice with phosphate-buffered saline (PBS) prior to the addition of experiment medium. This medium consists of N2B27 with 5% knockout serum replacement, 1 μg ml$^{-1}$ doxycycline and 10 μM Y-27632. Induced cells and wild-type ES cells were enzymatically dissociated 1 h after addition of 10 μM Y-27632 to wells containing cells for inducible human embryoid generation. Dissociated cells were pelleted and resuspended in experiment media and placed in gelatin-coated wells for mouse embryonic fibroblast depletion. After 15–30 min cells were counted, mixed and plated into an Aggrewell dish with a final calculation of 8 wild-type ES cells, 8 hypoblast-like cells and 16 trophoblast-like cells plated per microwell in the Aggrewell. At 8 dpf, in vitro cultured human embryos had 32:24:228 epiblast:hypoblast:trophoblast cells[2]. Importantly, however, many of the trophoblast cells are not in contact with the inner cell mass-derived tissues or are terminally differentiated. Additionally, we observed that in culture, inducible GATA6-SOX17 cells proliferate slower than the other two cell populations after doxycycline addition. Therefore, we used an initial seeding density with: (1) a total cell number similar to that used in mouse models that allowed for successful cell sorting[20,49]; (2) a ratio of cells that reflected the peri-implantation embryo; and (3) a reduced number of inducible GATA3-AP2γ cells and an increased number of inducible GATA6-SOX17 cells.

On day 1, the medium was subjected to two two-thirds changes of N2B27 medium with 5% knockout serum replacement and 1 μg ml$^{-1}$ doxycycline. On day 2, Aggrewells were subjected to a half change with hIVC1 medium with 25 ng ml$^{-1}$ hIGF1 (78022.1 STEMCELL Technologies) and 1 μg ml$^{-1}$ doxycycline. hIVC1 medium consists of Advanced DMEM/F12 (12634-010 Thermo Fisher Scientific) supplemented with 20% inactivated FBS (10270106, Thermo Fisher Scientific), 1× Glutamax, 1× non-essential amino acids, 1× essential amino acids, 1× ITS-X, 25 U ml$^{-1}$ penicillin-streptomycin, 1.8 mM glucose (G8644, Sigma-Aldrich), 0.22%

sodium lactate (L7900, Sigma-Aldrich), 8 nM β-oestradiol (50-28-2, Tocris) and 200 ng ml$^{-1}$ progesterone (P0130, Sigma-Aldrich). This medium was used in half changes each day from day 3. On day 4, aggregates were manually picked using a mouth pipette under a dissecting microscope into individual wells of ultra-low attachment 96-well plates (CLS7007, Corning) in hIVC1 medium with IGF1 and doxycycline as above for subsequent culture.

## Immunostaining and image analysis

Samples were washed with PBS and fixed in 4% paraformaldehyde (1710, Electron Microscopy Sciences) at room temperature for 20 min. Samples were washed 3 times with PBS + 0.1% (vol/vol) Tween-20 (PBST) and incubated with 0.3% (vol/vol) Triton X-100 (T8787, Sigma-Aldrich) + 0.1 mM glycine (BP381-1, Thermo Fisher Scientific) in PBS at room temperature for 30 min. Samples were blocked in blocking buffer (PBST with 5% (w/vol) BSA, A9418, Sigma), then incubated with primary antibodies diluted in blocking buffer overnight at 4 °C. See Supplementary Table 5 for a list of primary antibodies. Samples were washed 3 times in PBST and incubated with fluorescently conjugated AlexaFlour secondary antibodies (Thermo Fisher Scientific, 1:500) and DAPI (D3571, Thermo Fisher Scientific, 1 μg ml$^{-1}$) diluted in blocking buffer for 2 h at room temperature. For pSMAD1.5 quantification, OCT4-positive or GATA6-positive nuclei were isolated and the fluorescent intensity of pSMAD1.5 was quantified. For SMAD2.3 quantification, OCT4-positive or GATA6-positive (excluding the outermost GFP$^+$ cell layer) nuclei fluorescence intensity was quantified, as well as cytoplasmic fluorescence intensity. Data are presented as the ratio of nuclear:cytoplasmic fluorescence intensity. Immunofluorescence images were analysed using FIJI. The spots tool with manual curation in Imaris software (version 9.1.2, Oxford Instruments) was used to quantify total cell numbers in day 4 embryoids and generate spot renders.

## Human embryo thawing and culture

Human embryos were thawed and cultured as described previously[2,8]. In brief, cryopreserved human blastocysts (5 or 6 dpf) were thawed using the Kitazato thaw kit (VT8202-2, Hunter Scientific) according to the manufacturer's instructions. The day prior to thawing, thawing solution (TS) from the Kitazato kit was placed at 37 °C overnight. The next day, IVF straws were submerged in 1 ml pre-warmed TS for 1 min. Embryos were then transferred to DS for 3 min, WS1 for 5 min, and WS2 for 1 min. These steps were performed in reproplates (Hunter Scientific) using a Stripper micropipette (Origio). Embryos were incubated at 37 °C and 5% CO$_2$ in normoxia and in pre-equilibrated human IVC1 supplemented with 50 ng ml$^{-1}$ IGF1 (78078, STEMCELL Technologies) under mineral oil for 1–4 h to allow for recovery. Following thaw, blastocysts were briefly treated with acidic Tyrode's solution (T1788, Sigma) to remove the zona pellucida and placed in pre-equilibrated human IVC1 in 8-well μ-slide tissue culture plates (80826, Ibidi) in approximately 400 μl volume per embryo per well. Half-medium changes were done every 24 h.

## Statistical analysis

Statistical analyses were performed using Graphpad Prism v9.4. Sample sizes were not predetermined, and the researchers were not blinded to conditions. All experiments were performed independently at least twice. Data were tested for normality using the Shapiro-Wilk test. Normally distributed data were analysed using parametric tests (unpaired $t$-test or ANOVA) and non-normally distributed data were analysed using non-parametric tests (Mann–Whitney $U$ tests or Kruskal–Wallis tests) as indicated in figure legends. Numbers of samples are indicated in figure legends. All statistical tests were two-tailed. For each test, individual samples were used. Replicates are biological unless otherwise stated. Within plots, all data are presented as mean ± s.e.m. For box plots, the box represents the 25th–75th centiles and whiskers represent minimum and maximum, with the central line representing median and + symbol representing mean. For multiple comparisons testing, comparisons

were made only to the control condition. Unmarked pairwise comparisons are not significant ($P > 0.05$).

## Collection, generation and sequencing of single-nuclei ATAC and RNA 10X libraries

To collect post-implantation embryo-like models for single-cell sequencing, correctly organized embryoids at day 4, 6, and 8 were picked visually and washed through PBS twice in a 4-well dish before transfer to TrypLE. Samples were agitated by pipetting every 5 min for 10–20 min until dissociated. Enzymatic activity was inactivated through addition of 20% FBS in PBS at 2x volume. Cells were collected in a Falcon tube, pelleted, and resuspended in freeze buffer consisting of 50 mM Tris at pH 8.0 (15-567-027, Fisher Scientific), 25% glycerol (G5516, Sigma-Aldrich), 5 mM magnesium acetate (63052, Sigma-Aldrich), 0.1 mM EDTA (15575020, Thermo Fisher Scientific), 5 mM DTT (R0861, Thermo Fisher Scientific), 1× protease inhibitor cocktail (P8340, Sigma-Aldrich), 1:2,500 dilution of superasin (AM2694, Invitrogen). For cell lines, 10,000 cells were counted, pelleted, and resuspended in the freeze medium above before slow freezing at −80 °C.

For nuclei isolation and library construction, low input nuclei isolation protocol from 10X Genomics was performed. In brief, frozen cell pellets were thawed in a 37 °C water bath for 30 s, centrifuged (500$g$ for 5 min at 4 °C) to pellet the cells, and then the supernatant was aspirated. The cell pellets were washed with 200 μl 1× PBS with 0.04% BSA twice, centrifuged, and supernatant was aspirated between washes. Subsequently, chilled lysis buffer (45 μl per sample) was added to the washed cell pellet and placed on ice for 3 min, then wash buffer (50 μl per sample) was added. Washed isolated nuclei were resuspended in a diluted nuclei buffer. In this study, the isolated nuclei were resuspended in 5 μl diluted nuclei buffer and were directly added to the transposition reaction. In all following steps, 10X Genomics' Single Cell Multiome ATAC and gene-expression protocols were followed according to the manufacturer's specifications and guidelines. The final libraries were loaded on the NextSeq 2000 using P2 100 cycle kit at 650 pM loading concentration with paired-end sequencing following the recommended sequencing reads from 10X Genomics (28, 10, 10 and 90 cycles for gene-expression libraries and 50, 8, 24 and 49 cycles for ATAC libraries).

## Single-cell sequencing analysis

**Processing and quality control.** Raw reads were analysed using the CellRangerARC pipeline to generate ATAC and RNA fastq files for each sample, and then to align genomic and transcriptomic reads. Matrices were then read into Seurat[50] and Signac[51] using the Read10X_h5 command. For ATAC-seq data, peaks from standard chromosomes were used and peaks were additionally called using macs2 to add an additional Signac assay. Cells with >500 RNA unique molecular identifier (UMI) counts, <20% mitochondrial reads, >500 ATAC reads, trasncription start site enrichment >1 and were called as singlets using scDblFinder[52] were retained for downstream analysis. For UMAP projections, SCTransform was used for RNA counts with percent mitochondrial counts and cell cycle scores regressed. Principal component analysis and latent semantic indexing graphs were used to generate a wnn embedding that accounts for both modalities. chromVAR[53] was run to calculate motif accessibility score on the peaks assay. Data visualization was performed using Seurat's DimPlot, FeaturePlot, VlnPlot, TSSPlot, FragmentHist functions as well as SCpubr's[54] do_Alluvialplot and do_Nebulosaplot functions.

**Comparisons to published datasets.** scmap[55] was used to project cell labels from other single-cell datasets onto post-implantation embryo-like model transcriptional data. All reference data used were publicly available with published cell type annotations. Cynomolgus monkey gene names were converted to HGNC gene symbols using biomaRt. For data generated with smart-seq2 or other non-UMI-based single-cell sequencing methods, the scmapCluster method was used with a similarity threshold of 0.5. For UMI-based methods the scmapCell followed by scmapCell2Cluster method was used with $w = 2$. Multiple datasets were used to draw conclusions with scmap, and transcriptionally similar clusters (such as trophoblast or amnion) may map incorrectly if both are not present due to the limited cell assignments in certain datasets. Upon cell type assignment and processing for the cell lines sequenced, a previously reported and validated logistic regression framework was applied to project cell line data onto published single-cell data and to project published cluster annotations (such as training data) onto post-implantation embryo-like model clusters (such as test data), resulting in a quantitative measure of predicted similarities[47]. Here, only differentially expressed genes (produced using Seurat's FindAllMarkers function on course cell assignments (which collapsed amnion and mesodermal clusters) were used.

**Multivelo RNA and chromatin velocity.** We applied a recently published method for velocity calculations that accounts for both single-cell ATAC-seq and RNA-sequencing data[56]. Multivelo was run on all cells which passed the quality control and processing described above. Analysis was based on available vignettes with 1,000 highly variable genes and the 'grid' method. Gene expression and chromVAR were plotted over latent time using the switchde[57] package.

**CellPhoneDB analysis.** CellPhoneDB 2.0 was used with default settings to assess potential tissue signalling crosstalk[58]. Course cell assignments which collapsed separated amnion (AM-1, AM-2, AM-3) and mesoderm (MESO-1, MESO-2) clusters was used for simplicity. Selected significant interactions were plotted as dot plots.

**Reanalysis of human in vitro cultured embryo datasets.** Previously published data from Yan et al.[22], Blakeley et al.[23], Petropoulos et al.[24], Zhou et al.[25] and Xiang et al.[26], were realigned to the hg38 human genome using kallisto or kb-bustools[59,60]. Datasets that were not sequenced with UMI-based technologies were normalized using quminorm to quasi-umis[61]. Using SCTransform-based integration, datasets were combined to generate a single-cell RNA-seq dataset of human embryos spanning zygote to 14 dpf[2,22–26]. Cells were clustered and identities assigned based on previous annotations and canonical marker expression. The dataset showed good overlap of datasets with separation of cell types and some temporal resolution. SCENIC was used with default settings in R, and the AUC–regulon table used to generate a new assay in the Seurat object. Using this assay, the epiblast, hypoblast and trophoblast lineages were then compared using Seurat's FindMarkers function to implement a Wilcoxon ranked test with Bonferroni correction to identify pairwise predicted differentially active regulons. Regulons that were enriched across both relevant comparisons (such as hypoblast versus epiblast or hypoblast versus trophoblast) were used as enriched active transcription factors for subsequent analyses (for example, in the hypoblast). These factors were then plotted in relation to each other in Cytoscape.

## Data sources

For aligning sequencing data, GRCh38 (https://www.ncbi.nlm.nih.gov/assembly/GCF_000001405.26/) and GRCm38 (https://www.ncbi.nlm.nih.gov/assembly/GCF_000001635.20/) were used. Previously published data are publicly available. Human data were from Molè et al.[2] (ArrayExpress accession E-MTAB-8060), Xiang et al.[26] (Gene Expression Omnibus (GEO) accession GSE136447), Zhou et al.[25] (GEO accession GSE109555). Petropoulos et al.[24] (ArrayExpress accession E-MTAB-3929), Blakeley et al.[23] (GEO accession GSE66507) and Yan et al.[22] (GEO accession GSE36552). Cynomolgus monkey data were from Yang et al.[39] (GEO accession GSE148683), Ma et al.[38] (GEO accession GSE130114) and Nakamura et al.[40] (GEO accession GSE74767). Human ES cell-derived datasets were from Pham et al.[34] (GEO accession

GSE191286), Kagawa et al.[11] (GEO accession GSE177689) and Zheng et al.[15] (GEO accession GSE134571).

## Reporting summary

Further information on research design is available in the Nature Portfolio Reporting Summary linked to this article.

## Data availability

10X multiome data for cell lines and inducible human embryoids are available from GEO accession GSE218314. Source data are provided with this paper.

## Code availability

No custom code was used in this manuscript. Code used to analyse the data is available at https://github.com/bweatherbee/human_model.

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

**Acknowledgements** The authors thank CARE Fertility, Herts and Essex Fertility Centre, Bourn Hall Fertility Clinic and King's Fertility Clinic for their collaboration in the donation of supernumerary human embryos; all members of the M.Z.-G. and T. E. Boroviak laboratories, G. Amadei and D. Glover for feedback; and M. Shahbazi for the gift of the Shef6-mKate2 cell line and for feedback on the manuscript. The authors would also like to thank Robin Skory and Nicolas Plachta for advice on image analysis. This work is supported by Wellcome Trust (207415/Z/17/Z), in part by European Research Council (669198), Open Atlas, and NOMIS award grants to M.Z.-G., Allen Discovery Center for Cell Lineage Tracing grants to J.S. and M.Z.-G., in addition to individual funding from the Gates Cambridge Trust (to B.A.T.W.) and Leverhulme Trust Early Career Fellowship (to C.W.G.). J.S. is an investigator of the Howard Hughes Medical Institute and M.Z.-G. is NOMIS Distinguished Scientist and Scholar.

**Author contributions** B.A.T.W. and C.W.G. designed, carried out and analysed experiments. B.A.T.W. and C.W.G. independently reproduced human embryoid generation and results. B.A.T.W., C.W.G. and L.K.I.-S. performed human embryo work. L.K.I.-S. performed mouse embryo–human pluripotent stem cell chimera experiments. N.H. and R.M.D. prepared 10X multiome libraries and performed RNA sequencing and ATAC-seq, supervised by J.S. B.A.T.W. analysed single-cell sequencing data. C.W.G. and B.A.T.W. wrote the manuscript with input from all co-authors. M.Z.-G. conceived and supervised the project.

**Competing interests** The authors are inventors on the following patents: (1) Patent applicant: Caltech; inventors: M.Z.-G., B.S. and V.J.; application number: 17/692,790; specific aspect of the manuscript covered in patent application: reconstructing human early embryogenesis in vitro with pluripotent stem cells. (2) Patent applicant: Caltech and Cambridge Enterprise Limited; inventors: M.Z.-G., G.A. and C.H.; application number: 63/397,630; specific aspect of the manuscript covered in patent application: synthetic embryos. (3) Patent applicant: Caltech and Cambridge Enterprise Limited; inventors: M.Z.-G., B.W. and C.G.; application number: 63/403,684; specific aspect of the manuscript covered in patent application: stem cell-derived model of the human embryo.

**Additional information**
**Correspondence and requests for materials** should be addressed to Magdalena Zernicka-Goetz.

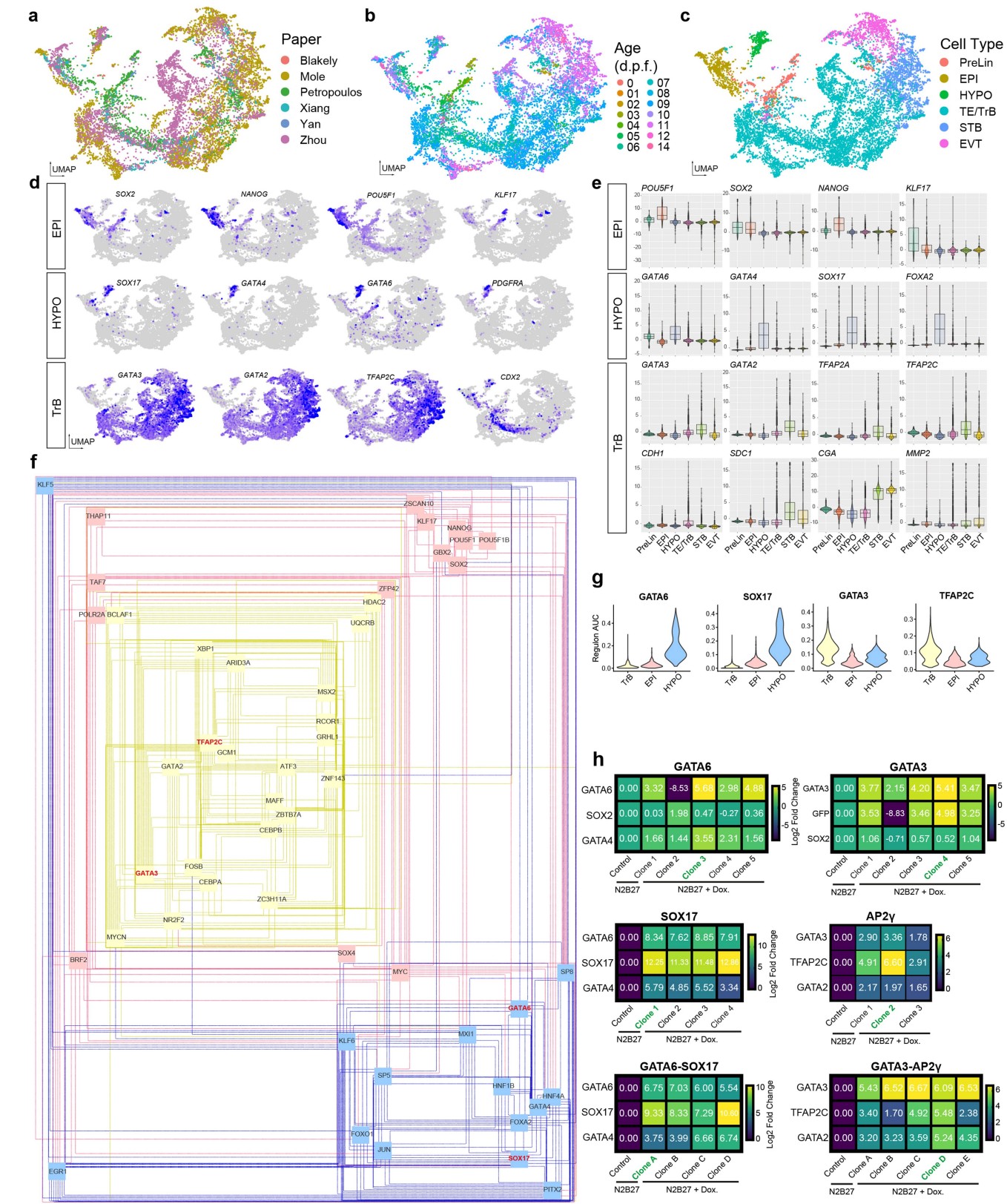

**Extended Data Fig. 1** | See next page for caption.

**Extended Data Fig. 1 | Selection of transgenes to drive extraembryonic-like cells. a**, Uniform Manifold Approximation Projection (UMAP) showing combined human pre- to post-implantation datasets colored by original publication. **b**, UMAP of combined human datasets colored according to stage of embryo (days post-fertilization; d.p.f.). **c**, UMAP of combined human datasets colored according to cell type. **d**, Cardinal cell type gene expression on UMAP of human datasets. **e**, Plots from single cell RNA sequencing of key marker gene expression in human datasets separated by cell type. n = 10223 cells. **f**, Inferred epiblast, hypoblast, and trophoblast gene regulatory network generated by SCENIC during peri-implantation human embryo development. See methods for details on datasets and processing. Candidate factors are in bold red. **g**, Regulon activity scored by SCENIC for hypoblast markers GATA6, SOX17, and TrB markers GATA3 and TFAP2C. n = 10223 cells. **h**, qRT-PCR analysis of individual inducible cell lines. Doxycycline inducible constructs were inserted in Shef6 hESC using piggybac transposase (inducible GATA6, SOX17, GATA6-SOX17, GATA3, AP2γ and GATA3-AP2γ). Colonies were manually isolated after single cell plating and propagated. Appropriate transgene expression was validated by RT-qPCR after 72 h of 1μg/mL doxycycline addition in basal N2B27 conditions. N = 3 technical replicates. Clones selected for further analysis are in green. For box plots, box encompasses the 25th–75th quartile with whiskers to minimum and maximum. Central line marks median.

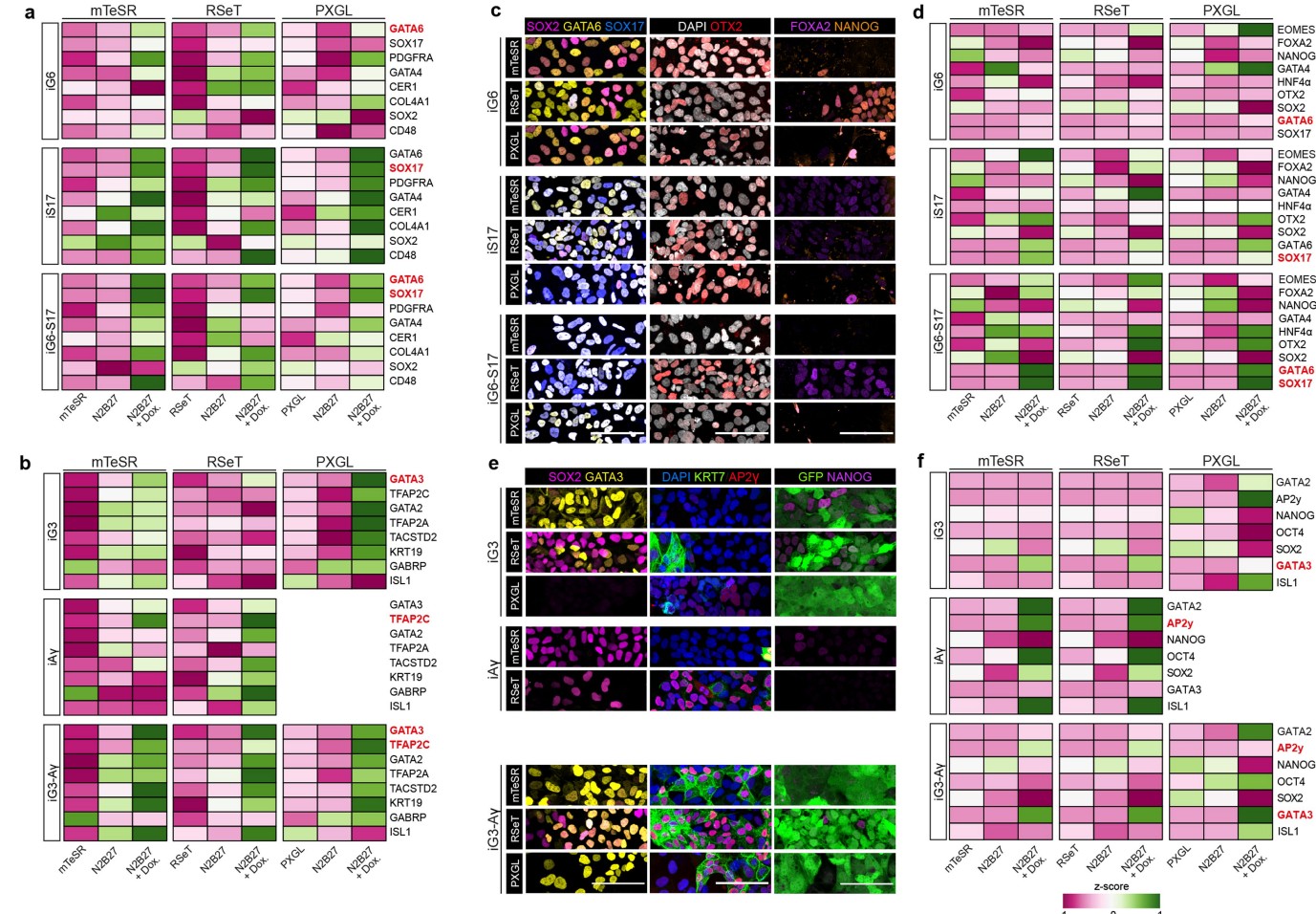

**Extended Data Fig. 2 | Immunofluorescence analysis of cardinal marker genes of hypoblast and trophoblast after doxycycline induction across pluripotent states. a**, qRT-PCR analysis after 3 days of doxycycline-induction of induced GATA6 (iG6), induced SOX17 (iS17), or induced GATA6-SOX17 (iG6-S17) singly and together from three pluripotent states. **b**, qRT-PCR analysis after 3 days of DOX-induction of induced GATA3 (iG3), induced AP2γ (iAγ), or induced GATA3-AP2γ (iG3-Aγ) from multiple pluripotent starting states. For a-b, N = 3 technical replicates from 3 independent experiments. **c**, Immunofluorescence analysis of iG6, iS17, or iG6-S17 cells after 3 days induction from multiple pluripotent states. **d**, Quantification of immunofluorescence levels of c. **e**, Immunofluorescence analysis of iG3, iAγ, or iG3-Aγ after 3 days induction from multiple pluripotent states. **f**, Quantification of immunofluorescence levels of e. For c-f, N = 3 technical replicates from 2 independent experiments. Cells are initially cultured in either mTeSR, RSeT, or PXGL conditions, and then cultured for 3 days either under the same conditions or alternatively transferred to either basal N2B27 media or basal N2B27 media with the addition of doxycycline. The induced transgenes are marked in red. Scale bars = 100μm.

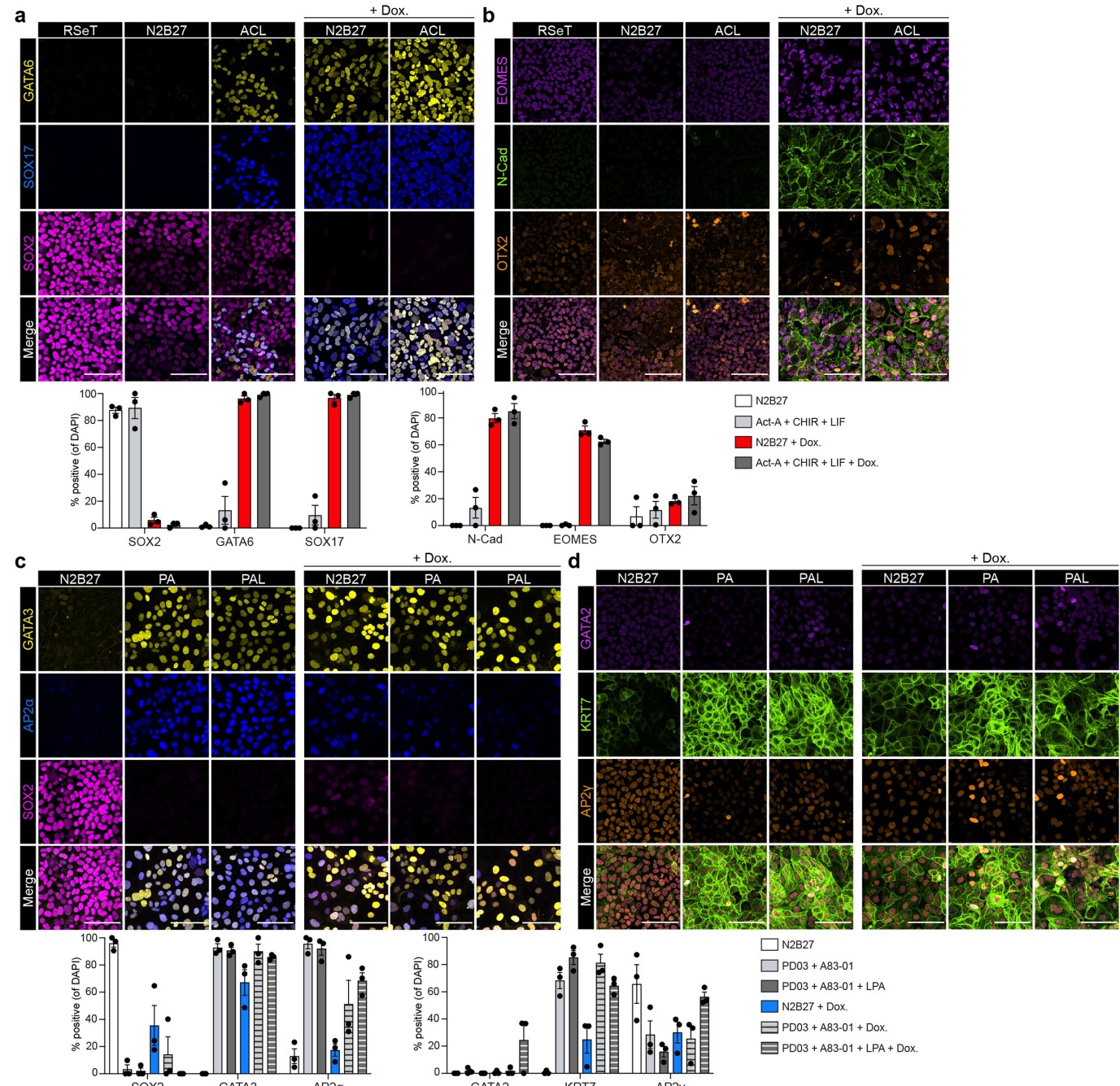

**Extended Data Fig. 3 | Comparison of transcription factor-mediated induction with published directed differentiation methods. a**, Comparison and quantification of GATA6, SOX17 and SOX2 after yolk sac-like cell (Activin-A, CHIR99021 and LIF) directed differentiation, doxycycline-mediated induction in inducible GATA6-SOX17 cells, or both. Cells were differentiated from RSeT conditions. **b**, Comparison and quantification of EOMES, N-Cadherin, and OTX2 after yolk sac-like cell (Activin-A, CHIR99021, and LIF) directed differentiation, doxycycline-mediated induction in inducible GATA6-SOX17 cells, or both. For a-b, N2B27: n = 717; ACL: n = 1211; N2B27+Dox.: n = 522; and ACL+Dox.: n = 544 cells from 3 fields of view from 2 independent experiments.

**c**, Comparison and quantification of GATA3, AP2α, and SOX2 after PA (PD0325901 and A83-01) or PAL (PD0325901, A83-01, and LPA) directed differentiation, doxycycline-mediated induction in inducible GATA3-AP2γ cells, or both. **d**, Comparison and quantification of GATA2, KRT7, and AP2γ after PA (PD0325901 and A83-01) or PAL (PD0325901, A83-01, and LPA) directed differentiation, doxycycline-mediated induction in inducible GATA3-AP2γ RseT cells, or both. For c-d, N2B27: n = 443; PA: n = 487; PAL: n = 371; N2B27+Dox.: n = 357; PA+Dox.: n = 412; and PAL+Dox.: n = 287 cells from 3 fields of view from 2 independent experiments. Scale bars = 100 μm. For a-d, mean ± SEM is plotted. Differentiation was carried out on hESC in RSeT conditions.

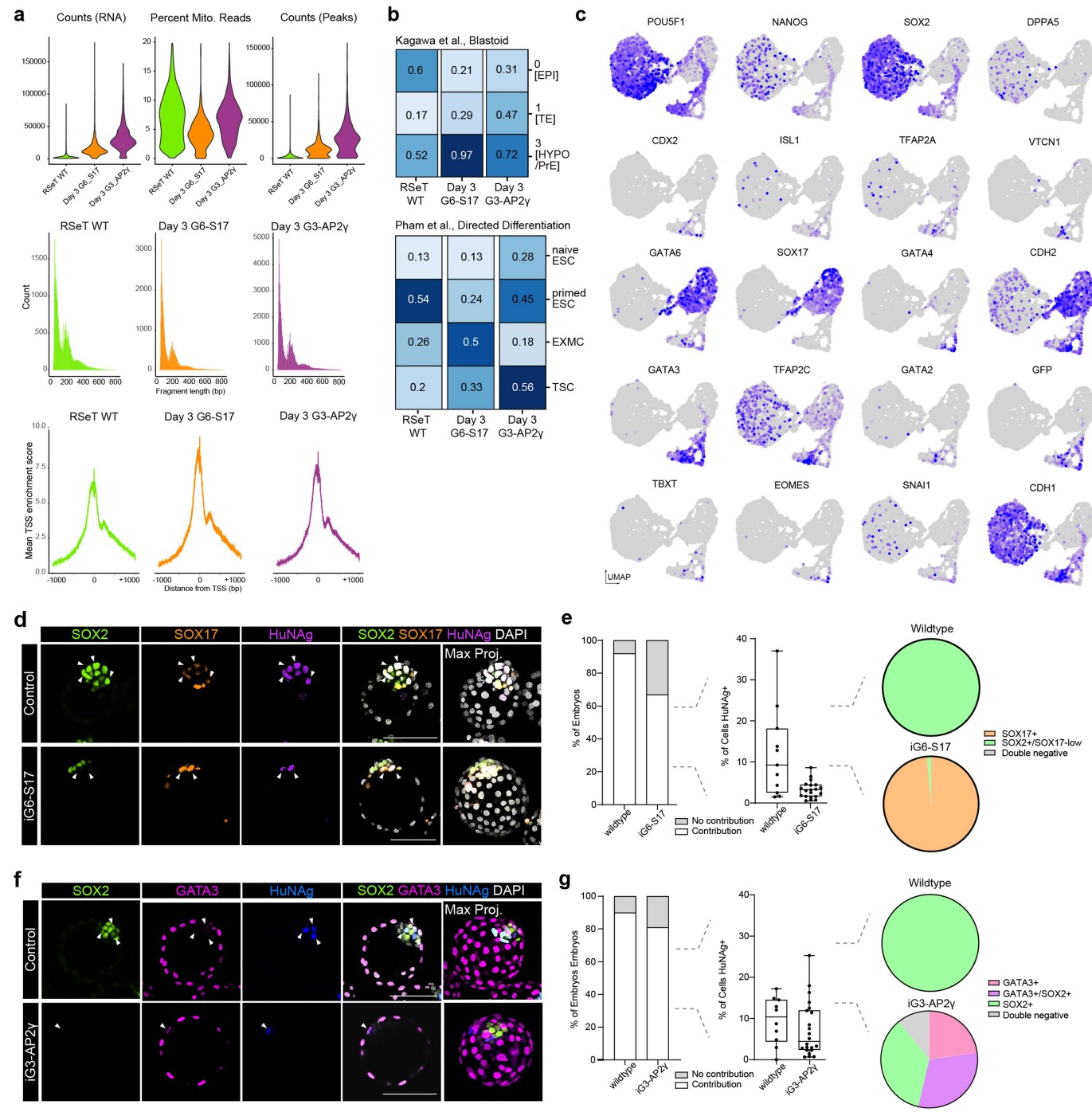

**Extended Data Fig. 4 | Assessing extraembryonic-like induction from RseT cells. a**, Quality control plots of cell line 10x multiome sequencing data. n = 5328 cells. Violin plots go from minimum to maximum. **b**, Logistic regression framework to assess similarity between clusters was applied to cell line RNA-sequencing data using published *in vitro* blastoid and directed differentiation protocols as training data. **c**, Gene expression of selected genes after 3 days of doxycycline-induction from sequencing of wildtype, inducible GATA6-SOX17 (iG6-S17), and inducible GATA3-AP2γ (iG3-AP2γ) RSeT hESC populations visualized on a uniform manifold projection and approximation (UMAP). Visualization of sample distribution in the UMAP is shown in Fig. 1c. **d**, Immunofluorescence images of human cell-mouse embryo chimeras at the late blastocyst stage show shift of human cells marked by human nuclear antigen (HuNAg) contributing to

the SOX2-positive epiblast to the SOX17-positive primitive endoderm upon iG6-S17 induction. **e**, Quantification of the contribution of HuNAg-positive cells stained for SOX2 and SOX17. Control: n = 12 and iG6-S17: n = 30 embryos from 3 independent experiments. **f**, Immunofluorescence images of human cell-mouse embryo chimeras at the late blastocyst stage show a shift of human cells from the SOX2-positive epiblast to the GATA3-positive trophectoderm upon iG3-AP2γ induction. **g**, Quantification of the contribution of HuNAg-positive cells stained for SOX2 and GATA3. control: n = 10 and iG6-AP2γ: n = 27 embryos from 3 independent experiments. Induction was carried out from hESC in RSeT conditions. For box plots, box encompasses the 25th–75th quartile with whiskers to minimum and maximum. Central line marks median. Scale bars = 100 μm.

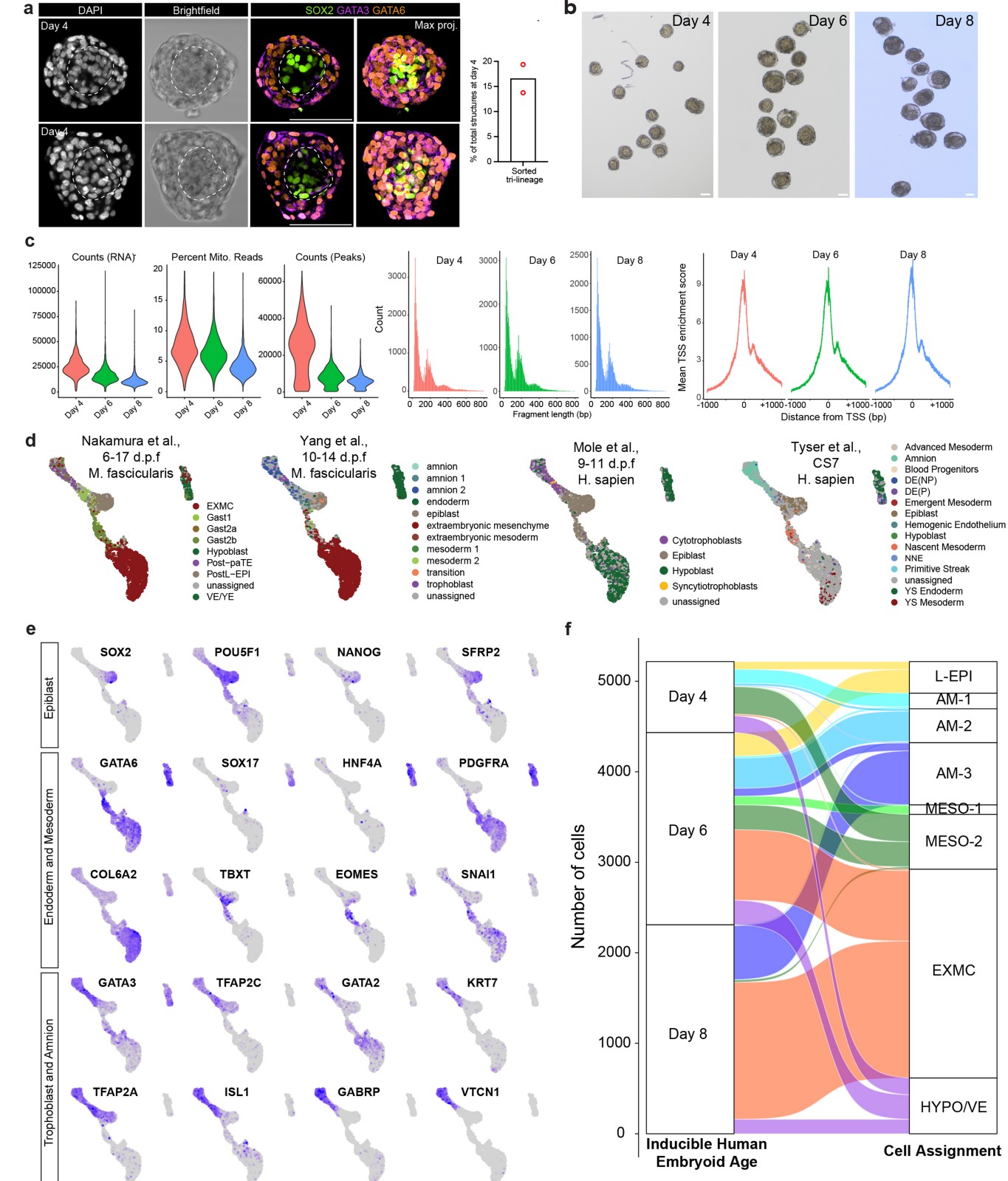

**Extended Data Fig. 5 | Post-implantation human embryo-like model cluster identification. a**, Day 4 embryoids generated from a second hESC line, RUES2. N = 371 structures from 2 independent experiments. **b**, Brightfield image of inducible human embryoids selected at days 4, 6, and 8 for sequencing (n = 12 at each stage). Note the presence of an inner domain surrounded by two concentric domains. **c**, Quality control plots for embryo sequencing data at days 4, 6, and 8 post-aggregation. n = 5217 cells. Violin plots go from minimum to maximum.

**b**, scmap projection of inducible human embryoid cells onto cynomolgus macaque (*M. fasicularis*) and human datasets (*H. sapien*) spanning peri-implantation to gastrulation stages. **e**, Cardinal marker gene expression for epiblast, endoderm and mesoderm, and trophoblast and amnion within the stem cell-derived model. **f**, Alluvial plot showing the contribution of day 4, 6, or 8 embryoids to assigned cell type. Scale bars = 100 μm. Inner domains of embryoids are surrounded by a dashed line.

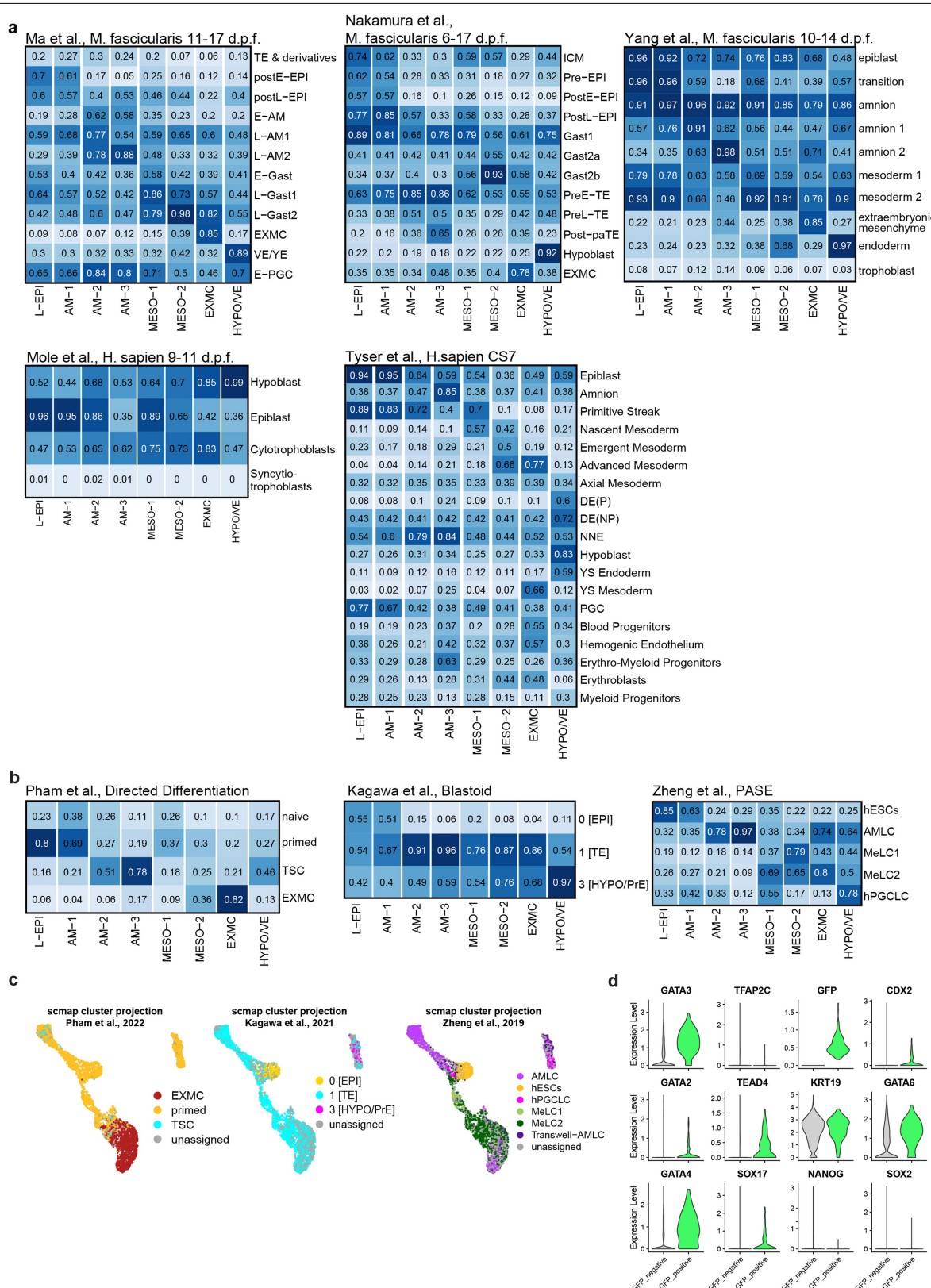

**Extended Data Fig. 6 | Embryoid cluster comparison to human and cynomolgus monkey datasets. a**, Logistic regression analysis comparing annotated clusters from cynomolgus macaque (*M. fasicularis*) and human datasets (*H. sapien*) spanning peri-implantation to gastrulation stages (training data) to post-implantation human embryo-like model clusters (test data). Cynomolgus data from Ma et al.[38], Nakamura et al.[40], Yang et al.[39]; Human data from Molè et al.[2] and Tyser et al., 2021. **b**, Logistic regression analysis comparing *in vitro* human embryo-like model and directed differentiation datasets (training data) to inducible human embryoids (test data). **c**, scmap projection of human inducible embryoid dataset onto *in vitro* datasets. *In vitro* datasets from Pham et al.[34], Kagawa et al., 2021, and Zheng et al.[15]. **d**, Violin plots of gene expression in GFP-negative versus GFP-positive cells derived from induced GATA3-AP2γ (iG3-AP2γ) cells from inducible human embryoid sequencing dataset. n = 5217 cells. Violin plots go from minimum to maximum.

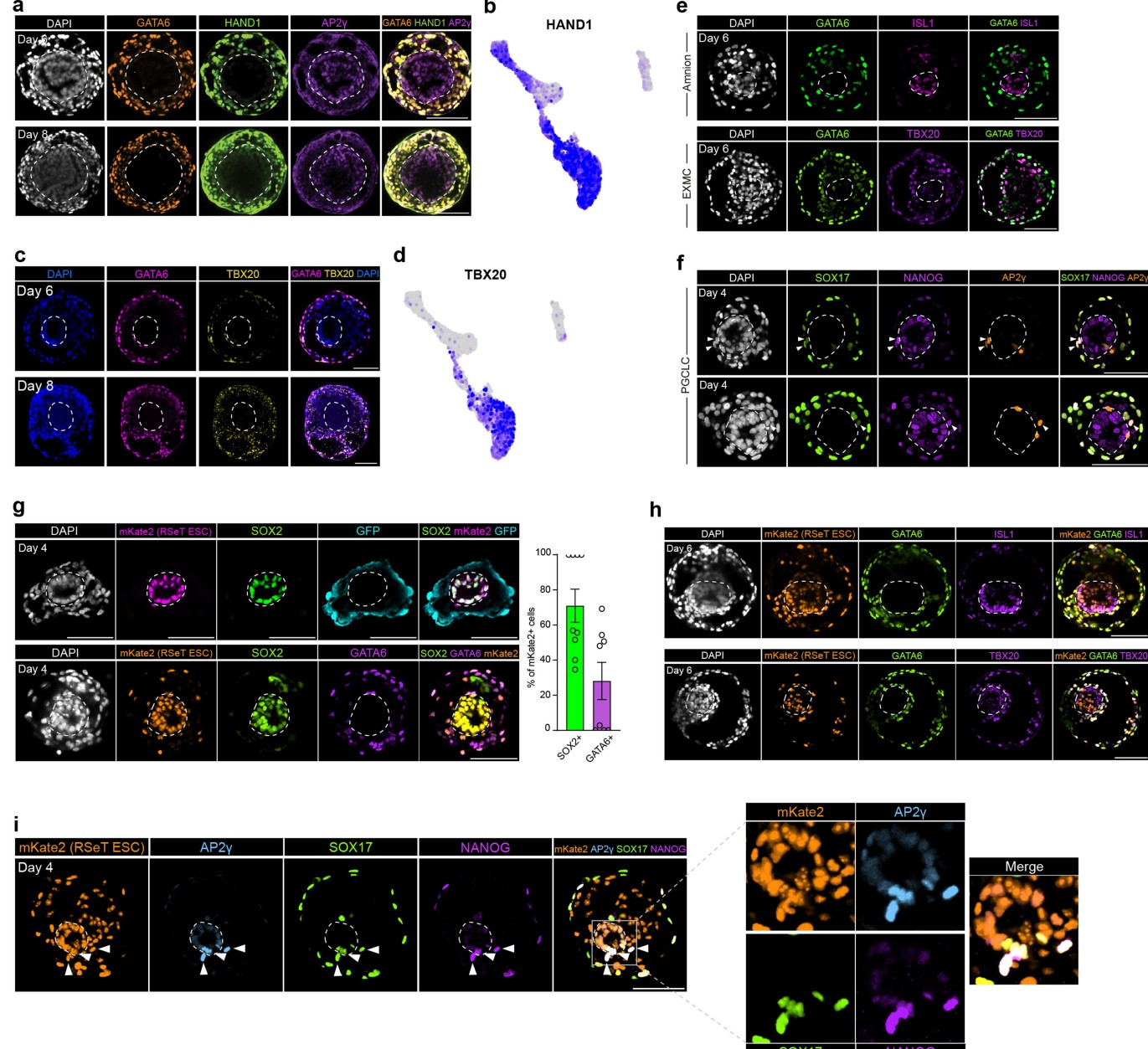

**Extended Data Fig. 7 | Extraembryonic mesenchyme trajectory and wildtype cell differentiation capacity. a**, Immunofluorescence of HAND1 demonstrating expression in GATA6-positive cells (putative extraembryonic mesenchyme) and upregulation between days 4 and 6 in putative amnion (AP2γ-positive). Representative of 2 experiments. **b**, Expression of *HAND1* in the inducible human embryoid single cell sequencing dataset. **c**, Immunofluorescence of TBX20 demonstrating high expression in a subset of GATA6-positive cells (putative extraembryonic mesenchyme). Representative of 5 experiments. **d**, Expression of *TBX20* in the inducible human embryoid single cell sequencing dataset demonstrating enrichment in the extraembryonic mesenchyme cluster. **e**, Differentiation of ISL1-positive amnion and GATA6/TBX20-positive extraembryonic mesenchyme in structures derived from a second cell background, RUES2. Representative of 2 experiments. **f**, Differentiation of primordial germ cell-like cells in embryoids derived from a second cell

background, RUES2. Representative of 2 experiments. **g**, Examples and quantification of day 4 embryoids. Embryoids exhibit an outer layer of GFP-positive induced GATA3-AP2γ (iG3-AP2γ) cells, an inner domain comprised of mKate2-positive wildtype hESCs, and an interstitial GATA6-positive population largely comprised of unlabeled induced GATA6-SOX17 (iG6-S17) cells. n = 9 embryoids from 2 independent experiments. **h**, ISL1-positive amnion-like cells overlap with mKate2-positive wildtype cells. Representative of 3 experiments. **i**, Expression of GATA6 and TBX20-positive extraembryonic mesenchyme-like cells overlap with mKate2-positive wildtype cells. Representative of 3 experiments. **j**, Expression of AP2γ, SOX17, and NANOG triple-positive primordial germ cell-like cells overlaps with mKate2-positive wildtype cells. Representative of 3 experiments. Scale bars = 100 μm. For g, mean ± SEM is plotted. Inner domains of embryoids are surrounded by a dashed line.

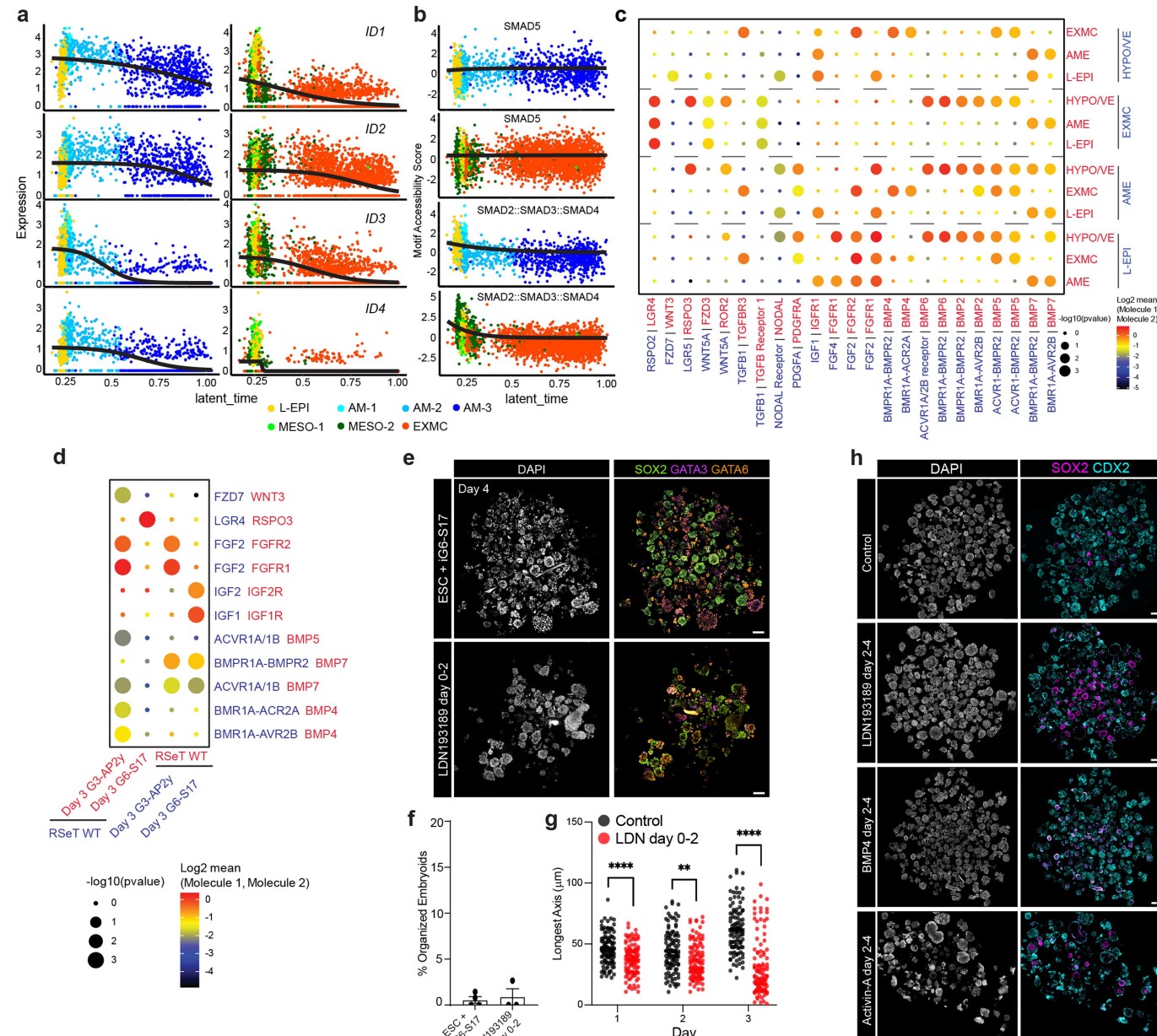

**Extended Data Fig. 8 | Importance of BMP and inducible GATA3-AP2γ cells in generating inducible human embryoids. a**, Expression of *ID1-4* fit over a latent time, colored by cell type assignment. **b**, Motif accessibility scored by chromVAR for SMAD5 and SMAD2::SMAD3::SMAD4 fit over latent time, colored by cell type assignment. **c**, Predicted ligand-receptor pairings in inducible human embryoids generated by CellPhoneDB (Efremova et al.[58]). **d**, Predicted interactions of inducible GATA6-SOX17 (G6-S17) and inducible GATA3-AP2γ (G3-AP2γ) cells after 3 days induction with wildtype RSeT hESCs, which are the cell types aggregated to generate inducible human embryoids. **e**, Inducible human embryoids do not form if induced GATA3-AP2γ cells are excluded or if the BMP signaling antagonist LDN193189 (LDN) is added between days 0–2. **f**, Quantification of embryoid formation efficiency from e. n = 535

ESC+iG6-S17 and 500 LDN-treated structures from 4 independent experiments. **g**, Quantification of embryoid size after LDN193189 addition between days 0–2. n = 105 structures per condition for each day from 3 independent experiments. Statistics: Two-sided Mann Whitney between Control and LDN at each timepoint. Day 1 *P* < 0.0001. Day 2 *P* = 0.0019. Day 3 *P* < 0.0001. **h**, Overview of whole Aggrewells demonstrating the effect of BMP inhibition, BMP4 addition, and NODAL activation. Representative of 5 experiments. Note the significant increase in well-organized structures expressing SOX2 after BMP inhibition. Scale bars = 100 μm. **\*\****P* < 0.01. **\*\*\*\****P* < 0.0001. For f, mean ± SEM is plotted. For g, all individual datapoints are plotted.

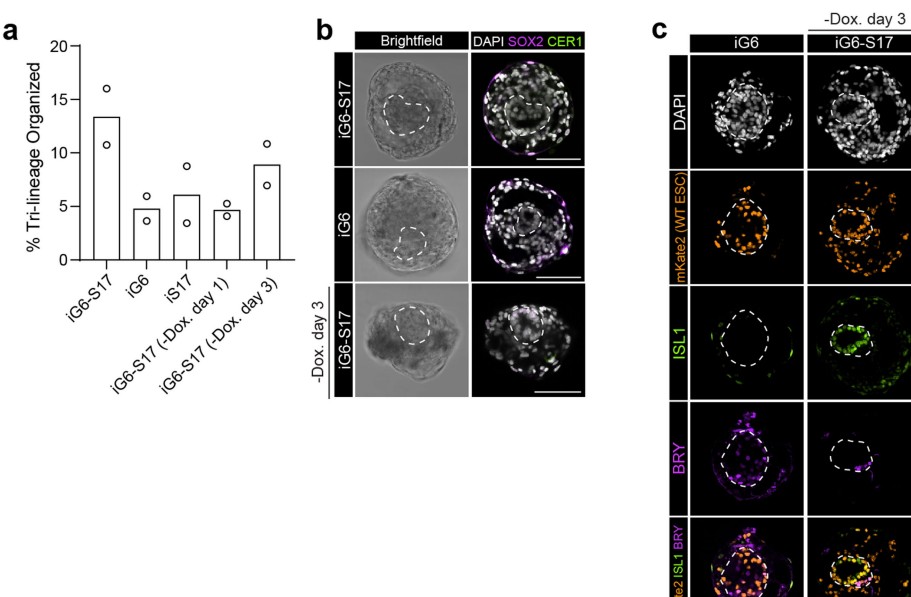

**Extended Data Fig. 9 | CER1 expression is downregulated upon extended culture of inducible human embryoids. a**, Formation efficiency of embryoids generated with different conditions. Note the highest efficiency is our standard condition with consistent addition of doxycycline and using GATA6-SOX17 inducible cells (iG6-S17). iG6-S17: n = 224; induced GATA6 (iG6): n = 276; induced SOX17 (iS17): n = 247; iG6-S17 with doxycycline removed at day 1: n = 370; and iG6-S17 with doxycycline removed at day 3: n = 410 structures from 2 independent experiments. **b**, Immunofluorescence of CER1 and SOX2 at day 6 post-aggregation demonstrates downregulation of both SOX2 and CER1 at this stage in structures generated with both inducible iG6 or iG6-S17

hypoblast-like cells (with consistent addition or doxycycline, or early removal at day 3 post-aggregation) together with wildtype ESCs and inducible GATA3-AP2γ (iG3-AP2γ) cells. Representative of 3 experiments. **c**, Embryoids generated with Shef6-mKate2 ESCs demonstrate that both the ISL1-positive and BRY-positive cell populations differentiate from the wildtype cells in structures generated with either iG6 or iG6-S17 hypoblast-like cells with early removal of doxycycline at day 3 post-aggregation. Representative of 3 experiments. Scale bars = 100 µm. Inner domains of embryoids are surrounded by a dashed line.

# Reporting Summary

## Statistics

For all statistical analyses, confirm that the following items are present in the figure legend, table legend, main text, or Methods section.

| n/a | Confirmed | |
|---|---|---|
| ☐ | ☒ | The exact sample size (*n*) for each experimental group/condition, given as a discrete number and unit of measurement |
| ☐ | ☒ | A statement on whether measurements were taken from distinct samples or whether the same sample was measured repeatedly |
| ☐ | ☒ | The statistical test(s) used AND whether they are one- or two-sided *Only common tests should be described solely by name; describe more complex techniques in the Methods section.* |
| ☒ | ☐ | A description of all covariates tested |
| ☐ | ☒ | A description of any assumptions or corrections, such as tests of normality and adjustment for multiple comparisons |
| ☐ | ☒ | A full description of the statistical parameters including central tendency (e.g. means) or other basic estimates (e.g. regression coefficient) AND variation (e.g. standard deviation) or associated estimates of uncertainty (e.g. confidence intervals) |
| ☐ | ☒ | For null hypothesis testing, the test statistic (e.g. *F*, *t*, *r*) with confidence intervals, effect sizes, degrees of freedom and *P* value noted *Give P values as exact values whenever suitable.* |
| ☒ | ☐ | For Bayesian analysis, information on the choice of priors and Markov chain Monte Carlo settings |
| ☒ | ☐ | For hierarchical and complex designs, identification of the appropriate level for tests and full reporting of outcomes |
| ☒ | ☐ | Estimates of effect sizes (e.g. Cohen's *d*, Pearson's *r*), indicating how they were calculated |

*Our web collection on statistics for biologists contains articles on many of the points above.*

## Software and code

Policy information about availability of computer code

| Data collection | No software was used for Data Collection. |
|---|---|
| Data analysis | No custom code was used in this manuscript. Scripts to used to analyze these data is available at: https://github.com/bweatherbee/human_model List of software used - command line tools: CellRangerARC (2.0.0) R - R (4.1.2); Seurat (4.2.0), Signac (1.8.0), EnsDb.Hsapiens.v86 (2.99.0), dplyr (1.0.10), ggplot2 (3.3.6), Scillus (0.5.0), RColorBrewer(1.1-3), magrittr (2.0.3), cowplot(1.1.1), viridis (0.6.2), data.table (1.14.2), scmap (1.16.0), biomaRt (2.50.3), SingleCellExperiment (1.16.0), tidyverse (1.3.2), scDblFinder (1.11.4), SCpubr (1.1.2), switchde (1.3.2), scillus (0.5.0) Python - CellPhoneDB (2.1.7), python (3.8), macs2 (2.2.7), scvelo (0.2.3), matplotlib (3.3.0), anndata (0.7.5), scanpy (1.5.1), numpy(1.17.5), multivelo (0.1.2) GraphPad Prism (9.4), FIJI (2.1.0), Imaris (9.1.2) |

For manuscripts utilizing custom algorithms or software that are central to the research but not yet described in published literature, software must be made available to editors and reviewers. We strongly encourage code deposition in a community repository (e.g. GitHub). See the Nature Portfolio guidelines for submitting code & software for further information.

## Data

Policy information about availability of data

All manuscripts must include a data availability statement. This statement should provide the following information, where applicable:
- Accession codes, unique identifiers, or web links for publicly available datasets
- A description of any restrictions on data availability
- For clinical datasets or third party data, please ensure that the statement adheres to our policy

For aligning sequencing data, GRCh38 (https://www.ncbi.nlm.nih.gov/assembly/GCF_000001405.26/) and GRCm38 (https://www.ncbi.nlm.nih.gov/assembly/GCF_000001635.20/) were used.

Previously published data is publicly available:
Human data
Molè et al., 2021: ArrayExpress E-MTAB-8060
Xiang et al., 2020: Gene Expression Omnibus GSE136447
Zhou et al., 2019: Gene Expression Omnibus GSE109555
Petropoulos et al., 2016: ArrayExpress E-MTAB-3929
Blakely et al., 2015: Gene Expression Omnibus GSE66507
Yan et al., 2013: Gene Expression Omnibus GSE36552

Cynomolgus Monkey
Yang et al., 2021: Gene Expression Omnibus GSE148683
Ma et al., 2019: Gene Expression Omnibus GSE130114
Nakamura et al., 2016: Gene Expression Omnibus GSE74767

hESC derived datasets
Pham et al., 2022: Gene Expression Omnibus GSE191286
Kagawa et al., 2022: Gene Expression Omnibus GSE177689
Zheng et al., 2019: Gene Expression Omnibus GSE134571

10x multiome data for cell lines and inducible human embryoids: Gene Expression Omnibus GSE218314

Source Data are provided with this manuscript

## Human research participants

Policy information about studies involving human research participants and Sex and Gender in Research.

| | |
|---|---|
| Reporting on sex and gender | We do not have access to prenatal genetic testing for the vast majority of embryos. Therefore, the composition of sex chromosomes of embryos cultured in the lab is largely unknown. |
| Population characteristics | According to the United Kingdom's Human Fertilisation and Embryology Act, which governs human embryo research, identifiable information of parents donating embryos to research is redacted. Therefore population characteristics of donating patients and their embryos is unknown. |
| Recruitment | Human embryos are donated by patients in the UK from collaborating IVF clinics under HFEA licence R0193. Patients undergoing IVF at CARE Fertility, Bourn Hall Fertility Clinic, Herts & Essex Fertility Clinic, and King's Fertility was given the option of continued storage, disposal, or donation of embryos to research (including project specific information) or training at the end of their treatment. Patients were offered counseling, received no financial benefit, and could withdraw their participation at any time until the embryo had been used for research.<br><br>All information of patients is required to be redacted prior to donation to research. Therefore, potential biases based on recruitment is unknown. Please note this manuscript does not perform any experimentation on human embryos, rather we seek to provide a single example of an embryo cultured in vitro as a reference image for the natural post-implantation embryo. |
| Ethics oversight | Ethical oversight is provided both by the HFEA and the Human Biological Research Ethics Committee at the University of Cambridge. The recruitment of patients to donate human embryos to research follows the Human Fertilisation and Embryology Authority's guidelines. This includes the provision of project-specific information, the offering of counseling, and the ability to withdraw consent at any time until the embryos have been used. Stem cell work is approved by the UK Stem Cell Bank. |

Note that full information on the approval of the study protocol must also be provided in the manuscript.

# Field-specific reporting

Please select the one below that is the best fit for your research. If you are not sure, read the appropriate sections before making your selection.

☒ Life sciences  ☐ Behavioural & social sciences  ☐ Ecological, evolutionary & environmental sciences

For a reference copy of the document with all sections, see nature.com/documents/nr-reporting-summary-flat.pdf

# Life sciences study design

All studies must disclose on these points even when the disclosure is negative.

| | |
|---|---|
| Sample size | No tests were used to predetermine sample size. Sample sizes for experimentation was determined based on our previous experience with human embryos (Shahbazi et al., 2016, Mole et al., 2021), human stem cells and 3D stem cell models (Shahbazi et al., 2017, Mackinlay et al., 2021), and mouse embryoid models (Harrison et al., 2017, Sozen et al., 2018, Amadei et al., 2021, Amadei et al., 2022, Lau et al., 2022).<br><br>Embryoid generation depends on the aggregation of stem cells in Aggrewell dishes which contain 1200 individual microwells. Individual structures are then recovered for analysis. Given intra-experiment variability, it is difficult to predict the number of individual samples each experiment will yield. Therefore, rather than determining predetermined sample sizes, we ensured all experiments were reproduced between 2 researchers across multiple independent experiments and cell lines. |
| Data exclusions | Exclusion criteria for single cell sequencing data was as follows to ensure high quality barcodes were used: Cells with >500 RNA UMI counts, <20% mitochondrial reads, >500 ATAC reads, TSS enrichment >1 and were called as singlets using scDblFinder were retained for downstream analysis.<br>For human embryoid efficiency quantifications and quantifications of cell line qPCR or immunofluorescence data, no data was excluded.<br>For quantification of 'inner domain' expression patterns or anterior hypoblast expression patterns of human embryoids (Figures 4 and 5), as well as selection of structures to subject to single cell sequencing, only aggregates with an organized epithelial inner domain, an intermediate tissue, and an outer layer of GFP-positive cells were included as these are the criteria that we define the human model by. |
| Replication | All experiments were repeated independently across multiple freeze-thaw cycles, and multiple conversions to the naive pluripotency state. Generation of inducible human embryoids was also repeated in 2 hESC backgrounds (Shef6 and RUES2). All experiments were performed independently at least twice. Embryoid generation was performed over 100 independent experiments throughout the duration of this project. Embryoid experimentation was performed by 2 authors and results reproduced consistently. |
| Randomization | Randomization is not relevant to this study. In experiments were embryoids were allocated to different groups (i.e. the addition of small molecule inhibitors), the media containing small molecules were added directly into Aggrewells where embryoids were generated. Given that embryoids were generated and treated within Aggrewell dishes, which contain 1200 microwells where individual embryoids may develop, and treatment between days 0-2 or days 2-4 was performed within these dishes, it would not have been possible to randomly allocate individual structures as they were developing within the dish. |
| Blinding | Investigators were not blinded to experimental groups. Given the necessity to keep clones and transgenic hESC lines pure and seperated, as well as the necessary step of calculating initial plating density, it would not be possible to blind the cell populations used to generate inducible human embryoids. Additionally, it would not be feasible to blind the media changes with the addition of small molecules as we made media in-house. |

# Reporting for specific materials, systems and methods

We require information from authors about some types of materials, experimental systems and methods used in many studies. Here, indicate whether each material, system or method listed is relevant to your study. If you are not sure if a list item applies to your research, read the appropriate section before selecting a response.

### Materials & experimental systems

| n/a | Involved in the study |
|---|---|
| ☐ | ☒ Antibodies |
| ☐ | ☒ Eukaryotic cell lines |
| ☒ | ☐ Palaeontology and archaeology |
| ☐ | ☒ Animals and other organisms |
| ☒ | ☐ Clinical data |
| ☒ | ☐ Dual use research of concern |

### Methods

| n/a | Involved in the study |
|---|---|
| ☒ | ☐ ChIP-seq |
| ☒ | ☐ Flow cytometry |
| ☒ | ☐ MRI-based neuroimaging |

## Antibodies

| | |
|---|---|
| Antibodies used | AP2-alpha (Santa Cruz Biotechnology; sc-12726; clone 3B5; 1:200); AP2-gamma (R&D Systems; AF5059; 1:500); AP2-gamma (Santa Cruz Biotechnology, sc-12762; clone 6E4/4; 1:200); Brachyury (R&D Systems; AF2085; 1:500); CDX2 (BioGenex; MU392-UC; clone |

CDX2-88; 1:200); CER1 (R&D Systems; AF1075; 1:500); Cytokeratin 7 (Aligent; M7018; clone OV-TL 12/30; 1:100); E-Cadherin (BD Biosciences; 610182; clone 36; 1:200); EOMES (Abcam; ab23345; 1:200); FOXA2 (R&D Systems; AF2400; 1:200); GATA2 (Novus Biologics; NBMP1-82581; 1:200); GATA3 (Abcam; ab199428; clone EPR16651; 1:500); GATA4 (Santa Cruz Biotechnology; sc-25310; clone G-4; 1:200); GATA4 (ThermoFisher Scientific; 14-9980-82; clone eBioEvan; 1:500); GATA6 (R&D Systems; AF1700; 1:500); GATA6 (Cell Signaling Technology; 5851; clone D61E4; 1:2000); GFP (Abcam; ab13970; 1:1000); GFP (Nacalai USA, GF090R; clone GF090R; 1:1000); HAND1 (DSHB; PCRP-HAND1-2A9; clone 2A9 1:200); HNF4-alpha (Abcam; ab201460; clone EPR16885-99; 1:2000); ISL1 (DSHB; PCRP-ISL1-1A9; clone 1A9; 1:100); Laminin (Sigma Aldrich; L9393; 1:200); N-Cadherin (Abcam; ab98952; clone 5D5; 1:200); NANOG (Cell Signaling Technology; 4903; clone D73G4; 1:200); OCT3/4 (Santa Cruz Biotechnology; sc-5279; clone C-10; 1:100); OTX2 (R&D Systems; AF1979; 1:1000); phospho-SMAD1/5 (Cell Signaling Technology; 9516S; clone 41D10; 1:200); Smad2/3 (Cell Signaling Technology; 8685S; clone D7G7; 1:200); SOX17 (R&D Systems; AF1924; 1:500); SOX2 (ThermoFisher Scientific; 14-9811-82; clone Btjce; 1:500); TBX20 (R&D Systems; MAB8124; clone 668710; 1:100); VTCN1 (Abcam; ab209242; clone EPR20236; 1:200).

AlexaFluor-405 Donkey Anti-Mouse (ThermoFisher Scientific; A48257; 1:500); AlexaFluor-488 Donkey Anti-Rat (ThermoFisher Scientific; A-21208; 1:500); AlexFluor-488 Donkey Anti-Mouse (ThermoFisher Scientific; A-21202; 1:500); AlexaFluor-488 Donkey Anti-Goat (ThermoFisher Scientific; A-11055; 1:500); AlexaFluor-568 Donkey Anti-Rabbit (ThermoFisher Scientific; A10042; 1:500); AlexaFluor-568 Donkey Anti-Rat (ThermoFisher Scientific; A78946; 1:500); AlexaFluor-568 Donkey Anti-Mouse (ThermoFisher Scientific; A10037; 1:500); AlexaFluor-647 Donkey Anti-Rat (ThermoFisher Scientific; A78947; 1:500); AlexaFluor-647 Donkey Anti-Goat (ThermoFisher Scientific; A-21447; 1:500); AlexaFluor-647 Donkey Anti-Rabbit (Thermofisher Scientific; A-31573; 1:500); AlexaFluor-647 Donkey Anti-Mouse (ThermoFisher Scientific; A32787; 1:500).

Antibody table is provided in Supplementary Table 5

| Validation | All primary antibodies are validated for detection of the human antigen of interest according to manufacturer's websites. Details of the validation statement, antibody profiles and relevant citations can be found on the manufacturer's website. In addition to that, all antibodies in this study showed expected staining patterns based on protein type (e.g. transcription factors in the nucleus, membrane-bound proteins at the membrane) in human embryonic stem cells. |
|---|---|

## Eukaryotic cell lines

Policy information about cell lines and Sex and Gender in Research

| Cell line source(s) | UK Stem Cell Bank |
|---|---|
| Authentication | Cell lines were authenticated by STR analysis. |
| Mycoplasma contamination | All cell lines were regularly tested for mycoplasma contamination, and were negative. |
| Commonly misidentified lines (See ICLAC register) | N/A |

## Animals and other research organisms

Policy information about studies involving animals; ARRIVE guidelines recommended for reporting animal research, and Sex and Gender in Research

| Laboratory animals | CD1 and F1 wildtype males aged 6 to 45 weeks and CD1 and F1 wildtype females aged 6 to 18 weeks were used for this study. |
|---|---|
| Wild animals | No wild animals were used in this study. |
| Reporting on sex | Sex was not considered in this study as embryos were recovered from the mother and used for experimentation at the 8-cell stage. Genotyping was not performed. |
| Field-collected samples | No field-collected samples were used in this study. |
| Ethics oversight | Mice were kept in an animal house on 12:12 hour light-dark cycle with ad libitum access to food and water. Experiments with mice are regulated by the Animals (Scientific Procedures) Act 1986 Amendment Regulations 2012 and conducted following ethical review by the University of Cambridge Animal Welfare and Ethical Review Body (AWERB). Experiments were approved by the Home Office under Licenses 70/8864 and PP3370287. CD1 and F1 wildtype males aged 6 to 45 weeks and CD1 and F1 wildtype females aged 6 to 18 weeks were used for this study. Animals were inspected daily and those showing health concerns were culled by cervical dislocation. |

Note that full information on the approval of the study protocol must also be provided in the manuscript.

