## [Peer Review File · Nature]

Manuscript Title: Pluripotent stem cell-derived model of the post-implantation human embryo

Reviewer Comments & Author Rebuttals

Reviewer Reports on the Initial Version:

Referees' comments:

Referee #1 (Remarks to the Author):

Weatherbee et al. describe a model of post-implantation human embryonic development derived from human pluripotent stem cells (hPSC). The authors use transcription factors to drive differentiation of hPSC into trophoblast and extraembryonic endoderm lineages. Then, they aggregate these cells with undifferentiated hPSC and show using immunofluorescence microscopy and scRNA-seq that in 3D culture, structures develop that contain multiple lineages of the post-implantation embryo in appropriate anatomical relationships. They demonstrate that BMP signaling drives formation of amnion and extraembryonic mesenchyme, in accordance with previous results from other groups.

Cell culture models of the post-implantation embryo have the potential to provide unique insights into this stage of human development. Many groups are actively pursuing the goal of generating embryo models that capture all relevant cell types and recapitulate the morphogenetic events that give rise to all the characteristic embryonic structures from implantation to gastrulation. To date, this goal remains elusive. Strengths of this study include a novel approach to producing trophoblast and primitive endoderm precursors, and a model that yields key lineages including amnion, extraembryonic mesenchyme and primordial germ cells in appropriate spatial configurations. Weaknesses of the study include the use of a cell of origin (RSeT) whose precise developmental equivalent is unclear, the inability of the system to support maintenance of trophoblast and epiblast lineages, and its reliance on one hPSC line only to support conclusions of the study.

Specific comments:

L83-it is true that establishment and maintenance of trophoblast, extraembryonic endoderm and pluripotent epiblast in vitro requires different culture conditions. However, post-implantation primate embryos can be cultured in vitro with maintenance of these lineages. The authors should discuss this apparent discrepancy somewhere, since the incompatibility of the culture systems for the three lineages is the basis of their strategy, and apparently epiblast and trophoblast-like cells do not persist long in their system.

L 106 and elsewhere-it appears that only one cell line was used in this study. The authors should show that their system is sufficiently robust to support induced differentiation and post-implantation development across different hPSC lines. There are many variables in the protocol (starting media, transgene induction, aggregation) and what is described here may not work well if

at all for other cell lines. Key experiments should be repeated in at least one independent hPSC line.

L 108-what is meant by cells across the pluripotency spectrum? Specify here what was actually done.

L131-what exactly RSeT cells correspond to in developmental terms is not fully clear. The choice (and apparently superiority) of this platform is curious, does it really represent something markedly different to conventional primed cells? The starting point for generating the model is critical, but it is not clear why RSeT perform better. Reference 26 employed the naive cells described by the Smith and Nichols lab as well as RSeT; those described by Smith and colleagues are bonafide naive cells and can certainly generate trophectoderm and extraembryonic endoderm.

L 115 Extended Data Figure 1 a-it is very difficult to read this plot, consider reformatting.

L 136 Extended Data Figure 1cd-explain the three conditions listed on the bottom of the heatmap; each cell line is grown initially in three media (top of heatmap), but how do these relate to what is shown below.

L148 What is quantified in Extended Figure 2 bd-qPCR or immunostaining

L180-Extended data Figure 4-explain that this figure relates to 1c

L 214 Figure 2 ef-it is excellent that the authors report the efficiency of embryoid formation. Can they tell us what proportion of structures displayed all of the features scored in 2f and illustrated in the fluorescence images?

L236 what proportion of structures met the criteria for scRNA-seq?

L 244 Figure 3b epiblast present only at D6?

L258 why no trophoblast present? If these cells are not trophoblast, were they wrongly specified to begin with or did the culture conditions somehow convert them to something different?

L264-are the authors stating that the epiblast vanished prematurely? What about cells said to express late epiblast markers in scRNA-seq experiments? What proportion of the total cells did these represent?

L316- Boroviak and colleagues reported high expression of Nodal in hypoblast; does the absence of this factor account for early extinction of epiblast in the model

L365- what if the transgene is switched off at Day 4-6after aggregation, does CER1 domain form and can epiblast be protected

L 404-are RSeT cells naive cells? What are the authors trying to claim here?

Referee #2 (Remarks to the Author):

I was asked to conduct an ethics review of the manuscript “Transgene directed induction of a stem cell-derived human embryo model.”

The ethics statement within the methods section notes appropriate approvals were obtained from UK authorities for use of human embryonic stem cells and human embryos in research. Additional approval for the project was obtained from the University of Cambridge’s Human Biology Research Ethics Committee. However, the several authors are located (or co-located) in the United States, but no US approval was listed. The author contribution statements note that the work on embryoids and using human embryos seems to have been conducted in the UK. Therefore, I would request that the authors add a specific statement to the ethic session that the work was conducted in the UK to make this clear, if it is the case. I would also suggest the authors add a note that the research follows the 2021 International Society for Stem Cell Research (ISSCR) guidelines, since several co-authors are US-based.

Aside from these minor edits, this manuscript and the research conducted seem adequate for publication, from an ethical perspective.

Referee #3 (Remarks to the Author):

In this manuscript, the authors present an invitro model of human post-gastrulating embryo using transgene induction. Hypoblast-like cells are derived using induction of Gata6 and Sox17 in ES cells, tropho-blast like cells are derived using induction of Gata3 and Sox2, and the transduced cells are cultured with ES cells in a system pioneered by the authors that successfully recapitulated mouse embryogenesis invitro. The authors first verified the successful induction of trans genes and performed single-cell multiome experiments of individual lines to verify the reprogramming and then undertook extensive optimization of culture conditions for co-culture. The co-cultured embryos were profiled using single-cell multiomics following which cell-types were annotated using known genes, and signatures from single-cell profiles of other primates. Cell types resulting from the invitro embryos include embryonic, extra-embryonic, amniotic cells and primordial germ cells. Next, single-cell data was used to identify a role for BMP specification in these embryos and finally the authors demonstrate that Sox17 induction, a transgene used opposes the specification of anterior hypoblast.

The central contribution of the manuscript is a toolkit with an ability to mimic process of post gastrulation human embryos, invitro. While this is undoubtedly of importance, the manuscript suffers from several major issues which preclude a reliable interpretation of presented results in many scenarios including lack of detail of computational analysis, unclear importance of the induced trophoblast cells, and lack of evidence beyond marker expression for identification of cell types such as PGCs. The results presented by the authors denote sufficiently justify the claimed modular nature of the model. The comments are detailed below.

1. General computational analysis of the data

The methods description of the manuscript is lacking sufficient detail to critically assess many components of the analysis and for reproducing the analysis in the data. Even though the authors provide the code used, the details should be clearly described, choices justified. General examples are listed below:

- a. Integration of public single-cell data: This contains a wide variety of datasets across single-cell platforms and organisms. The authors should present a detailed description and visuals of the integration and discuss the quality of integration. Does the integration sufficiently correct for the batch effects? Does the FindMarkers approach use the appropriate statistics to identify regulons since it was designed for gene expression? The details presented currently make it hard to assess whether the TFs chosen for induction are data-driven.
 - b. No details are provided on the logistic regression approach used in multiple instances? How were the genes chosen? What are the training, test and validation sets? What datasets were used? Without these details, the logistic regression results cannot be reliably interpreted
 - c. Cell type transfers for the newly generated data were performed using scmap. The authors provide details on how scmap was applied but additional discussion is needed to discuss any potential pitfalls: for example, how does scmap handle cell-types that are relevant to the dataset?
 - d. QC plots of the single-cell data should be provided given the highly novel nature of the data
2. Derivation of transduced cells and the tri-lineage embryo
- a. The authors need to discuss how critical it is for the transduced cells to have shut down the embryonic / pluri potent program given the target roles. In particular Oct4 is missing in most of the qRT-PCR / immunofluorescence experiments. Oct4 in my opinion is a necessary gene to include in this analysis. Do the transduced cells retain their identity in culture or are they plastic to revert back to embryonic cells? While this is demonstrated for the trophoblast-like cells using GFP, no such evidence is provided for the hypoblast-like cells.
 - b. The logistic regression results in 1D suggest that Day 3 G6-S17 are more similar to trophoblast rather than Day 3 G3-Ap cells, and that Day G3-Ap are also similar to Epiblast cells? Is this an issue with signature, gene sets or does it reflect the underlying identity of the cells?

Minor comments

- a. More details needed to be provide the choice of mixing ratios and how it relates the invivo embryo
 - b. Panel name is missing for "2H"
 - c. Day of collection is not specified for images in Fig. 2
3. Role of G3-Ap cells:
- a. The authors claim through the results in Fig. 2C, that there is a single layer of Gata3 positive cells surrounding the embryo - however it appears that the Gata6+ cells substantially outnumber the Gata3+ even on the boundary? A detailed analysis of this particularly critical given point 2b. Perhaps showing these genes on a separate panels will provide better clarity. Given the centrality of the claim, the authors need to generate an analysis pipeline to count the number of Gata6+ cells on the boundary (and interior) and provide statistical evidence rather than the observational evidence.
 - b. Further, the authors also mention that G3-Ap are most likely not bonafide trophoblast cells. Given this and observations of low numbers of G3-Ap cells on the embryo boundary and their lack of signal in the logistic regression, raises the question whether G3-Ap cells are necessary for the embryo formation through this culture system.
4. Generation of primordial germ cells: The evidence provided for generation of bonafide PGCs are

not convincing.

a. The signature scores in Fig. 3E show a surprisingly small range and the difference between PGC and non-PGC scores are rather minimal. Do the scores show a similar range in the dataset derived or is there more contrast?

b. How were the cluster of cells termed "PGCs" in Fig. 3F determined? How were the genes chosen?

5. BMP signaling response

a. All the observations about trends and dynamics of gene expression in 4A are based on observations on umap and is highly qualitative. The authors should provide better estimates of these dynamics by fitting these trends along a pseudo-time axis (eg: multivelo latent time) and provide more concrete evidence for the dynamics described in the manuscript.

b. While it is interesting to investigate the trends of ID1-4, are the gene signatures scores of BMP signaling response genes consistent with these observations?

c. Similarly to (a), the TF signals in 4B are purely qualitative inferred using observations on umap. The insilico-ChIP tool (<https://doi.org/10.1101/2022.06.15.496239>) might be more suitable for inferring TF signals and dynamics should be demonstrated by fitting trends along pseudo-latent time.

6. Sox17 effect on hypoblast and modularity of tool: The evidence provided to demonstrate that the invitro model is modular is not definitive.

a. The viability, efficiency and recover of cell-types through induction of Gata6 is not described in detail to justify the claim of modularity.

b. The details of reanalysis in Fig. 5B in the Methods does not include several details such as how were cells chosen, what is the full umap, what are the different cell types presented in the umap etc.

c. Cer1 and Sox17 coregulation statistics are not presented and the significance does not seem to be assessed.

7. The authors use known cell-type markers to infer the derivation of different cell-types. For example, Hand1, Foxf1, Tbx20 - please add relevant citations and discuss the reliability of these markers for cell-type identification.

Referee #4 (Remarks to the Author):

This manuscript by Weatherbee et al. reports the establishment of an pluripotent stem cell-derived model of early human embryo development achieved via aggregation of both extraembryonic and embryonic cellular components in vitro. The two distinct extraembryonic cellular components, putative hypoblast- and throphoblast-like cells are generated by direct reprogramming of intermediate pluripotent stem cells via the overexpression of specific transcription factors, GATA6 and SOX17 for the hypoblast and GATA3 and AP2γ for the trophoblast. The resulting aggregates and embryo-like structures are obtained with an efficiency of 23% and recapitulate several morphological and molecular hallmarks of post-implantation human embryo development including the presence of a central SOX2 positive cell population surrounded by two concentric layers of GATA6 and GATA3 positive cells. This model shows furthermore the presence of extraembryonic mesenchyme, amnion and PGCs cells.

Overall, it is a well written and timely manuscript which focuses on the in vitro recapitulation of peri-implantation human development. The obtained data and shown results appear to be largely of good/acceptable quality but the meaning and potential usefulness of this model is unfortunately overhyped and exaggerated by the authors. This is also reflected by the misleading title of the manuscript, which implies that the authors have succeeded to establish a proper and complete model of the human embryo.

This is by no means the case. This “embryoid” model system shows and recapitulates some aspects of embryonic development, especially cellular differentiation but the overall morphological characteristics of an actual post-implantation human or primate embryo are not recapitulated and critical cell types such as bona fide trophoblasts are missing.

Major Concerns and Comments to the Authors:

One of the core issues of this manuscript lies within the way the different extraembryonic cell lines are initially generated and ultimately assessed. Based on computational analysis the authors identified several gene candidates for the direct specification of pluripotent stem cells into hypoblast cells: GATA4, GATA6, SOX17, FOXA2 and trophoblast cells: GATA2, GATA3, NR2F2, TFAP2C. From this shortlist of transcription factors they then selected two genes as candidates for direct reprogramming of human PSCs into hypoblast or trophoblast cells and assessed the efficiency of such induction starting from cells at three different states of pluripotency, i.e. naïve, intermediate peri-implantation and primed pluripotency states. Transcripts were expressed in these cells for three (3) days via doxycycline induction and efficiency of conversion & differentiation was evaluated using RT-qPCR.

(1) The outcome and efficiency of these experiments seems to be very variable depending of the pluripotent state of the initial cells. Could the authors elaborate on this? Can we imagine that the transcription factors used are not sufficient to rewire the differentiation/reprogramming barriers preexisting in the different cell types used in this study?

(2) The generation of cell lines was done using the piggyBac transposon system. Different clones were selected using single colony manual picking which does not exclude heterogeneity in the copy numbers and levels of expression of the introduced transgenes even between different cells of a clone. This raises the question of the relevance of the level of expression of those transgenes for obtaining cells with the desired identity, i.e. hypoblast or trophoblast. Was there any quality control of these cells? The authors need to provide info/data on heterogeneity, copy numbers, integration sites, levels of expression of the different transgenes compared to endogenous expression levels in a real embryo. How many cell lines/clones were used for the initial characterization and for the actual experiments? Did the authors observe any differences between these clones for both extra embryonic lineages? What was the variability and reproducibility of outcomes between clones when using same conditions?

(3) Reference 42 suggests that GATA6 expression in hESCs (primed) results in iXEN-like cells with important heterogeneity in hypoblast related protein expression and implies that timing/conditions need to be adjusted to improve differentiation outcomes. Reference 43 indicates that SOX17

expression in hESCs (primed) orients cells toward a mesendoderm identity whereas another paper from Janet Rossant and colleagues addressing a similar topic in murine ESCs (naïve) (McDonald et al. 2014) suggests that transient expression of SOX17 for only 48h is sufficient to efficiently convert mESCs to XEN cells. Similarly, regarding trophoblast induction, reference 44 used as the basis for the selection of GATA3 and TFAP2C, clearly indicates that GATA3 is important but does not support a role for TFAP2C in this process. TE reprogramming in human has still not been clarified and explained conclusively and is suggested to rely on a complex interaction of at least four genes (Papuchova et al., 2022). What was the scientific rationale for excluding the other two genes? The authors furthermore report in the manuscript that the selected genes only work in combination with particular conditions and fail to generate bona fide trophoblast stem cells. What are these cells if not trophoblast cells? These cells seem to also express genes of other lineages which clearly indicates that the initially intended/postulated differentiation of PSCs into the two major extra embryonic lineages has not worked (at least not as claimed by the authors).

(4) The authors should provide further data and evidence for their experimental rationale and claim that these identified transcription factors either alone or in combination with each other can give rise to defined extra embryonic lineages and cell types. Other combinations which were excluded should be also tested and assessed accordingly.

(5) The authors evaluate the efficiency of hypoblast or trophoblast differentiation at day three of doxycycline-driven transcript induction in all three conditions (naïve, intermediate, primed) and identify several differences in this regard. Could we hypothesize that the observed differences between cells induced from initially naïve or primed pluripotent stem cells could be the effect of the temporality of the reprogramming process, and that earlier or later time points might show different results and even overlap?

(6) The authors decided to use ReSET ESCs expressing the gene combination GATA6-SOX17 to induce hypoblast cells. Under these conditions GATA4 expression is not detected both by qPCR and IF (Ext. Data Fig. 1 C and 2B). Could the authors elaborate on this?

(7) In Ext. Data Fig. 1D and 2D panels for the naïve PXGL state are missing for iAy whereas they are present for the primed state mTeSR, while in the text it is clearly mentioned that primed cells die or lose transgene expression upon iAy expression. Were the panels switched by mistake or is the annotation inverted?

(8) Based on the result shown in Ext. Data Fig. 1D and 2D it seems that ReST iAy is better than ReST iG3-iAy. Why was this cell line not used instead?

(9) Did the authors compare the RNA seq data of the induced cells to previously reported XEN cells generated by overexpression of SOX17 (PMID:25373912) or GATA6 (PMID:26109048)?

(10) For the comparison of protein expression between transgene induced hypoblast and trophoblast cells the authors used RSeT cells as a starting material. The authors should compare these with the cells obtained via naïve cell direct differentiation and reported previous models of blastoids.

(11) In Ext. Data Fig. 4 TFAP2C population seems to overlap with the NANOG/SOX2/POU5F1 population rather than with the GATA3 population, as one would normally expect. Moreover the same population seems to also overlap with the CDH1+ population and E-Cad staining in Figure 2G is restricted to the pluripotent population. Is the TFAP2C population in the structure equivalent to or part of the epiblast and potentially not related to the putative trophoblast?

By aggregating the previously reprogrammed cells with wild-type (control) RSeT pluripotent stem cells in the specific proportion of 1:1:2 of WT:GATA6-SOX17:GATA3-AP γ the authors manage to generate 3D structures that self-organize with an efficiency of 23%. Those structures have an inner cellular domain expressing pluripotent stem cell markers, and an outer cellular domain constituted of two concentric layers of cells expressing GATA6 or GATA3.

(12) How was the proportion 1:1:2 defined and selected? Did the authors test any other cell proportions and what were the results? The authors mentioned that a proportion of 1:1:1 cells was initially evaluated and deemed appropriate. How and why did they change these proportions? The authors should assess this potentially critical point more systematically.

(13) Related to this aspect, no data is shown for the relevance of a seven (7) day doxycycline induction. The previous analysis were done at three (3) days of induction. What is the rationale for this choice? Other reports in the literature suggest that a short pulse of SOX17 is enough to induce an extraembryonic endoderm phenotype. What is the temporal rationale for the putative trophoblast cells? Given the fact that the authors opt for continuous (one week long) induction the relevance of an inducible system is questionable. Wouldn't a constitutive or a combination of both inducible and constitutive expression be more relevant for the model proposed by the authors?

(14) The authors assess (the efficiency of) their model system using PXGL and primed state cells as a starting material, but always using a combination of the two transgenes. The authors should also perform and provide data for such assessment/comparison using single transgene expression as described to be important in the latter half of the manuscript when talking about single GATA6 expressing cells.

(15) Throughout the manuscript, indications in blue for Laminin, N-Cad and other used antibodies (blue color) are very hard to read, please change the shade of blue used.

(16) In Figure 2G, N-Cadherin staining seems to be not restricted to the membrane of cells. The authors should validate their results using an alternative antibody and/or alternative methods.

(17) E-Cad staining is membranous but seems to be present on the basal and apical pole of cells, in contradiction with the statement of the authors that epiblast equivalents are properly epithelialized. Could the authors describe the expression pattern of other apical or basal markers such as ZO1, aPKC or Integrins?

(18) In the described model and in vitro structures the morphology of an actual human/primate embryo can clearly not be recapitulated. This raises the question of the relevance of this model

system as an in vitro embryo of human embryogenesis. The strong and largely exaggerated claims by the authors should be toned down and make clear that this is an “embryoid” model system that only partially recapitulates some aspects of the actual human embryo.

(19) In Figure 2G we can see an “embryoid” with 2 lumens in the central putative epiblast domain. What is the proportion of embryoids containing a single lumen vs multiple lumens at Day4, 6 and 8? This aspect should be worked out and mentioned clearly as it is relevant to back-up the core claim of the authors that they can recapitulate aspects of post-implantation human development.

(20) Regarding the 23% success rate claimed by the authors, what were the inclusion and exclusion criteria used to measure this success rate? How was it calculated and what are the success rates for the other conditions the authors have assessed?

(21) In Figure 2G bottom panel, what is the relevance of the human embryo shown here? The staining is done for CER1 which has no comparable staining for the induced structures in this figure.

Next, the authors performed single cell RNA/ATAC (multiome) sequencing of their induced structures at Days 4, 6 and 8 of culture and compared the obtained gene signatures with available datasets of human and cynomolgus monkey embryos. Based on these results they concluded that bona fide trophoblast cells are absent from their structures, but that they have extraembryonic mesenchyme, amnion and PGC cells.

(22) The RNA-seq data obtained by the authors should be compared with other relevant datasets available for both human embryo models and in vitro cellular components, e.g. trophoblast cells generated in vitro (Dong et al., 2020; Liu et al., 2020, Viukov et al., 2022), extraembryonic mesenchyme (Pham TXA et al., 2022), human peri-implantation embryos (Yuan et al., 2022; <https://doi.org/10.1101/2022.06.02.494565>), primate embryos (Zhai et al., 2022) or the latest human blastoid structures (Kagawa et al., 2022)

(23) The authors identify no trophoblast cells in their model system. Please clearly state what these cells are then and elaborate on their role as it seems that they are needed to generate embryoids/embryo-like structures.

(24) In Figure 3G, from what original cell population do the putative PGC cells (SOX17/NANOG/AP2γ) cells originate from? Are they emerging from the population that expresses SOX17 or AP2g or the wild type pluripotent stem cells?

(25) In Ext. Data Fig. 5C, why is the SOX17 population so small?

(26) In Ext. Data Fig. 6, the HAND1 staining seems to overlap with the AP2g one, but in Ext. Data Fig. 5C and Ext Data. Fig. 6B the RNA signatures do not overlap. The authors should explain this discrepancy. What are these cells TFAP2C negative HAND1 positive cells?

(27) In Ext. Data Fig. 6, FOXF1 expression seems to be polarized. Has this been previously reported? What is the putative biological meaning of this expression pattern? What do the FOXF1 high or low

populations correspond to?

(28) The statement regarding the segregation of hypoblast cells (GATA6+/FOXF1 or TBX20 low) compared to extraembryonic mesenchyme cells (GATA6+ FOXF1 or TBX20 high) is very hard to grasp as FOXF1 high cells seems to be localized to one pole of the structure whereas TBX20 is very diffuse. If possible, staining for FOXF1 and TBX20 should be done in parallel on the same structure. Moreover, other techniques such as in situ hybridization e.g. HCR could be used to rule out any non-specificity or cross-reactivity of used antibodies.

(29) Please add references that support the authors statement that FOXF1 and TBX20 levels of expression are used to segregate those two cell populations.

(30) Regarding the two embryoids shown for TBX20 staining in Ext. Data Fig. 6E, one seems to be TBX20 high whereas the other TBX20 low, but there seems to be no obvious variability in TBX20 staining inside same embryo. Could the authors please elaborate and explain?

(31) The model system of the authors does not show any regionalization of amnion cells vs. pluripotent epiblast cells. It seems that the entire central epithelial population acquires either one or the other fate. The claim of the authors that proper amnion cell development can be recapitulated in their model system needs to be toned down. This is also one of the major discrepancies of this model with actual human/primate embryos. This should be clarified and reflected in the manuscript in a critical and appropriate manner.

Next, the authors analyze the effect of BMP signaling on amnion specification by analyzing target genes ID1-4 and SMAD5 vs SMAD2-4 gene expression and the phosphorylation status of SMAD 1/5 vs 2/3 in the induced structures.

(32) PGCs cell development is known to be dependent on BMP4 signaling. Can the authors elaborate on whether PGCs are still present inside the structures when BMP is inhibited and how they might be affected by modulation of BMP signaling.

(33) In Figure 4F and G, structures treated with BMPs or BMP agonists should be added as a control.

(34) What is the source of BMPs in the structures? Is it the GATA3-APy cell population? Could the authors stain for BMP4s in their in vitro derived structures with or without CER1 population?

(35) In the structures which express CER1, could the authors confirm using IF that these cells express Noggin?

Finally the authors use their system to analyze the effect of sustained SOX17 expression on the formation of the anterior hypoblast and its effect on embryo morphogenesis.

(36) GATA6 induction alone can generate structures with CER1 and increased TBXT expression at day 6. It seems that these cells could be used instead of and replace the GATA6-SOX17 cells used throughout the manuscript to generate embryoids. The authors should elaborate this point in detail

and especially in relation to our previous comment on the relevance of long-term induction of GATA6 and SOX17 alone or in combination, especially as the literature indicates that short-term induction might be sufficient to obtain similar results.

(37) What would be the effect on CER1 and TBXT expression when doxycycline is removed earlier during the aggregation process for example on Day 1 or Day 2?

(38) What are the TBXT cells in the structures? What is their origin? Are they derived from wild type PSCs or could they also emerge from the other induced cell populations including putative hypoblast cells? Do these TBXT cells population organize and form a primitive streak as suggested by the authors? The “standard model” of the authors appears to fail to induce TBXT positive cells. SOX17 is known to be involved in the differentiation of both mesoderm and endoderm. Addition of SOX17 could bias the differentiation of cells into endoderm while lack of it might lead to the formation of TBXT mesodermal cells in the hypoblast population present in the model of the authors. The authors should elaborate and comment on this possibility.

(39) What are the patterns of expression of pSMADs in the GATA6 positive structures where CER1 is expressed? What is the effect of SOX17 only expression and can these cells generate equivalent structures?

Author Rebuttals to Initial Comments:

Summary

We thank the reviewers for their thoughtful feedback, which has significantly improved the manuscript. Below we respond to specific comments in blue. Please note that this document has its own reference list, and the reference numbers may differ from those in the manuscript. Throughout, we provide figure panels where new data has been added or revised and provide details on where this is included in the updated manuscript. Line references are specific to the updated manuscript file submitted alongside this *response to reviewers* document. Text added

In summary, we have added significant new data, including:

1. New experiments investigating the effects of withdrawing doxycycline from our post-implantation human embryo model at day 1 or day 3 post-aggregation and showing that withdrawal of doxycycline at day 3 is sufficient to rescue CER1 expression within the anterior hypoblast-like population.
2. Evidence that the origin of amnion, primordial germ cell-like cells, and extraembryonic mesenchyme is indeed from the epiblast-like domain of the embryoids as expected.
3. Validation with an independent embryonic cell line (RUES2; RUES2 inducible GATA6-SOX17; and RUES2 inducible GATA3-AP2 γ) which reproduce our data with Shef6 cells, including self-organization and differentiation.
4. Characterization of the expression of several additional markers in our inducible human embryo model – specifically, GATA4, ZO-1, PARD6, and pSMAD1.5.
5. Further transcriptomic comparisons to other *in vitro* models, including blastoids¹, post-implantation amniotic sac embryoids (PASE)², and directed differentiation protocols for trophoblast and extraembryonic mesenchyme³.
6. Examination of the effects of BMP and NODAL perturbations on primordial germ cell-like cell differentiation, demonstrates that BMP is essential for primordial germ cell-like cell differentiation in our human embryo model.
7. Generation of interspecies chimeras to support the extraembryonic-like identity of induced cells *in vitro*.
8. Further clarification of the rationale and methodology where appropriate.

We believe that the reviewers' comments have enabled us to significantly improve our study. We hope the reviewers would agree that the work marks a meaningful step towards generating a multilineage, stem cell-derived model of the post-implantation human embryo, which will permit investigations of this critical and understudied period.

Specific Comments

Referee #1:

Weatherbee et al. describe a model of post-implantation human embryonic development derived from human pluripotent stem cells (hPSC). The authors use transcription factors to drive differentiation of hPSC into trophoblast and extraembryonic endoderm lineages. Then, they aggregate these cells with undifferentiated hPSC and show using immunofluorescence microscopy and scRNA-seq that in 3D culture, structures develop that contain multiple lineages of the post-implantation embryo in appropriate anatomical relationships. They demonstrate that BMP signaling drives formation of amnion and extraembryonic mesenchyme, in accordance with previous results from other groups.

Cell culture models of the post-implantation embryo have the potential to provide unique insights into this stage of human development. Many groups are actively pursuing the goal of generating embryo models that capture all relevant cell types and recapitulate the morphogenetic events that give rise to all the characteristic embryonic structures from implantation to gastrulation. To date, this goal remains elusive. Strengths of this study include a novel approach to producing trophoblast and primitive endoderm precursors, and a model that yields key lineages including amnion, extraembryonic mesenchyme and primordial germ cells in appropriate spatial configurations. Weaknesses of the study include the use of a cell of origin (RSeT) whose precise developmental equivalent is unclear, the inability of the system to support maintenance of trophoblast and epiblast lineages, and its reliance on one hPSC line only to support conclusions of the study.

We thank the reviewer for their thoughtful comments on our manuscript and address their comments below.

Specific comments:

L83-it is true that establishment and maintenance of trophoblast, extraembryonic endoderm and pluripotent epiblast *in vitro* requires different culture conditions. However, post-implantation primate embryos can be cultured *in vitro* with maintenance of these lineages. The authors should discuss this apparent discrepancy somewhere, since the incompatibility of the culture systems for the three lineages is the basis of their strategy, and apparently epiblast and trophoblast-like cells do not persist long in their system.

We are grateful for this suggestion. We agree with the referee that post-implantation primate embryos can be cultured in conditions supporting all three lineages, indicating that the embryo can successfully differentiate and maintain the three lineages of the blastocyst/peri-implantation embryo. Importantly, the differentiation of the amnion, extraembryonic mesenchyme, and primordial germ cell-like cells is observable in cynomolgus macaque, but not robustly in human embryo culture⁴⁻⁸, which we note in the manuscript. To generate a stem cell model of the human embryo, we chose to utilize a transcription factor-mediated reprogramming approach as this allows us to circumvent the opposing signaling requirements to derive trophoblast and/or hypoblast *in vitro*. Once combined in our system we also use basal media (the same as used for *in vitro* human embryo culture and which can support the three major lineages of the embryo at these time periods) with the addition of doxycycline to drive extraembryonic-like identity. However, as the reviewer notes our human embryo model does not maintain all lineages. We have now added the following in the discussion (L508-511):

<< Our post-implantation embryoids are able to self-organize and in rare cases show axis formation. Similar to other human embryo models such as blastoids, further optimization is needed to permit the maintenance of all major lineages of the post-implantation embryo with their full developmental potential and embryo-like morphology. >>

L 106 and elsewhere-it appears that only one cell line was used in this study. The authors should show that their system is sufficiently robust to support induced differentiation and post-implantation development across different hPSC lines. There are many variables in the protocol (starting media, transgene induction, aggregation) and what is described here may not work well if at all for other cell lines. Key experiments should be repeated in at least one independent hPSC line.

We agree with the reviewer that reproducibility across cell lines is very important. We have therefore carried out new experiments in which we generated inducible cell lines in a second cell background (RUES2, human embryonic stem cell line derived Rockefeller University⁹) (Extended Data Figs. 5a and 7e-f). Inducible post-implantation human embryo models generated with RUES2 wildtype ESC, RUES2-inducible GATA6-SOX17, and RUES2-inducible GATA3-AP2 γ self-organize similarly to embryoids derived from Shef6 ESCs, including an inner epiblast-like domain. Importantly, we confirmed our major findings in this second line, including the generation of primordial germ cell-like cells, formation of the amnion, and appearance of extraembryonic mesenchyme. We have included the relevant panels below:

Response to Reviewers Figure 1.1: Generation of Inducible Human Embryoids from an independent cell line (RUES2).

(A) RUES2-derived embryoids exhibit similar tri-lineage organized structure as Shef6-derived embryoids with an inner SOX2-positive domain, an outer GATA3-positive cells, and an interstitial GATA6-positive compartment. (B) Quantification of embryoid formation efficiency at day 4 post-aggregation for RUES2-derived embryoids. (C) RUES2-derived embryoids generate ISL1-positive amnion-like cells and GATA6/TBX20-positive extraembryonic mesenchyme-like cells at day 6 post-aggregation. (D) RUES2-derived embryoids generate SOX17/NANOG/AP2 γ triple-positive primordial germ cell-like cells. Scale bars = 100 μ m.

L 108-what is meant by cells across the pluripotency spectrum? Specify here what was actually done.

We have specified now in the text the three cell states used, and where we view them along the spectrum. The text now reads as follows (L125-128):

<< We cultured cells using three established starting conditions: PXGL, which supports pre-implantation-like naïve hESCs; RSeT, which generates intermediate peri-implantation-like pluripotent hESCs; and conventional mTeSR1 conditions to maintain post-implantation-like primed hESCs^{10,11}. >>

L131-what exactly RSeT cells correspond to in developmental terms is not fully clear. The choice (and apparently superiority) of this platform is curious, does it really represent something markedly different to conventional primed cells? The starting point for generating the model is critical, but it is not clear why RSeT perform better. Reference 26 employed the naïve cells described by the Smith and Nichols lab as well as RSeT; those described by Smith and colleagues are bonafide naïve cells and can certainly generate trophoblast and extraembryonic endoderm.

We agree with the reviewer that RSeT cells are not *bona fide* naïve cells but rather represent an intermediate pluripotency state, as noted in other studies¹⁰⁻¹².

We have previously found that cells generated by the RSeT protocol allowed for the derivation *post-implantation*-like lineages. Alternatively, cells generated by alternative naïve protocols such as those described in the Smith and Nichols lab (PXGL) derived *pre-implantation*-like lineages when subjected to similar differentiation regimes¹³. This was one of the reasons we tested RSeT cells as the starting material to generate a post-implantation model. We rigorously tested both PXGL (Smith/Nichols conditions) and primed cells for their ability to generate post-implantation embryoids, and they were routinely unable to do so (Fig. 2d). However, we do note that our use of RSeT cells may hinder our ability to derive *bona fide* trophoblast. For example, we note that induction of GATA3 alone upregulates a trophoblast-like gene program with markedly higher efficiency in PXGL cultured ESC in comparison to RSeT ESC (one reason we chose to use the dual inducible GATA3-AP2γ cells).

L 115 Extended Data Figure 1 a-it is very difficult to read this plot, consider reformatting.

We agree and have increased the font size of the genes in this plot.

Note this panel is now Extended Data Fig. 1f.

L 136 Extended Data Figure 1cd-explain the three conditions listed on the bottom of the heatmap; each cell line is grown initially in three media (top of heatmap), but how do these relate to what is shown below.

We have clarified this in the figure caption as follows:

<< Cells are initially cultured in either mTeSR, RSeT, or PXGL conditions, and then cultured for 3 days either under the same conditions or alternatively transferred to either basal N2B27 media or basal N2B27 media with the addition of doxycycline. >>

Note these panels are now Extended Data Fig. 2a-b.

L148 What is quantified in Extended Figure 2 bd-qPCR or immunostaining

These panels, which are now Extended Data Fig. 2d, f, present quantifications of immunostaining results. We have clarified this in the figure caption.

L180-Extended data Figure 4-explain that this figure relates to 1c

We have noted in the figure caption that the relevant UMAP visualization with cells colored according to sample of origin is visible in Fig. 1c.

L 214 Figure 2 ef-it is excellent that the authors report the efficiency of embryoid formation. Can they tell us what proportion of structures displayed all of the features scored in 2f and illustrated in the fluorescence images?

We thank the reviewer and agree that open reporting of efficiency is a crucial requirement for any novel stem cell model. As noted in the manuscript, ~23% of structures are sorted appropriately ('Sorted tri-lineage'), and ~17% have a SOX2-positive domain with a podocalyxin-positive lumen and Laminin-positive basement membrane ('SOX2+lumen+ECM'). Unfortunately, given the limited number of antibodies we can use for each immunofluorescence analysis we cannot provide a percentage of structures that are sorted appropriately ('Sorted tri-lineage') and that also have a podocalyxin-positive lumen ('Lumen') and a Laminin-positive ('ECM') basement membrane.

L236 what proportion of structures met the criteria for scRNA-seq?

The criteria used to select structures for single cell sequencing are similar to those described and visually apparent as 'tri-lineage organized' in Fig. 2f, which gives an efficiency of 23% at day 4 post-aggregation. For sequencing we selected 12 structures that met our criteria at each timepoint. These structures used for single cell sequencing were imaged and we show them in Extended Data Fig. 5b.

L 244 Figure 3b epiblast present only at D6?

The epiblast-like cluster is present also at Day 4 post-aggregation and we now revise the text to make it clearer. We agree that this is difficult to see due to the overlay of single cell points in the UMAP presented. To show this clearly, we have now included an alluvial plot (Extended Data Fig. 5f) to directly show the proportion of cells at each timepoint and their assignment to each cluster. We also include this plot below:

Response to Reviewers Figure 1.2: Alluvial Plot of Post-Implantation Embryoid Cluster Assignments.

Alluvial representation of cell types within embryoids present at each timepoint sequenced. For example, both day 4 and day 6 structures contribute to the L-EPI cluster, while contribution to differentiated lineages (including AM-2, AM-3 and EXMC) increases over time.

L258 why no trophoblast present? If these cells are not trophoblast, were they wrongly specified to begin with or did the culture conditions somehow convert them to something different?

We now include a direct analysis of GFP-positive GATA3-AP2 γ inducible cells within embryoids from our RNA-sequencing dataset (Extended Data Fig. 6d and below in Response to Reviewer Fig 1.3 below). Our analysis shows that cells derived from GATA3-AP2 γ inducible cells co-express alongside trophoblast genes (*GATA3*, *TEAD4*, *KRT19*, *GATA2*), endodermal genes (*GATA6*, *GATA4*, *SOX17*). We observe limited endodermal gene expression in 2D after GATA3-AP2 γ induction. In our opinion, this mixed gene signature is likely due to their position on the outside of the structure after aggregation, which has been shown to drive endodermal cell fate in embryoid bodies^{14,15}.

However, despite the loss of a clear trophoblast identity by day 4 after aggregation, the reviewer correctly points to the necessity of this cell type for successful embryoid formation. Excluding inducible GATA3-AP2 γ cells results in a failure of embryoid formation (Extended Data Fig. 8e). Given that CellPhoneDB analysis indicated that GATA3-AP2 γ cells may be the initial source of BMP at aggregation (Extended Data Fig. 8d), we hypothesize that these cells initially support aggregation and survival partially through acting as a BMP source. In support of this role, inhibition of BMP signaling between day 0-2 of embryoid formation results in decreased survival and lack of organized structures, highlighting the clear importance of BMP signaling in embryoid survival and organization (Response to Reviewers Fig 1.3H-J).

The following is included in the Discussion section (L482-490):

<< For example, while GATA3 and AP2 γ induction drive trophoblast-like gene programs in 2D, upon aggregation in our human embryo model, this cell population aberrantly upregulates

endodermal markers (including SOX17 and GATA6). This is potentially due to their position at the outermost layer of the structure^{14,15}. Nevertheless, the GATA3-AP2 γ inducible cells are required for successful embryoid organization and likely act as a crucial source of BMP at aggregation. Optimization to preserve the expression of trophoblast-like gene programs while preventing the aberrant upregulation of endodermal identity in these cells will be important to improve future iterations of this model. >>

Response to Reviewers Figure 1.3: Inducible GATA3-AP2 γ cells upregulate endodermal markers upon aggregation.

(A) The GATA3-AP2 γ inducible population (iG3-AP2 γ), marked by GFP, localizes to the outer layer of day 4 embryoids. (B) The GATA3-AP2 γ inducible population persists at the outer layer of embryoids at day 6 and day 8 post-aggregation. (C) GFP-positive cells in the embryoid RNA-sequencing data show enrichment of the endodermal markers *GATA6*, *GATA4*, and *SOX17*, in addition to *GATA3*, *TEAD4*, *GATA2* and other trophoblast associated genes. (D) Uniform manifold approximation projection visualization (UMAP) of sequencing data from cells used to generate embryoids prior to aggregation (i.e. after 3 days of doxycycline induction in 2D) colored by sample type. (E) Gene expression profiles of key lineage markers after 3 days of doxycycline induction in 2D. Note the lack of enrichment of *GATA6*, *SOX17*, and *GATA4* in the inducible GATA3-AP2 γ population. (F) CellPhoneDB analysis showing potential receptor-ligand interactions between inducible cell populations and wildtype ESCs, which may influence embryoid survival and/or organization. CellPhoneDB predicts the BMP4 source to RSeT hESCs to be the GATA3-AP2 γ inducible population. Colors for clusters on the x-axis match the receptor and/or ligand in the y-axis. (G) Violin plots showing gene expression of BMP ligands and receptors in the cells used to generate inducible human embryoids, confirming expression of BMP ligands in the GATA3-AP2 γ inducible population. (H) Inducible human embryoids do not form if induced GATA3-AP2 γ cells are excluded or if LDN193189

is added between days 0-2. (I) Quantification of embryoid formation efficiency from H. (J) Quantification of embryoid size after LDN193189 addition between days 0-2. Scale bars = 100µm. Statistics: (J) unpaired t-test. **p<0.01, ****p<0.0001.

L264 are the authors stating that the epiblast vanished prematurely? What about cells said to express late epiblast markers in scRNA-seq experiments? What proportion of the total cells did these represent?

We apologize for the lack of clarity on this point. As noted above, we have now included an alluvial plot that shows the contribution of embryoids at day 4, 6, and 8 post-aggregation to each of our sequencing clusters (Extended Data Fig. 5f; included above as Response to Reviewers Figure 1.2). We generally observe a progression of epiblast to amnion/EXMC identities, in contrast to what is observed in primate embryo samples. We have rephrased this section to reflect our results more appropriately (L276-279):

<< These assigned clusters showed differences in their composition based on the day of sample collection, for example with the L-EPI cluster comprised of only Day 4 and 6 structures and a progressive shift from AM-1 to AM-2 and AM-3 over time. (**Extended Data Fig. 5f**). >>

L316- Boroviak and colleagues reported high expression of Nodal in hypoblast; does the absence of this factor account for early extinction of epiblast in the model

We agree with the reviewer this is a possible factor and have included this point in our discussion as follows (L458-460):

<< Additionally, the low NODAL environment, combined with lack of an anterior hypoblast-like population may contribute to the differentiation of the epiblast-like population over time. >>

Notably, we have also carried out experiments where Activin-A (a NODAL agonist) is supplemented into the media between days 2 and 4. This results in a clear increase in embryo models containing a SOX2-positive pluripotent Epiblast-like domain and a concomitant decrease in the expression of genes corresponding to amnion maturation (TFAP2A, ISL1; Fig. 4e-f; included below for convenience).

Response to Reviewers Figure 1.4: Inhibition of BMP or activation of NODAL delays amnion differentiation. (A) Inhibition of BMP or activation of NODAL signaling blocks exit from pluripotency and upregulation of amnion markers AP2 α and CDX2. (B) Quantification of the percentage of inner domains expressing SOX2 and CDX2 at day 4. Scale bars = 100 μ m. Statistics: (B) RM Two-way ANOVA with Holm-Sidak's multiple comparison test. * $p < 0.05$, *** $p < 0.001$.

L365- what if the transgene is switched off at Day 4-6 after aggregation, does CER1 domain form and can epiblast be protected

As the referee suggests, we have now performed experiments in which we remove doxycycline either at day 1 post-aggregation or at day 3 post-aggregation and examined CER1 expression at day 4 and day 6. We have chosen these timepoints to capture two distinct phases: (1) removal of doxycycline at day 1 after aggregation and organization; and (2) removal of doxycycline at day 3 when structures are organized fully and are receptive to CER1 expression.

As shown below and in Fig. 5c-h, the removal of doxycycline at day 1 post-aggregation did not rescue CER1 expression, indicating that the hypoblast-like cells were not yet able to undergo anterior hypoblast-like differentiation upon the cessation of SOX17 overexpression. However, when doxycycline was removed at day 3 post-aggregation, we observed an increase in CER1 expression at day 4 post-aggregation and a respective increase in TBXT/BRY expression at day 6, similar to structures generated with GATA6 inducible cells only.

Response to Reviewers Figure 1.5: Prolonged SOX17 overexpression is antagonistic to anterior hypoblast-like differentiation.

(A) Efficiency of tri-lineage organized structure recovery from embryoids derived using distinct induction regimes for hypoblast-like cell induction or by removing doxycycline at day 1 or day 3 of embryoid formation. N=2 independent experiments for each. (B) Representative examples and quantification of inducible human embryoids and quantification showing CER1-positive cells surrounding the epiblast-like domain generated using induced GATA6 (iG6) but not induced GATA6-SOX17 (iG6-S17) dual induction of hypoblast-like cells. CER1 expression is also observed if doxycycline is withdrawn at day 3 post-aggregation but not at day 1. n=247 structures from 7 independent experiments. (C) Representative examples of inducible human embryoids and quantification at day 6 post-aggregation showing an increase in BRACHYURY/TBXT expression in conditions where CER1 was observable at day 4. n=101 structures from 6 independent experiments. Scale bars = 100µm. Statistics (B-C): RM Two-way ANOVA with Holm-Sidak's multiple comparisons test. ****p<0.0001.

L 404-are RSeT cells naive cells? What are the authors trying to claim here?

We have rephrased the text to describe the state of RSeT cells more appropriately (L467-469). We agree with the referee that these cells are an intermediate cell type between naïve and primed.

<< These results contrast with embryoids derived from consistent GATA6-SOX17 induction, which predominantly generate amnion, suggesting that the intermediate peri-implantation-like RSeT hESCs capacitate over time, as reported for pre-implantation-like naïve hESCs in culture¹⁶. >>

Referee #2:

I was asked to conduct an ethics review of the manuscript “Transgene directed induction of a stem cell-derived human embryo model.”

The ethics statement within the methods section notes appropriate approvals were obtained from UK authorities for use of human embryonic stem cells and human embryos in research. Additional approval for the project was obtained from the University of Cambridge’s Human Biology Research Ethics Committee. However, the several authors are located (or co-located) in the United States, but no US approval was listed. The author contribution statements note that the work on embryoids and using human embryos seems to have been conducted in the UK. Therefore, I would request that the authors add a specific statement to the ethic session that the work was conducted in the UK to make this clear, if it is the case. I would also suggest the authors add a note that the research follows the 2021 International Society for Stem Cell Research (ISSCR) guidelines, since several co-authors are US-based.

Aside from these minor edits, this manuscript and the research conducted seem adequate for publication, from an ethical perspective.

We thank the reviewer for their assessment of our ethics approvals and our work. We have now specified that all work with embryos and embryoids was conducted in the UK.

Referee #3:

In this manuscript, the authors present an invitro model of human post-gastrulating embryo using transgene induction. Hypoblast-like cells are derived using induction of Gata6 and Sox17 in ES cells, tropho-blast like cells are derived using induction of Gata3 and Sox2, and the transduced cells are cultured with ES cells in a system pioneered by the authors that successfully recapitulated mouse embryogenesis invitro. The authors first verified the successful induction of trans genes and performed single-cell multiome experiments of individual lines to verify the reprogramming and then undertook extensive optimization of culture conditions for co-culture. The co-cultured embryos were profiled using single-cell multiomics following which cell-types were annotated using known genes, and signatures from single-cell profiles of other primates. Cell types resulting from the invitro embryos include embryonic, extra-embryonic, amniotic cells and primordial germ cells. Next, single-cell data was used to identify a role for BMP specification in these embryos and finally the authors demonstrate that Sox17 induction, a transgene used opposes the specification of anterior hypoblast.

The central contribution of the manuscript is a toolkit with an ability to mimic process of post gastrulation human embryos, invitro. While this is undoubtedly of importance, the manuscript suffers from several major issues which preclude a reliable interpretation of presented results in many scenarios including lack of detail of computational analysis, unclear importance of the induced trophoblast cells, and lack of evidence beyond marker expression for identification of cell types such as PGCs. The results presented by the authors denote sufficiently justify the claimed modular nature of the model. The comments are detailed below.

We thank the reviewer for their thoughtful comments on our manuscript and respond to their specific points below.

1. General computational analysis of the data

The methods description of the manuscript is lacking sufficient detail to critically assess many components of the analysis and for reproducing the analysis in the data. Even though the authors provide the code used, the details should be clearly described, choices justified. General examples are listed below:

a. Integration of public single-cell data: This contains a wide variety of datasets across single-cell platforms and organisms. The authors should present a detailed description and visuals of the integration and discuss the quality of integration. Does the integration sufficiently correct for the batch effects? Does the FindMarkers approach use the appropriate statistics to identify regulons since it was designed for gene expression? The details presented currently make it hard to assess whether the TFs chosen for induction are data-driven.

We are grateful for this suggestion and now provide a more detailed description of the integration and discuss its quality. The integration we describe in this manuscript is used as the basis to generate gene networks underlying the three major lineages of the peri-implantation embryo: the epiblast, hypoblast, and trophoblast. We then utilized this dataset to guide our choice of transgenes to induce extraembryonic identity from hESCs. We now include analysis of the post-integration datasets (included now as Extended Data Fig. 1a-e) and provide details in the Methods, in addition to the code used in the GitHub repository:

<< Previously published data from Yan et al., Blakely et al., Petropolous et al., Zhou et al., and Xiang et al., were realigned to the hg38 human genome using kallisto or kb-bustools^{17,18}.

Datasets which were not sequenced with UMI-based technologies were normalized using quasinorm to quasi-umis¹⁹. Using SCTransform-based integration, datasets were combined to generate a single-cell RNA-seq dataset of human embryos spanning zygote to day 14 post-fertilization^{4,7,20-23}. Cells were clustered and identities assigned based on previous annotations and canonical marker expression. The dataset showed good overlap of datasets with separation of cell types and some temporal resolution. SCENIC was used with default settings in R, and the AUC-regulon table used to generate a new assay in the Seurat object. Using this assay, the epiblast, hypoblast, and trophoblast lineages were then compared using Seurat's FindMarkers function to implement a Wilcoxon ranked test with Bonferroni correction to identify pairwise predicted differentially active regulons. Regulons that were enriched across both relevant comparisons (e.g. hypoblast versus epiblast; hypoblast versus trophoblast) were used as enriched active transcription factors for subsequent analyses (e.g. in the hypoblast). These factors were then plotted in relation to each other in Cytoscape. >>

Response to Reviewers Figure 3.1: Integration of Published Peri-Implantation Human Single Cell RNA-Sequencing Datasets.

(A-C) Uniform Manifold Projection and Approximation (UMAP) colored by original publication^{4,7,21-23} (A), age of embryo in days post-fertilization (d.p.f.) (B), and cell type (C). (D) Gene expression of canonical lineage markers plotted on UMAPs. (E) Violin plots of key gene expression separated by cell type. PreLin = Pre-lineage commitment (e.g. days 0-4 post-fertilization), EPI = Epiblast, HYPO = Hypoblast, TE/TrB = Trophectoderm/Trophoblast, STB = Syncytiotrophoblast, EVT = Extravillous trophoblast.

Regarding the use of the FindMarkers function with regulons, the implementation specifically employs a Wilcoxon ranked test with Bonferroni correction as well as regulon data. More complex statistical tests designed specifically for gene expression (e.g. those with negative binomial distribution assumptions) were not utilized.

Taken together, we hope that the referee will agree that the integration approach utilized to generate this combined dataset allows us to delineate the major lineages of the embryo without splitting the data based on technology or culture condition.

b. No details are provided on the logistic regression approach used in multiple instances? How

were the genes chosen? What are the training, test and validation sets? What datasets were used? Without these details, the logistic regression results cannot be reliably interpreted

We thank the reviewer for the chance to clarify our approach and apologize that this was not clear. In revising the manuscript, we have updated the logistic regressions to ensure training and test datasets as well as the gene sets used are standardized. In all cases, previously published data is used as training data, and the data from the cells and embryoids produced in this work is the test data. In all cases, the differentially expressed genes from our datasets (cells and embryoids) are used as the gene set. This has been further clarified in the Methods section:

<< Upon cell type assignment and processing for the cell lines sequenced, a previously reported and validated logistic regression framework was applied to project cell line data onto published single cell data and to project published cluster annotations (e.g. training data) onto post-implantation embryo model clusters (e.g. test data), resulting in a quantitative measure of predicted similarities²⁴. Here, only differentially expressed genes (produced using Seurat's FindAllMarkers function on course cell assignments (which collapsed amnion and mesodermal clusters) were used. >>

Code used to generate the logistic regressions and a description of its implementation can be found in Young et al., 2018 (DOI: 10.1126/science.aat1699)²⁴. We have also previously used and published with this approach in Molè et al., 2021 (DOI: 10.1038/s41467-021-23758-w)⁴. We now include the updated logistic regression plots below and in Fig. 1d and Extended Data Figs. 4b and 6a-b.

Response to Reviewers Figure 3.2: Updated and additional logistic regression comparisons of embryoid-derived RNA-sequencing clusters to published datasets.

(A) Comparison of wildtype RSeT hESC (RSeT WT), inducible GATA6-SOX17 cells after 3 days of induction (Day 3 G6-S17), and inducible GATA3-AP2y cells after 3 days of induction (Day 3 G3-AP2y) to published human embryo, blastoid¹, and directed differentiation datasets³. (B) Comparison of inducible human embryoid clusters with published cynomolgus macaque²⁵⁻²⁷ and human embryo^{4,28} datasets spanning peri-implantation to gastrulation stages. (C) Comparison of inducible human embryoid clusters with published *in vitro* models including directed differentiation³, blastoids¹, and post-implantation amniotic sac embryoids (PASE)². In all cases, published data was used as training data and data produced in this manuscript was used as test data. The gene set used was differentially expressed genes identified in this manuscript (presented in Extended Data Table 2).

c. Cell type transfers for the newly generated data were performed using scmap. The authors provide details on how scmap was applied but additional discussion is needed to discuss any potential pitfalls: for example, how does scmap handle cell-types that are relevant to the dataset?

We thank the reviewer for this suggestion and have now added further discussion on this point in the methods section:

<< Multiple datasets were used to infer annotations with scmap as transcriptionally similar clusters (e.g. trophoblast and amnion) may map incorrectly if either population is not present in a certain dataset. >>

Specifically, we note that when we use scmap with datasets containing cell types with known gene expression overlap (e.g. trophoblast and amnion) we observe confident mapping to a single lineage (for example our cells map well to primate amnion but not to trophoblast). Conversely, in datasets that have limited cell assignments (e.g. our previous human *in vitro* cultured embryo dataset – Molè et al., 2021⁴) we observe cells being mapped to highly similar cell types (e.g. embryoid amnion cells mapping to ‘cytotrophoblast’). Alternatively, the Tyser et al., 2021²⁸ dataset contains several cell types, which do not map to any embryoid-derived cells (e.g. Axial Mesoderm, Erythro-Myeloid Progenitors, Erythroblasts, Myeloid Progenitors) and we see no evidence of these cells either in sequencing or by immunofluorescence, giving confidence that the thresholding in scmap is working correctly. However, while we feel that scmap allows us to assign cell types with less bias than a targeted marker gene approach, it is important to acknowledge the limitations of this approach. For this reason, we have chosen to include multiple primate, human, and stem cell model datasets to bolster our conclusions and to validate scmap-generated annotations with cardinal marker profiles.

d. QC plots of the single-cell data should be provided given the highly novel nature of the data

We agree and have added plots to show the final datasets’ characteristics including: UMI counts, number of Unique Features, number of peaks, TSS enrichment, percent mitochondrial reads, fragment distribution, and more in Extended Data Figs. 4a and 5c. We have included the relevant panels below for convenience:

Response to Reviewers Figure 3.3: Quality control plots for 10x multiome sequencing.

(A-B) Plots for the number of counts (UMI; RNA-sequencing), percentage of mitochondrial reads in RNA-sequencing, number of peaks called (ATAC-sequencing), fragment distributions of ATAC-sequencing, and the

mean transcription start site (TSS) enrichment score for individual cells used to generate inducible embryoids (A) and embryoids at day 4, 6, and 8 days post-aggregation (B).

Further information on data quality based on the CellRanger-ARC alignments is now presented in Extended Data Table 1.

2. Derivation of transduced cells and the tri-lineage embryo:

a. The authors need to discuss how critical it is for the transduced cells to have shut down the embryonic / pluripotent program given the target roles. In particular Oct4 is missing in most of the qRT-PCR / immunofluorescence experiments. Oct4 in my opinion is a necessary gene to include in this analysis. Do the transduced cells retain their identity in culture or are they plastic to revert back to embryonic cells? While this is demonstrated for the trophoblast-like cells using GFP, no such evidence is provided for the hypoblast-like cells.

We thank the reviewer for this thoughtful comment and agree that the downregulation of the pluripotency network is an important requirement for deriving extraembryonic lineages.

We have included immunofluorescence OCT4 quantification after 3 days of doxycycline induction in Extended Data Fig. 2d (GATA3-AP2 γ induction for 72 hours). In addition, *POU5F1/OCT4* transcript expression is presented in Extended Data Fig. 4c (included below) and in the dotplot in Fig. 1e for all cell populations used to generate embryoids. OCT4 transcript expression is also included for embryoid cell types after aggregation in the dotplot in Fig. 3c, where it is enriched in the L-EPI cluster. Crucially, we also show the loss of OCT4 at the protein level in extraembryonic-like compartments of embryoids by day 4 (Figs. 2g and 4c-d; included below). Other nodes in the pluripotency network (e.g. SOX2, NANOG) are presented in the same panels and give confidence that the pluripotency network is downregulated after transgene induction, albeit at different rates in hypoblast- or trophoblast-like induction.

Response to Reviewers Figure 3.4: Expression of *POU5F1/OCT4* in cell sequencing and embryoids.

(A) Uniform manifold projection and approximation visualizations (UMAPs) showing the cell type samples sequenced and the expression of *POU5F1*. (B) Expression of *OCT4*, *GATA3*, and *GATA4* in a representative day 4 human embryoid. (C) Expression of *OCT4*, *GATA6*, and phosphorylated (p)SMAD1.5 or total SMAD2.3 in day 4 and 6 embryoids. Note the outer signal in the bottom day 6 embryoid is GFP background. Scale bars = 100µm.

Regarding reversion to an ESC-like state after doxycycline induction, we have now generated structures using Shef6-mKate2 hESCs. These cells make up the vast majority of the SOX2+ Epiblast-like domain (~71% of mKate2+ cells are SOX2+), and of those that are GATA6 positive we believe this reflects extraembryonic mesenchyme differentiation, as reflected by TBX20 expression at day 6, and a small amount of early extraembryonic mesenchyme cells detected in day 4 embryoids in our sequencing data (Extended Data Figs. 5f and 7g-h; included below). Unlabeled cells rarely, if ever, re-upregulate pluripotency markers, indicating that the GATA6-SOX17 induced cells do not (or very rarely) revert to an ESC-like state.

**Response to Reviewers Figure 3.5: Wildtype ESCs generate the inner epiblast-like domain and extraembryonic mesenchyme.**

(A) Immunofluorescence images showing the contribution and localization of Shef6-mKate2-positive ESCs, unlabeled inducible GATA6-SOX17 cells, and GFP-positive GATA3-AP2γ cells in day 4 embryoids. (B) Quantification of mKate2-positive cells and co-expression with SOX2 or GATA6. Most mKate2-positive cells are present in the SOX2-positive, epiblast-like domain, with some mKate2-positive cells also co-expressing GATA6 and being present in the interstitial domain in line with some early epiblast-like differentiation to extraembryonic mesenchyme. (C) Alluvial plot showing the distribution of cell types based on timepoint of embryoids from sequencing analysis. (D) mKate2-positive cells at day 6 post-aggregation generate both the inner domain and GATA6/TBX20-positive extraembryonic mesenchyme. Scale bars = 100µm.

b. The logistic regression results in 1D suggest that Day 3 G6-S17 are more similar to trophoblast rather than Day 3 G3-Ap cells, and that Day 3 G3-Ap are also similar to Epiblast cells? Is this an issue with signature, gene sets or does it reflect the underlying identity of the cells?

We apologize that the formatting of these plots made their interpretation difficult. In this case, these plots should only be read vertically. This is because the individual clusters shown on the y-axis have been projected onto the clusters of the x-axis (e.g. y-axis clusters are used as training data, and x-axis clusters are used as test data). Therefore, it is impossible to directly compare 'similarity' between the two populations. However, it is clear that day 3 induced GATA3-AP2γ cells are also similar to epiblast, and that the GATA6-SOX17 cells show high similarity with the trophoblast.

It is important to note that this is likely related to the gene list used here: differentially expressed genes between the *in vitro* cell lines. However, we believe it does reflect the underlying identity of the cells. For example, Day 3 GATA3-AP2 γ cells have not fully downregulated pluripotency markers, and particularly not to the extent of the Day 3 GATA6-SOX17 inducible cells, which can be seen in Extended Data Fig. 4c, and above in Response to Reviewer Figure 3.4. Throughout the manuscript, we have now made sure not to overstate the extent of 'reprogramming' that the overexpression of these factors has achieved.

Minor comments:

a. More details needed to be provide the choice of mixing ratios and how it relates the invivo embryo

We now include the ratio of epiblast/hypoblast/trophoblast cells at day 8 of human embryos cultured *in vitro* and discuss how this informed our decisions regarding initial seeding densities of the three cell types. We have added the following to the methods section:

<< At 8 d.p.f., *in vitro* cultured human embryos have 32:24:228 epiblast:hypoblast:trophoblast cells⁴. Importantly, however, many of the trophoblast cells are not in contact with the inner cell mass-derived tissues or are terminally differentiated. Additionally, we observed that in culture, inducible GATA6-SOX17 cells proliferate slower than the other two cell populations after doxycycline addition. Therefore, we utilized an initial seeding density with: (1) a total cell number similar to that used in mouse models that allowed us for successful cell sorting^{29,30}; (2) a ratio of cells that reflected the peri-implantation embryo; and (3) a reduced number of inducible GATA3-AP2 γ cells and an increased number of inducible GATA6-SOX17 cells. >>

b. Panel name is missing for "2H"

We thank the reviewer for catching this and have corrected it in the updated submission.

c. Day of collection is not specified for images in Fig. 2

We have now specified Day of image for structures in Fig. 2a. All immunofluorescence images for individual structures have the day of analysis noted in the top left, and graphs note the day of collection on the y-axis.

3. Role of G3-Ap cells:

a. The authors claim through the results in Fig. 2C, that there is a single layer of Gata3 positive cells surrounding the embryo - however it appears that the Gata6+ cells substantially outnumber the Gata3+ even on the boundary? A detailed analysis of this particularly critical given point 2b. Perhaps showing these genes on a separate panels will provide better clarity. Given the centrality of the claim, the authors need to generate an analysis pipeline to count the number of Gata6+ cells on the boundary (and interior) and provide statistical evidence rather than the observational evidence.

This is an excellent suggestion, and we now provide new quantification of cell number proportions in day 4 structures. We also provide Imaris-generated spot render models of the

structures presented in Fig. 2b (also provided below) which allows for easier interpretation of the spatial distribution of cells. The outermost GATA3-positive layer also expresses GATA6, unlike the interstitial GATA6-positive layer which does not express GATA3 (shown in Fig. 2b). The contribution of inducible GATA3-AP2 γ cells to the outermost layer of our embryoids can be observed using GFP fluorescence as GFP is directly joined to GATA3 in the inducible construct used to generate this line (see Fig. 2g; provided below).

Response to Reviewers Figure 3.6: Organization and quantification of tri-lineage embryoids at day 4 post-aggregation.

(A) Embryoids form with an inner SOX2-positive domain, interstitial GATA6-positive domain, and an outer layer of GATA3-positive cells. (B) Quantification of cell type proportions in day 4 embryoids. (C) GATA3-AP2 γ inducible cells, marked by GFP, surround the periphery of day 4 embryoids. Scale bars = 100 μ m.

b. Further, the authors also mention that G3-Ap are most likely not bonafide trophoblast cells. Given this and observations of low numbers of G3-Ap cells on the embryo boundary and their lack of signal in the logistic regression, raises the question whether G3-Ap cells are necessary for the embryo formation through this culture system.

We agree with the reviewer that the points mentioned above raise the question of whether GATA3-AP2 γ cells are necessary for embryoid formation, and we, therefore, carried out experiments to determine this. To test this, we attempted to generate embryoids with only wildtype RSeT hESCs and GATA6-SOX17 inducible cells (i.e. without inducible GATA3-AP2 γ) and found that embryoids do not form without inducible GATA3-AP2 γ (Extended Data Fig. 8e). To investigate this further, we utilized CellPhoneDB to analyze signaling within our embryoids and inducible cells. This analysis indicated that GATA3-AP2 γ inducible cells are likely the initial source of BMP at aggregation (Extended Data Fig. 8d). We hypothesize that these cells initially support aggregation and survival partially through acting as a BMP source. Addition of LDN193189 (BMP signaling antagonist) during aggregation (between days 0-2 post aggregation) blocked embryoid organization and formation. Together, these experiments highlight that the inducible GATA3-AP2 γ cells are crucial to embryoid formation and act, at least partially, via secretion of BMP to the other two populations.

Response to Reviewers Figure 3.7: GATA3-AP2 γ inducible cells are required for embryoid formation.

(A) CellPhoneDB analysis showing potential receptor-ligand interactions between inducible cell populations and wildtype ESCs, which may influence embryoid survival and/or organization. CellPhoneDB predicts the BMP4 source to RSeT hESCs to be the GATA3-AP2 γ inducible population. Colors for clusters on the x-axis match the receptor and/or ligand in the y-axis. (B) Violin plots showing gene expression of BMP ligands and receptors in the cells used to generate inducible human embryoids, confirming expression of BMP ligands in the GATA3-AP2 γ inducible population. (C) Inducible human embryoids do not form if induced GATA3-AP2 γ cells are excluded or if LDN193189 is added between days 0-2. (D) Quantification of embryoid formation efficiency from C. (E) Quantification of embryoid size after LDN193189 addition between days 0-2. Scale bars = 100 μm . Statistics: (E) Unpaired t-test. **p<0.01, ****p<0.0001.

4. Generation of primordial germ cells: The evidence provided for generation of bonafide PGCs are not convincing:

a. The signature scores in Fig. 3E show a surprisingly small range and the difference between PGC and non-PGC scores are rather minimal. Do the scores show a similar range in the dataset derived or is there more contrast?

To bolster evidence for PGCLC formation within our embryoids, we have now examined co-expression of several additional markers. This data is presented in a nebuloza plot in Fig. 3f (provided below) and shows the co-expression density of *PRDM1* (BLIMP1) and *NANOS3* in addition to the markers used to quantify PGCLCs by immunofluorescence (*TFAP2C/AP2 γ* , *SOX17* and *NANOG*). This analysis shows a very similar, though slightly restricted, distribution of positive cells to the module score and gives further evidence that PGCLCs are captured in our sequencing dataset.

We utilized the module scoring function in-built in the Seurat package (AddModuleScore; <https://satijalab.org/seurat/reference/addmodulescore>). AddModuleScore functions by taking the expression of the genes of interest and subtracting an aggregated expression of control feature sets that are selected from the same bins as genes of interest. Therefore, positive final values indicate an enrichment of the input gene list/feature set compared to background. However, given that the normalization is within dataset and is based on aggregated

subtraction, comparisons of absolute values and contrasts across datasets are not valid given potential differences in sequencing depth and coverage. The developers of Seurat have specified previously that the scores themselves are unitless (<https://github.com/satijalab/seurat/issues/522>), therefore it is difficult to directly compare ranges across datasets.

Response to Reviewers Figure 3.8: Expression pattern of primordial germ cell-associated genes.

(A) Module scoring for primordial germ cell marker genes (Jo et al., 2022)³¹. (B) Nebulosa plot visualizing joint expression density of key primordial germ cell genes (PRDM1, NANOS3, TFAP2C, SOX17, and NANOG) in inducible human embryoids.

b. How were the cluster of cells termed "PGCs" in Fig. 3F determined? How were the genes chosen?

We annotated cells as "PGCs" within the 98th percentile of the module score, and the genes plotted are those used for the module scoring (gene list from Jo et al., eLife 2022 (DOI: 10.7554/eLife.72811)³¹. We have now specified this in the text and added (L316-318):

<< We labeled cells in the 98th percentile of primordial germ cell-like cell gene expression³¹ module scores as putative primordial germ cell-like cells ("PGC").>>

5. BMP signaling response:

a. All the observations about trends and dynamics of gene expression in 4A are based on observations on umap and is highly qualitative. The authors should provide better estimates of these dynamics by fitting these trends along a pseudo-time axis (eg: multivelo latent time) and provide more concrete evidence for the dynamics described in the manuscript.

We thank the reviewer for this suggestion. We have now used a combination of the multivelo latent time with the SwitchDE package to fit the trends (both for gene expression and chromVAR-scored motif accessibility) along a pseudo-temporal (latent time) axis (Extended Data Fig. 8a-b). These data provide further evidence that the BMP response genes *ID1-4* are expressed within embryoids across latent time. Further, plotting of the SMAD5 motif accessibility demonstrates that this motif remains enriched over latent time in embryoids, in contrast to the SMAD2::SMAD3::SMAD4 motif, which decreases over latent time. Together, these data demonstrate that BMP signaling remains high in embryoids, while NODAL signaling decreases.

Response to Reviewers Figure 3.9: *ID* gene expression and SMAD motif accessibility over latent time.

(A) *ID*1-4 expression over latent time (calculated using multivelo). (B) Motif accessibility score of SMAD5 (indicative of BMP signaling) and SMAD2::SMAD3::SMAD4 (indicative of NODAL signaling) over latent time (calculated using multivelo)

b. While it is interesting to investigate the trends of *ID*1-4, are the gene signatures scores of BMP signaling response genes consistent with these observations?

We thank this reviewer for this suggestion. We have not been able to generate a consensus set of BMP response genes given its wide function across tissues. However, if the reviewer has a reference with such a list in mind, we would be happy to examine this. The *ID* factors are one of the most consistent response genes, which is why they were utilized in our study (Hollnagel et al., 1998)³². To support the role of BMP in driving differentiation of the epiblast-like compartment, we have also directly quantified SMAD1.5 phosphorylation (an indicator of active BMP signaling) in embryoids at day 4 and day 6 post-aggregation. This data shows BMP is enriched in the epiblast-like compartment and that inhibition of BMP delays amnion differentiation. We have also added additional functional data on PGC specification (Fig. 4g-h; included below) in response to BMP signaling modification.

Response to Reviewers Figure 3.10: BMP signaling drives differentiation of the epiblast-like domain in embryoids.

(A) Representative immunofluorescence staining and quantification of representative inducible human embryoids at days 4 (n=120 cells) and 6 (n=80 cells) showing heightened pSMAD1.5 in the epiblast-like domain. N=3 independent experiments. (B) Representative immunofluorescence staining and quantification of SMAD2.3 in representative inducible human embryoids at days 4 (n=80 cells) and 6 (n=80 cells). N=2 independent experiments. (C) Inhibition of BMP signaling blocks exit from pluripotency and upregulation of amnion markers AP2α and CDX2. (D) Quantification of the percentage of inner domains expressing SOX2 and CDX2 at day 4 (n=1971 structures from 5 independent experiments). (E) Inhibition of BMP reduces the number of primordial germ cell-like cells in embryoids. (F) Quantification of the number of SOX17/NANOG/AP2γ triple-positive primordial germ cell-like cells (PGCLC) at day 4 (n=136 embryoids from 6 independent experiments). Scale bars = 100μm. Statistics: (A-B) Unpaired T-Test, (D) RM two-way ANOVA with Holm-Sidak's multiple comparisons test. (F) One-way ANOVA with Holm-Sidak's multiple comparisons test *p<0.05, **p<0.01, ***p<0.001, ****p<0.0001.

c. Similarly to (a), the TF signals in 4B are purely qualitative inferred using observations on umap. The insilico-ChIP tool (<https://doi.org/10.1101/2022.06.15.496239>) might be more suitable for inferring TF signals and dynamics should be demonstrated by fitting trends along pseudo-latent time.

We thank the reviewer for this suggestion. We have now used a combination of the multivelo latent time with the SwitchDE package to fit the trends along a pseudo-temporal axis. Similar to the insilico-ChIP tool cited we used chromvar to score motif accessibility - a key component of the tool referenced above. We have fit this motif accessibility score over latent time as the reviewer suggests (Extended Data Fig. 8b; Response to Reviewers Figure 3.9B above). At this time, the insilico-ChIP code that is available is relevant to reproduce the data reported in the preprint, but not readily adaptable for wider use. Therefore, we have chosen to not utilize it at this time.

6. Sox17 effect on hypoblast and modularity of tool: The evidence provided to demonstrate that the invitro model is modular is not definitive.

a. The viability, efficiency and recover of cell-types through induction of Gata6 is not described in detail to justify the claim of modularity.

We now provide efficiency quantifications for structures generated with GATA6 and SOX17 single inducible hypoblast-like cells. Indeed, we find that these structures are capable of self-organizing similarly to embryoids characterized earlier in the manuscript, but at lower efficiency (Extended Data Fig. 9a; included below), and with the differences in cell subpopulations and differentiation events emphasized in Fig. 5.

Regarding evidence for the effect of SOX17 overexpression on CER1, we now add significant new experiments with: (1) SOX17 single induction only, which demonstrate that these embryoids poorly upregulate CER1; and (2) removal of doxycycline induction at either day 1 or day 3 post-aggregation for GATA6-SOX17 inducible cell-containing structures, showing that removal of doxycycline at day 3 is sufficient to upregulate CER1 expression. Together, this data supports the repressive role of SOX17 on CER1 expression and indicates that embryoids at day 1 post-aggregation are not yet competent to generate CER1-positive anterior hypoblast-like cells in the absence of doxycycline, likely because they are not sufficiently specified at this stage.

Response to Reviewers Figure 3.11: Prolonged SOX17 overexpression is antagonistic to anterior hypoblast-like differentiation.

(A) Efficiency of tri-lineage organized structure recovery from embryoids derived using distinct induction regimes for hypoblast-like cell induction or by removing doxycycline at day 1 or day 3 of embryoid formation. N=2 independent experiments for each. (B) Representative examples and quantification of inducible human embryoids and quantification showing CER1-positive cells surrounding the epiblast-like domain generated using induced GATA6 (iG6) but not induced GATA6-SOX17 (iG6-S17) dual induction of hypoblast-like cells. CER1 expression is also observed if doxycycline is withdrawn at day 3 post-aggregation but not at day 1. n=247 structures from 7 independent experiments. Scale bars = 100µm. Statistics (B): RM two-way ANOVA with Holm-Sidak's multiple comparisons test. ****p<0.0001.

b. The details of reanalysis in Fig. 5B in the Methods does not include several details such as how were cells chosen, what is the full umap, what are the different cell types presented in the umap etc.

We now include a full UMAP with cell types annotated and subclusters for the hypoblast as in Molè et al., 2021⁴. The updated panel is included in the following comment.

c. Cer1 and Sox17 coregulation statistics are not presented and the significance does not seem to be assessed.

The statistical analysis has previously been presented in Molè et al., 2021⁴ (Supplemental Data File 8). This has now been specified in the text (L403-406):

<< Reanalysis of our previously published 10x single-cell RNA sequencing data from post-implantation human embryos cultured *in vitro* revealed that SOX17 regulon activity was significantly enriched in the *CER1*-negative hypoblast subcluster (**Fig. 5b**, Supplementary Data Table 8 of Molè et al., 2021)⁴. >>

Response to Reviewers Figure 3.12: Reanalysis of published human anterior hypoblast in *in vitro* cultured human embryos.

(A) Analysis of GATA6 and SOX17 regulon activity scored by SCENIC and SOX17 and CER1 co-expression in the hypoblast of post-implantation human embryos (9-11 days post-fertilization). Data from Molè et al. 2021.

7. The authors use known cell-type markers to infer the derivation of different cell-types. For example, *Hand1*, *Foxf1*, *Tbx20* - please add relevant citations and discuss the reliability of these markers for cell-type identification.

We thank the reviewer for the chance to add these citations. We now include references to human and nonhuman primate datasets where these markers have been shown to be expressed in the relevant clusters (e.g. *HAND1*^{27,33}, *VTCN1*^{27,33}, *ISL1*²⁷, *FOXF1*^{3,27}, *TBX20*³⁴). It is important to note that our cell type identification does not rely on these markers only. Rather, we utilize scmap to project published cluster annotations onto our dataset, and this approach takes a holistic view of transcriptomic similarity. After applying scmap using 3 separate cynomolgus macaque datasets and 2 human embryo datasets, we then performed confirmatory checks of canonical markers and finalized cluster identities. Then, we applied a logistic regression framework²⁴ to back-compare our identified clusters to both the cynomolgus macaque and human embryo datasets, and now to 3 additional *in vitro* hESC-derived models. The use of only a handful of markers for cell type identification has a drawback in that individual markers may be expressed, but that may not reflect other markers that are regulated under different mechanisms. However, we circumvent this risk by applying several approaches for cluster identification/comparisons. We have included violin plots from a non-human primate dataset²⁷ for canonical markers including *HAND1*, *FOXF1*, *TBX20*, *VTCN1*, *ISL1*, *SOX2* below:

Response to Reviewers Figure 3.13: Expression of canonical markers in Yang et al., 2021 cynomolgus macaque dataset²⁷.

(A) Violin plots showing clear expression patterns of HAND1, VTCN1, ISL1, FOXF1, and TBX20 in cynomolgus macaque embryos at post-implantation stages.

Referee #4:

This manuscript by Weatherbee et al. reports the establishment of an pluripotent stem cell-derived model of early human embryo development achieved via aggregation of both extraembryonic and embryonic cellular components in vitro. The two distinct extraembryonic cellular components, putative hypoblast- and throphoblast-like cells are generated by direct reprogramming of intermediate pluripotent stem cells via the overexpression of specific transcription factors, GATA6 and SOX17 for the hypoblast and GATA3 and AP2γ for the trophoblast. The resulting aggregates and embryo-like structures are obtained with an efficiency of 23% and recapitulate several morphological and molecular hallmarks of post-implantation human embryo development including the presence of a central SOX2 positive cell population surrounded by two concentric layers of GATA6 and GATA3 positive cells. This model shows furthermore the presence of extraembryonic mesenchyme, amnion and PGCs cells.

Overall, it is a well written and timely manuscript which focuses on the in vitro recapitulation of peri-implantation human development. The obtained data and shown results appear to be largely of good/acceptable quality but the meaning and potential usefulness of this model is unfortunately overhyped and exaggerated by the authors. This is also reflected by the misleading title of the manuscript, which implies that the authors have succeeded to establish a proper and complete model of the human embryo.

This is by no means the case. This “embryoid” model system shows and recapitulates some aspects of embryonic development, especially cellular differentiation but the overall morphological characteristics of an actual post-implantation human or primate embryo are not recapitulated and critical cell types such as bona fide throphoblasts are missing.

We thank the reviewer for their thoughtful comments on our manuscript. We agree that our model does not recapitulate all developmental aspects of the *in vivo* primate embryo, which we now state with increased clarity throughout the manuscript.

We have also adjusted our title to << A post-implantation human embryo model derived from pluripotent stem cells >> to more accurately reflect our data.

Major Concerns and Comments to the Authors:

One of the core issues of this manuscript lies within the way the different extraembryonic cell lines are initially generated and ultimately assessed. Based on computational analysis the authors identified several gene candidates for the direct specification of pluripotent stem cells into hypoblast cells: GATA4, GATA6, SOX17, FOXA2 and trophoblast cells: GATA2, GATA3, NR2F2, TFAP2C. From this shortlist of transcription factors they then selected two genes as candidates for direct reprogramming of human PSCs into hypoblast or trophectoderm cells and assessed the efficiency of such induction starting from cells at three different states of pluripotency, i.e. naïve, intermediate peri-implantation and primed pluripotency states. Transcripts were expressed in these cells for three (3) days via doxycycline induction and efficiency of conversion & differentiation was evaluated using RT-qPCR.

(1) The outcome and efficiency of these experiments seems to be very variable depending of the pluripotent state of the initial cells. Could the authors elaborate on this? Can we imagine

that the transcription factors used are not sufficient to rewire the differentiation/reprogramming barriers preexisting in the different cell types used in this study?

We agree with the referee that the pluripotent state of the initial cells is very important for the outcome and have noted it in the discussion as follows (L490-505):

<< Initial pluripotent state is a crucial factor in induction of downstream gene regulatory networks, and we provide evidence that embryoids form efficiently using peri-implantation stage RSeT hESCs but not more naïve PXGL hESCs or primed hESCs. However, given induction of trophoblast gene networks appears more robust in naïve hESCs, the use of discordant pluripotent states (e.g. induction of GATA3 and/or AP2 γ in naïve PXGL hESCs combined with the other two cell types from a starting RSeT condition) presents a promising avenue to generate embryoids better recapitulating the embryo. The ability of hESC to generate extraembryonic fates should also reflect the differential timing of trophectoderm (16-32 cell stage) and hypoblast (early blastocyst) specification during embryonic development. Likewise, different combinations of transcription factors may be necessary for lineage specification from different starting states. Of note, however, PXGL hESCs require extensive capacitation *in vitro* to give rise to germ layers¹⁶ and blastoid models generated from naïve hESCs fail to model post-implantation stages robustly^{1,35,36}, in contrast to cynomolgus macaque blastoids^{8,37,38}. Similarly, primed ESCs appear to give rise to amnion-like cells in trophoblast differentiation conditions, in contrast to naïve ESCs^{8,37,38}. Further work interrogating the epigenetic landscape and binding sites of these factors might be able to improve strategies to generate *bona fide* extraembryonic cells. >>

(2) The generation of cell lines was done using the piggyBac transposon system. Different clones were selected using single colony manual picking which does not exclude heterogeneity in the copy numbers and levels of expression of the introduced transgenes even between different cells of a clone. This raises the question of the relevance of the level of expression of those transgenes for obtaining cells with the desired identity, i.e. hypoblast or trophoblast. Was there any quality control of these cells? The authors need to provide info/data on heterogeneity, copy numbers, integration sites, levels of expression of the different transgenes compared to endogenous expression levels in a real embryo. How many cell lines/clones were used for the initial characterization and for the actual experiments? Did the authors observe any differences between these clones for both extra embryonic lineages? What was the variability and reproducibility of outcomes between clones when using same conditions?

We now describe in detail the selection of clones for this study. Individual clones do indeed show heterogeneity in induction levels, likely due to integration number and site differences. In all cases, we selected clones that showed robust upregulation of the transgenes (e.g. *GATA6* and *SOX17*), as well as upregulation of another gene important in the specific gene program (e.g. *GATA4* for hypoblast-like induction) after addition of doxycycline for 72 hours in basal N2B27 media. Upregulation was assessed by qPCR and this data is presented below (Response to Reviewers Figure 4.1; included in the manuscript as Extended Data Fig. 1h).

Response to Reviewers Figure 4.1: Validation of transgene expression by RT-qPCR in individual picked clones.

(A) Doxycycline inducible constructs were inserted in Shef6 hESC using piggybac transposase (inducible GATA6, SOX17, GATA6-SOX17, GATA3, AP2 γ and GATA3-AP2 γ). Colonies were manually isolated after single cell plating and propagated. Appropriate transgene expression after doxycycline addition was validated by RT-qPCR after 72 hours of doxycycline addition in basal N2B27 conditions. Selected clones used in the manuscript are highlighted in green.

We have also included the following text in the methods section to clarify this point:

<< Clones were generated by manually picking single colonies under a dissecting microscope. Transgene activation was triggered by the addition of 1 μ g/mL doxycycline hyclate (D9891, Sigma). To select clones for downstream experiments, isolated colonies that survived manual picking were induced for 72 hours and cell pellets were collected for qPCR or stained for immunofluorescent analysis (note: this was performed in primed hESCs). Transgene expression and another key lineage marker were assessed for changes in expression compared to uninduced controls. Clones with robust transgene upregulation and downstream upregulation of an uninduced lineage marker were selected for subsequent experimentation (e.g. 1-2 clones per transgenic line). >>

Clones varied in reproducibility for generating our human post-implantation embryo model. However, we have not rigorously assessed this and believe it is outside the scope of the manuscript. Importantly, the use of an independent cell line (RUES2) and the ability to generate inducible embryoids from this line demonstrates that this process is robust to small differences in transgene expression levels. Data from RUES2-derived embryoids are included in Extended Data Figs. 5a and 7e-f; provided below.

Response to Reviewers Figure 4.2: Generation of Inducible Human Embryoids from an independent cell line (RUES2).

(A) RUES2-derived embryoids exhibit similar tri-lineage organized structure as Shef6-derived embryoids with an inner SOX2-positive domain, an outer GATA3-positive cells, and an interstitial GATA6-positive compartment. (B) Quantification of embryoid formation efficiency at day 4 post-aggregation for RUES2-derived embryoids. (C) RUES2-derived embryoids generate ISL1-positive amnion-like cells and GATA6/TBX20-positive extraembryonic mesenchyme-like cells at day 6 post-aggregation. (D) RUES2-derived embryoids generate SOX17/NANOG/AP2 γ triple-positive primordial germ cell-like cells. Scalebars = 100 μ m.

As with most studies driving overexpression of factors *in vitro*, the levels of overexpression here are far above those of the actual embryo, at least at the RNA level.

(3) Reference 42 suggests that GATA6 expression in hESCs (primed) results in iXEN-like cells with important heterogeneity in hypoblast related protein expression and implies that timing/conditions need to be adjusted to improve differentiation outcomes. Reference 43 indicates that SOX17 expression in hESCs (primed) orients cells toward a mesendoderm identity whereas another paper from Janet Rossant and colleagues addressing a similar topic in murine ESCs (naïve) (McDonald et al. 2014) suggests that transient expression of SOX17 for only 48h is sufficient to efficiently convert mESCs to XEN cells.

Wamaitha et al. (2015) and Séguin et al. (2008) utilize similar approaches to us to drive endodermal reprogramming from hESC. However, one crucial difference is the initial primed pluripotent state of the hESC in both studies. Therefore, the finding that SOX17 induction drives mesendodermal fates cannot be directly translated to SOX17 induction in more naïve hESCs. Indeed, SOX17, like GATA6, is expressed early in the hypoblast, supporting a role for this gene in driving endodermal/hypoblast identity, and as noted in the text, is included in our putative gene regulatory network in Extended Data Fig. 1f (provided below). In contrast, SOX7 is not expressed at these stages and is therefore not included in our putative network, indicating that while SOX7 may be able to drive extraembryonic endoderm identity from primed

cells this is not developmentally relevant. In addition, we have previously shown that the same directed differentiation protocol (Activin/Chiron/LIF) applied to PXGL, RSeT, or primed cells results in cells more closely correlated to pre-implantation hypoblast, post-implantation hypoblast, or definitive endoderm, respectively¹³, highlighting the distinct response to stimuli in each pluripotent state.

Response to Reviewers Figure 4.3: Putative gene regulatory network for epiblast (red), hypoblast (blue), and trophoblast (yellow). Inferred epiblast, hypoblast, and trophoblast gene regulatory network generated by SCENIC during peri-implantation human embryo development. Single cell sequencing datasets spanning peri-implantation were integrated, and their transcription factor activity inferred by SCENIC (see Response to Reviewers Figure 3.1 for details and visualization of integration). Transcription factors with significantly enriched AUC scores in the three major lineages by pairwise Wilcoxon ranked tests with Bonferroni correction are plotted relative to each other. Candidate factors are in bold red.

Similarly, regarding trophoctoderm induction, reference 44 used as the basis for the selection of GATA3 and TFAP2C, clearly indicates that GATA3 is important but does not support a role for TFAP2C in this process. TE reprogramming in human has still not been clarified and explained conclusively and is suggested to rely on a complex interaction of at least four genes (Papuchova et al., 2022). What was the scientific rationale for excluding the other two genes?

As noted in the manuscript, reference 44 indeed demonstrates the importance of GATA3 primarily, but also argues that “The chromatin occupancy analysis showed that GATA2, is bound by the three other TEtra factors and itself, pointing to its place at a step lower in the TEtra hierarchy compared with GATA3,” and shows in Figure 4C that GATA3 shares the highest number of co-binding sites with TFAP2C amongst the other 3 TEtra factors (>675 compared to ~650 for TFAP2A and <300 for GATA2). While TFAP2A could have also been a good option, TFAP2A has been purported as an early amnion marker, potentially important for its specification^{39,40}. Therefore, we adopted TFAP2C instead. While we could have overexpressed all four genes in combination, similar to our attempts to use minimal media, we strove to achieve the generation of a stem cell-derived model of the human embryo with minimal conditions. Importantly, work delineating the trophoblast gene network has been predominantly carried out in primed hESCs (e.g. Krendl et al., 2017⁴¹) and a smaller number of factors may be necessary to drive trophoblast-like differentiation from more naïve cells, similar to the explanation for hypoblast and definitive endoderm outlined above.

The authors furthermore report in the manuscript that the selected genes only work in combination with particular conditions and fail to generate bona fide trophoblast stem cells. What are these cells if not trophoblast cells? These cells seem to also express genes of other lineages which clearly indicates that the initially intended/postulated differentiation of PSCs into the two major extra embryonic lineages has not worked (at least not as claimed by the authors).

We agree with the reviewer that the GATA3-AP2γ inducible cells do not represent *bona fide* trophoblast cells. Throughout the manuscript we have attempted to make this clear, however, to ensure that the reader correctly interprets this important point we have added further clarification (L480-490):

<< However, the use of transcription factor overexpression to generate extraembryonic tissues may also lead to deficiencies in differentiation, as observed in models of mouse embryogenesis³⁰. For example, while GATA3 and AP2γ induction drive trophoblast-like gene programs in 2D, upon aggregation in our human embryo model, this cell population aberrantly upregulates endodermal markers (including SOX17 and GATA6). This is potentially due to their position at the outermost layer of the structure^{14,15}. Nevertheless, the GATA3-AP2γ inducible cells are required for successful embryoid organization and likely act as a crucial source of BMP at aggregation. Optimization to preserve the expression of trophoblast-like gene programs while preventing the aberrant upregulation of endodermal identity in these cells will be important to improve future iterations of this model. >>

Strikingly, in 2D, GATA3-AP2 γ induction drives, at least partially, a trophoblast-like gene program, without obvious aberrant expression of endodermal gene markers as seen in embryoids after aggregation. Therefore, we hypothesize that the failure to maintain this program in embryoids is likely due to the culture conditions and their position on the outside of the embryoids, which has been shown in embryoid bodies to drive an endodermal identity. However, it is important to note, that the GATA3-AP2 γ inducible cells are essential for successful embryoid formation and self-organization, indicating that the model remains useful to study tissue-tissue crosstalk. We hope the reviewer agrees that we explain the observed phenomenon sufficiently in the manuscript.

(4) The authors should provide further data and evidence for their experimental rationale and claim that these identified transcription factors either alone or in combination with each other can give rise to defined extra embryonic lineages and cell types. Other combinations which were excluded should be also tested and assessed accordingly.

We have added the clarifications listed above to ensure the manuscript does not exaggerate the extent of reprogramming toward extraembryonic lineages.

In addition, we have added new experiments with interspecies chimera that bolster our finding that induction of GATA3-AP2 γ or GATA6-SOX17 meaningfully shifts the identity of hESC toward extraembryonic cell fates and alters their contribution in the mouse blastocyst (Extended Data Fig. 4d-g and Response to Reviewer Fig 4.4 below).

Response to Reviewers Figure 4.4: Generation of interspecies chimeras with inducible cell lines after 3 days doxycycline induction in 2D.

(A) Immunofluorescence images of human cell-mouse embryo chimeras generated with control (wildtype ESC) or inducible GATA6-SOX17 (iG6-S17) at the late blastocyst stage demonstrating a shift of human cell identity after doxycycline induction. Human cells are marked by human nuclear antigen (HuNAg). (B) Quantification of the contribution of HuNAg-positive cells stained for SOX2 and SOX17. N=42 embryos. (C) Immunofluorescence images of human cell-mouse embryo chimeras generated with control (wildtype ESC) or inducible GATA3-AP2 γ (iG3-AP2 γ) at the late blastocyst stage demonstrating a shift of human cell identity from the SOX2-positive epiblast to the GATA3-positive trophoblast. (D) Quantification of the contribution of HuNAg-positive cells stained for SOX2 and GATA3. N=37 embryos. Induction was carried out from hESC in RSeT conditions. Scale bars = 100 μ m.

We agree that investigating other combinations of transgenes may be a fruitful avenue to better generate *bona fide* extraembryonic cells *in vitro* but believe this is outside the scope of this model and might lead, in the future, to another human embryo model. We have included this point in the discussion (L498-499):

<< Likewise, different combinations of transcription factors may be necessary for lineage specification from different starting states. >>

(5) The authors evaluate the efficiency of hypoblast or trophoblast differentiation at day three of doxycycline-driven transcript induction in all three conditions (naïve, intermediate, primed) and identify several differences in this regard. Could we hypothesize that the observed differences between cells induced from initially naïve or primed pluripotent stem cells could be the effect of the temporality of the reprogramming process, and that earlier or later time points might show different results and even overlap?

We agree that overexpression of factors in cells cultured for longer in each condition, particularly naïve or intermediate states, could yield similar final cell types. This is particularly true given that recent work shows that following reversion, intermediate state-cells slowly re-adopt features of primed pluripotency¹⁰. Indeed, we specifically use RSeT cells for human inducible embryoid generation within a short time window following conversion to RSeT conditions (2-6 passages) for this reason, as noted in the Methods.

However, there are several caveats that argue against a temporal convergence:

1. The response of naïve or primed hESCs to either trophoblast or hypoblast differentiation cues. For example, trophoblast differentiation in primed cells drives an amnion-like state and continued culture of either naïve hESC-derived trophoblasts or primed hESC-derived amnion does not show convergence over time^{37,38}. This data, together with our observations with hypoblast induction from distinct pluripotent states¹³, does not support a temporal convergence in cell identity.
2. Induction of transgenes drives differing downstream gene programs dependent on the initial pluripotent state.

(6) The authors decided to use ReSET ESCs expressing the gene combination GATA6-SOX17 to induce hypoblast cells. Under these conditions GATA4 expression is not detected both by qPCR and IF (Ext. Data Fig. 1 C and 2B). Could the authors elaborate on this?

Indeed, we did not see expression of *GATA4* after 3 days of GATA6-SOX17 and we now provide this additional analysis of gene expression from the sequencing data in Extended Data Fig. 4c. However, when we examine *GATA4* expression in embryoids, we do see expression (Figure 2g) indicating its upregulation may require longer induction than 3 days or be facilitated by transitioning to a multi-cell type aggregate or 3D environment. We have added the following to the manuscript (L235-239):

<< The hypoblast-like compartment expresses *GATA4* (**Fig. 2g**), despite GATA6-SOX17 inducible cells not expressing *GATA4* after 3 days of induction in 2D (**Extended Data Fig. 4c**), indicating either an improvement in extraembryonic identity acquisition upon aggregation of the three cell types in 3D or a longer temporal requirement for *GATA4* upregulation after GATA6-SOX17 induction. >>

Response to Reviewers Figure 4.5: Expression of *GATA4* in cells and embryoids.

(A) Uniform manifold approximation and projection (UMAP) colored by sample of origin (top) and expression of *GATA4* (bottom) in cells used to generate human embryoids (i.e. after 3 days of doxycycline induction in 2D). (B) *GATA4* is expressed in embryoids at day 4 post-aggregation. (C) UMAP colored by cell type (top) and expression of *GATA4* (bottom) in human embryoids showing high expression in hypoblast (HYPO/VE) and extraembryonic mesenchyme (EXMC) clusters. Scale bar = 100 μ m.

(7) In Ext. Data Fig. 1D and 2D panels for the naïve PXGL state are missing for iAy whereas they are present for the primed state mTeSR, while in the text it is clearly mentioned that primed cells die or lose transgene expression upon iAy expression. Were the panels switched by mistake or is the annotation inverted?

We could not successfully convert inducible AP2 γ hESCs to the naïve state in PXGL conditions, as noted in the Methods section. Variability in the ability of lines or clones to revert to the PXGL naïve culture conditions is a well-known phenomenon in resetting to naïve pluripotency. In addition, AP2 γ induction in mTeSR conditions led to significant cell death. However, we were able to recover cells for analysis in this condition and this data is included in Extended Data Fig. 2.

(8) Based on the result shown in Ext. Data Fig. 1D and 2D it seems that ReST iAy is better than ReST iG3-iAy. Why was this cell line not used instead?

While AP2 γ upregulates *GATA2* expression, it poorly induces *GATA3*, while upregulating the amnion marker *ISL1* in RSeT conditions (Extended Data Fig. 2d; modified version provided below). *GATA3* is crucial for establishing the trophoblast gene program in the human embryo⁴². Additionally, co-induction of *GATA3* and AP2 γ significantly decreases expression of pluripotency genes, including *OCT4* and *SOX2*; which is not particularly surprising given that AP2 γ is expressed in naïve human ESCs⁴³ (including in RSeT cultured ESCs), and

therefore overexpression of AP2 γ alone is likely not as antagonistic to the existing pluripotency network. For these reasons, we utilized the GATA3-AP2 γ double inducible cell line to generate trophoblast-like cells.

Response to Reviewers Figure 4.6: Immunofluorescence quantification of selected genes in GATA3, AP2 γ , GATA3-AP2 γ inducible RSeT cells.

(A) Z-score of relative fluorescence levels of proteins in inducible GATA3 (iG3), inducible AP2 γ (iAY), and inducible GATA3-AP2 γ (iG3-AY) cells that were cultured in RSeT conditions and then induced for 72 hours with doxycycline in basal media. Note that AP2 γ induction does not induce GATA3 expression.

(9) Did the authors compare the RNA seq data of the induced cells to previously reported XEN cells generated by overexpression of SOX17 (PMID:25373912) or GATA6 (PMID:26109048)?

Human SOX17 overexpressing cells generated in Seguin et al., 2008 (PMID:18682240)⁴⁴ or GATA6 overexpressing cells generated in Wamaitha et al., 2015 (PMID: 26109048) were assessed using microarray technology and robust comparisons to our 10x dataset are not possible given the technological differences. Please note that in PMID: 25373912 mouse ESCs were used. However, we have compared our induced cells to multiple *in vitro* cultured embryo and differentiated cell types, which we feel allows their better comparison to *bona fide* extraembryonic lineages (please see also our response to next comment below).

(10) For the comparison of protein expression between transgene induced hypoblast and throphoblast cells the authors used RSeT cells as a starting material. The authors should compare these with the cells obtained via naïve cell direct differentiation and reported previous models of blastoids.

We now provide sequencing-based comparisons of these cells, and our embryoids, with the blastoid data from Kagawa et al., 2022¹ and directed differentiation in Pham et al., 2022³ (Extended Data Fig. 4b). In terms of protein level expression comparison, we have directly compared induction from primed, RSeT and PXGL hESC (Extended Data Fig. 2). In addition, we provide comparisons of induction to directed differentiation protocols (Extended Data Fig. 3.); however, this data is only from RSeT hESC as we had determined these cells as the ‘best’ approach for the specific goals of the present manuscript.

Response to Reviewers Figure 4.7: Comparison of cell populations to published datasets.

(A) Comparison of wildtype RSeT hESC (RSeT WT), inducible GATA6-SOX17 cells after 3 days of induction (Day 3 G6-S17), and inducible GATA3-AP2 γ cells after 3 days of induction (Day 3 G3-AP2 γ) to published human embryo, blastoid¹, and directed differentiation datasets³. In all cases, published data was used as training data and data produced in this manuscript was used as test data. The gene set used was differentially expressed genes identified in this manuscript (presented in Extended Data Table 2).

(11) In Ext. Data Fig. 4 TFAP2C population seems to overlap with the NANOG/SOX2/POU5F1 population rather than with the GATA3 population, as one would normally expect. Moreover the same population seems to also overlap with the CDH1+ population and E-Cad staining in Figure 2G is restricted to the pluripotent population. Is the TFAP2C population in the structure equivalent to or part of the epiblast and potentially not related to the putative trophoblast?

We thank the reviewer for this comment and for thoughtful examination of the data. TFAP2C has a pleiotropic role during human embryo development and is indeed expressed in the NANOG/SOX2/POU5F1-positive ESC and is important for human naïve pluripotency⁴³. However, co-expression plots of *GATA3* and *TFAP2C* show that the two genes overlap only in the GATA3-AP2 γ inducible cell population, albeit to varying levels.

In terms of CDH1/E-Cadherin staining, the outer layer of cells derived from the GATA3-AP2 γ inducible cells (marked by GFP) is E-Cadherin positive; however, this is difficult to see through the GFP in that image. Please find the presentation of a separate channel and zoom-in (below) that demonstrates E-Cadherin expression within the GFP-positive outermost population. As the reviewer notes, TFAP2C/AP2 γ is also expressed in the amnion as it differentiates from the epiblast-like compartment, as well as putative primordial germ cell-like cells.

We now note there is heterogeneity in the inducible populations in 2D in the manuscript (L186-189):

<< Together, these data demonstrate that transcription factor-mediated induction of extraembryonic cell fate from RseT hESCs drives hypoblast- or trophoblast-like gene programs without the need for exogenous factors, albeit heterogeneously and with some deficiencies in marker gene expression (**Extended Data Fig. 4c**). >>

Response to Reviewers Figure 4.8: *TFAP2C* and *GATA3* co-expression and *E-Cadherin* expression in inducible *GATA3-AP2γ* cells.

(A) *GATA3* and *TFAP2C* show some regional expression in the inducible *GATA3-AP2γ* cell sequencing but do overlap in a substantial proportion. Notably, *TFAP2C* is expressed in the RSeT ESC as expected for naïve ESC. (B) *E-Cadherin* is expressed in the outer, GFP-positive *GATA3-AP2γ* cells of the inducible human embryoids. Scale bar = 100µm.

By aggregating the previously reprogrammed cells with wild-type (control) RSeT pluripotent stem cells in the specific proportion of 1:1:2 of WT:*GATA6-SOX17*:*GATA3-APγ* the authors manage to generate 3D structures that self-organize with an efficiency of 23%. Those structures have an inner cellular domain expressing pluripotent stem cell markers, and an outer cellular domain constituted of two concentric layers of cells expressing *GATA6* or *GATA3*.

(12) How was the proportion 1:1:2 defined and selected? Did the authors test any other cell proportions and what were the results? The authors mentioned that a proportion of 1:1:1 cells was initially evaluated and deemed appropriate. How and why did they change these proportions? The authors should assess this potentially critical point more systematically.

We have now added our rationale for the selection of this ratio and included a discussion of the cell numbers in day 8 human embryos and expand on these choices:

<< At 8 d.p.f., *in vitro* cultured human embryos had 32:24:228 epiblast:hypoblast:trophoblast cells⁴. Importantly, however, many of the trophoblast cells are not in contact with the inner cell mass-derived tissues or are terminally differentiated. Additionally, we observed that in culture, inducible *GATA6-SOX17* cells proliferate slower than the other two cell populations after doxycycline addition. Therefore, we utilized an initial seeding density with: (1) a total cell number similar to that used in mouse models that allowed for successful cell sorting^{29,30}; (2) a ratio of cells that reflects the peri-implantation embryo; and (3) a reduced number of inducible *GATA3-AP2γ* cells and an increased number of inducible *GATA6-SOX17* cells. >>

We apologize for the lack of clarity regarding a 1:1:1 ratio in the text. This was not considered for the generation of a model of the human embryo. Rather, a 1:1:1 ratio was used when testing the survival of cells in 2D co-culture (Fig. 1f) and the ability of the three lineages to survive in basal media.

(13) Related to this aspect, no data is shown for the relevance of a seven (7) day doxycycline induction. The previous analysis were done at three (3) days of induction. What is the rationale for this choice? Other reports in the literature suggest that a short pulse of *SOX17* is enough

to induce an extraembryonic endoderm phenotype. What is the temporal rationale for the putative trophoblast cells? Given the fact that the authors opt for continuous (one week long) induction the relevance of an inducible system is questionable. Wouldn't a constitutive or a combination of both inducible and constitutive expression be more relevant for the model proposed by the authors?

We chose to analyze cells in 2D after three days of induction as this is the starting material taken forward to generate embryoids. Therefore, it was crucial to confirm that these cells had upregulated extraembryonic-like gene programs to the extent that they could self-organize after aggregation.

However, we chose not to adopt a constitutive overexpression approach because this severely limits our ability to probe the potential effects of pluripotent state on differentiation – as these cells could not be maintained as pluripotent cells. One significant advantage of inducible expression systems is the ability to easily maintain hESC in culture, which aids in the adoption of our model by other laboratories, which may not have the skills or reagents to culture multiple distinct cell lines. Taken together, constitutive overexpression would limit our ability to apply naïve culture conditions (which require conversion from the primed state) as a precursor to induction and embryoid formation.

Regarding the timing of overexpression, short pulses of induction are suitable for reprogramming of mESCs to extraembryonic fates and we utilize this ability when generating post-implantation embryoids using mESCs^{29,30}. However, it is established that the time required for reprogramming cell fate in the human context is markedly longer. For example, 144h⁴⁵ or 6 days of GATA6 induction in PMID:26109048. GATA3 overexpression was induced by Krendl et al. (2017)⁴¹ (reference 44; PMID:29078328) for 72-96 hours before analysis⁴¹.

However, to check whether extended induction of the transgenes may negatively impact our embryoids, we carried out new experiments in which we focused on elucidating the effect of withdrawing doxycycline 1 or 3 days after induction (see Response to Reviewers Figure 4.23 below). These data demonstrate that the removal of doxycycline at 3 days post-aggregation allows for an increase in the expression of anterior hypoblast marker CER1. Thus, the ability to modulate the timing of induction is a crucial benefit of an inducible system in comparison to constitutive overexpression systems.

(14) The authors assess (the efficiency of) their model system using PXGL and primed state cells as a starting material, but always using a combination of the two transgenes. The authors should also perform and provide data for such assessment/comparison using single transgene expression as described to be important in the latter half of the manuscript when talking about single GATA6 expressing cells.

While the use of only GATA6 inducible cells is an important part of the later part of our manuscript, we now include data showing these structures have lower initial efficiency of embryoid formation (Extended Data Fig. 9a; included below in Response to Reviewers Figure 4.23). We have tested GATA6 and GATA3 single-inducible lines to generate embryoids. While this combination did yield embryoids, it did so at significantly lower efficiency in RSeT conditions. Likewise, single induced structures failed to generate organized embryoids from PXGL or primed conditions. We feel that optimization of these conditions further is out of the scope of the current manuscript.

(15) Throughout the manuscript, indications in blue for Laminin, N-Cad and other used antibodies (blue color) are very hard to read, please change the shade of blue used.

We have adjusted the immunofluorescence labels to a lighter blue throughout the manuscript to make them easier to read.

(16) In Figure 2G, N-Cadherin staining seems to be not restricted to the membrane of cells. The authors should validate their results using an alternative antibody and/or alternative methods.

We can confirm that the N-Cadherin staining is membrane-specific. However, this is harder to see in the 3D structures presented (as in Fig. 2g). Clear examples of membrane staining can be seen Extended Data Fig. 3B (provided below for convenience).

Response to Reviewers Figure 4.9: N-cadherin expression 2D.

(A) Validation of N-Cadherin membrane localization, visualized in 2D, after induction of doxycycline for 3 days with or without exogenous factor addition (Activin-A, CHIR99021, and LIF - 'ACL').

(17) E-Cad staining is membranous but seems to be present on the basal and apical pole of cells, in contradiction with the statement of the authors that epiblast equivalents are properly epithelialized. Could the authors describe the expression pattern of other apical or basal markers such as ZO1, aPKC or Integrins?

We have added immunofluorescence for additional markers of apical polarity (ZO1 and PARD6) at day 4 (Fig. 2h; provided below), which show clear apical polarity and support our conclusion that the epiblast-like compartment is correctly polarized.

Response to Reviewers Figure 4.10: Apicobasal polarization of the epiblast-like domain of inducible human embryoids.

(A) Additional characterization of apicobasal polarity within the epiblast-like domain of embryoids. At day 4 we observe a clear PODXL+ lumen at the apical edge, juxtaposed by a basal basement membrane (marked by laminin). In addition, apical markers PARD6 and ZO-1 are localized correctly to the luminal side of the epiblast-like domain.

(18) In the described model and in vitro structures the morphology of an actual human/primate embryo can clearly not be recapitulated. This raises the question of the relevance of this model system as an in vitro embryo of human embryogenesis. The strong and largely exaggerated claims by the authors should be toned down and make clear that this is an “embryoid” model system that only partially recapitulates some aspects of the actual human embryo.

Our stem cell model mimics many but not all aspects of the embryo, as all other embryo model published so far – with none mimicking all features of the embryo. We have noted that in the manuscript (L508-514):

<< Our post-implantation embryoids are able to self-organize and in rare cases show axis formation. Similar to other human embryo models such as blastoids, further optimization is needed to permit the maintenance of all major lineages of the post-implantation embryo with their full developmental potential and embryo-like morphology. As this model cannot implant it does not have the capacity to develop toward fetal stages and does not mimic stages beyond primitive streak formation or contain all cell types of the gastrulation-stage embryo⁴⁶. >>

(19) In Figure 2G we can see an “embryoid” with 2 lumens in the central putative epiblast domain. What is the proportion of embryoids containing a single lumen vs multiple lumens at

Day4, 6 and 8? This aspect should be worked out and mentioned clearly as it is relevant to back-up the core claim of the authors that they can recapitulate aspects of post-implantation human development.

We have now quantified lumen number as the reviewer suggests (see Fig. 2h; also provided above in Response to Reviewers Figure 4.10). The vast majority of embryoids (84.21%) have a single lumen. In the image referenced, we note that the inner domain had folded in on itself, and the lumen was a single contiguous domain, which is clear when moving through z-planes.

(20) Regarding the 23% success rate claimed by the authors, what were the inclusion and exclusion criteria used to measure this success rate? How was it calculated and what are the success rates for the other conditions the authors have assessed?

We note that the 23% is in reference to structures that had successfully undergone self-organization (an inner SOX2-positive domain, an outer layer of GATA3-positive cells, and an intermediate GATA6-positive domain, and a lumen in the SOX2-positive inner domain). In terms of other conditions assessed, we have now presented quantifications of embryoid formation for variations on the culture protocol in Extended Data Fig. 9a (included below in Response to Reviewers Figure 4.23).

(21) In Figure 2G bottom panel, what is the relevance of the human embryo shown here? The staining is done for CER1 which has no comparable staining for the induced structures in this figure.

We apologize that the placement of the human embryo here was unclear. We include the embryo here to demonstrate the architecture of the embryo at the peri-implantation stages, with an epiblast rosette surrounded by hypoblast cells. We also include the CER1 channel as this becomes important to the manuscript in subsequent figures.

We now ensure that this panel is referenced later in the manuscript to demonstrate the appearance of CER1 in the embryo and to aid in comparison to our embryoids:

<< BMP signals are localized to the posterior of the embryo by the antagonistic action of the anterior hypoblast, which secretes inhibitors of BMP, WNT, and NODAL, including CER1 and LEFTY1⁴ (Fig. 2f). >>

Next, the authors performed single cell RNA/ATAC (multiome) sequencing of their induced structures at Days 4, 6 and 8 of culture and compared the obtained gene signatures with available datasets of human and cynomolgus monkey embryos. Based on these results they concluded that bona fide trophoblast cells are absent from their structures, but that they have extraembryonic mesenchyme, amnion and PGC cells.

(22) The RNA-seq data obtained by the authors should be compared with other relevant datasets available for both human embryo models and in vitro cellular components, e.g. trophoblast cells generated in vitro (Dong et al., 2020; Liu et al., 2020, Viukov et al., 2022), extraembryonic mesenchyme (Pham TXA et al., 2022), human peri-implantation embryos (Yuan et al., 2022; <https://doi.org/10.1101/2022.06.02.494565>), primate embryos (Zhai et al., 2022) or the latest human blastoid structures (Kagawa et al., 2022)

We now provide comparisons to extraembryonic mesenchyme (EXMC) and trophoblast stem cells (TSCs) from Pham et al., 2022³, blastoids in Kagawa et al., 2022¹, and to the post-implantation amniotic sac embryoids from Zheng et al., 2019², which show agreement between relevant cell types from these models and our embryoids (Extended Data Fig. 6b-c; provided below). These new analyses are in addition to our previous comparisons to several non-human primate and human embryo datasets spanning the peri-implantation period.

Response to Reviewer Figure 4.11: Comparisons of inducible human embryoid clusters to *in vitro* model datasets.

(A) Logistic regression analysis comparing *in vitro* human model and directed differentiation datasets (training data) to inducible human embryoids (test data). (B) scmap projection of human inducible embryoid dataset onto *in vitro* datasets. *In vitro* datasets from Pham et al., 2022³, Kagawa et al., 2022¹, and Zheng et al., 2019². In all cases, published data was used as training data and data produced in this manuscript was used as test data. The gene set used was differentially expressed genes identified in this manuscript (presented in Extended Data Table 2).

(23) The authors identify no trophoblast cells in their model system. Please clearly state what these cells are then and elaborate on their role as it seems that they are needed to generate embryoids/ embryo-like structures.

We now include a direct analysis of GFP-positive GATA3-AP2γ inducible cells within embryoids from our RNA-sequencing dataset (Extended Data Fig 6d and below in Response to Reviewer Fig 4.9 below). Our analysis shows that cells derived from GATA3-AP2γ inducible cells co-express endodermal genes (*GATA6*, *GATA4*, *SOX17*) alongside trophoblast genes (*GATA3*, *TEAD4*, *KRT19*, *GATA2*). This mixed gene signature is likely due to their position on the outside of the embryoids after aggregation, which has been shown to drive endodermal cell fate in embryoid bodies^{14,15}. Therefore, these cells do not have a developmentally relevant correlate in the embryo we know of.

However, despite the loss of a clear trophoblast identity (at least by day 4 after aggregation), the reviewer correctly points to the necessity of this cell type for successful embryoid formation. We tested this in additional series of experiments and show that excluding inducible GATA3-AP2γ cells results in a failure of embryoid formation (Extended Data Fig. 8e). Given that CellPhoneDB analysis indicated that inducible GATA3-AP2γ cells may be the initial source of BMP at aggregation (Extended Data Fig. 8d and Response to Reviewer Fig 4.11), we hypothesize that these cells initially support aggregation and survival partially through acting as a BMP source. In line with this, when LDN198189 is added to embryoids from day

0-2 post-aggregation to block BMP signaling, we are similarly unable to recover organized structures.

Response to Reviewers Figure 4.12: GATA3-AP2 γ inducible cells are required for embryoid formation.

(A) CellPhoneDB analysis showing potential receptor-ligand interactions between inducible cell populations and wildtype ESCs, which may influence embryoid survival and/or organization. CellPhoneDB predicts the BMP4 source to RSeT hESCs to be the GATA3-AP2 γ inducible population. Colors for clusters on the x-axis match the receptor and/or ligand in the y-axis. (B) Violin plots showing gene expression of BMP ligands and receptors in the cells used to generate inducible human embryoids, confirming expression of BMP ligands in the GATA3-AP2 γ inducible population. (C) Inducible human embryoids do not form if induced GATA3-AP2 γ cells are excluded or if LDN193189 is added between days 0-2. (D) Quantification of embryoid formation efficiency from C. (E) Quantification of embryoid size after LDN193189 addition between days 0-2. Scalebars = 100 μ m. Statistics: (E) Unpaired t-test. **p<0.01, ****p<0.0001.

As noted above, the following is included in the Discussion section (L482-490):

<< For example, while GATA3 and AP2 γ induction drive trophoblast-like gene programs in 2D, upon aggregation in our human embryo model, this cell population aberrantly upregulates endodermal markers (including SOX17 and GATA6). This is potentially due to their position at the outermost layer of the structure^{14,15}. Nevertheless, the GATA3-AP2 γ inducible cells are required for successful embryoid organization and likely act as a crucial source of BMP at aggregation. Optimization to preserve the expression of trophoblast-like gene programs while preventing the aberrant upregulation of endodermal identity in these cells will be important to improve future iterations of this model. >>

Response to Reviewer Figure 4.13: Inducible GATA3-AP2 γ cells upregulate endodermal markers upon aggregation. (A) GATA3-AP2 γ inducible population (iG3-AP2 γ) marked by GFP localizes to the outer layer of day 4 embryoids. (B) GATA3-AP2 γ inducible population marked by GFP persists at the outer layer of embryoids at day 6 and day 8 post-aggregation. (C) GFP-positive cells in the embryoid RNA-sequencing data show enrichment of endodermal markers *GATA6*, *GATA4*, and *SOX17* in addition to *GATA3*, *TEAD4*, *GATA2*, and other trophoblast-associated genes. Scale bars = 100 μ m.

(24) In Figure 3G, from what original cell population do the putative PGC cells (SOX17/NANOG/AP2 γ) cells originate from? Are they emerging from the population that expresses SOX17 or AP2 γ or the wild type pluripotent stem cells?

We have now addressed this question by generating structures with wildtype Shef6 cells labeled with nuclear mKate2 to enable tracking of these cells within our embryoids. Using this, we show that SOX17/NANOG/AP2 γ triple positive cells overlap with mKate2, demonstrating that the wildtype ESC population generates primordial germ cell-like cells in embryoids (Extended Data Fig. 7i; provided below).

Response to Reviewers Figure 4.14: Wildtype ESCs generate primordial germ cell-like cells in embryoids. (A) Expression of AP2 γ , SOX17, and NANOG triple-positive primordial germ cell-like cells overlaps with mKate2-positive wildtype cells, demonstrating that these cells are derived from the initial RSeT ESC (wildtype) population. Scale bar = 100 μ m.

(25) In Ext. Data Fig. 5C, why is the SOX17 population so small?

This is related to the cutoffs given when plotting the UMAPs. This has now been corrected and included below. We will note that *SOX17* specifically marks the hypoblast in the primate embryo at this stage^{26,27}, and is not expected to be expressed in the extraembryonic mesenchyme-like cluster.

Response to Reviewers Figure 4.15: *SOX17* expression in human embryoids.

(A) Uniform manifold approximation and projection (UMAP) colored by cell type assignment (left) and *SOX17* expression (right) showing expression in the hypoblast, as well as within putative primordial germ cells (scattered expression within AM-1 cluster).

(26) In Ext. Data Fig. 6, the *HAND1* staining seems to overlap with the *AP2g* one, but in Ext. Data Fig. 5C and Ext Data. Fig. 6B the RNA signatures do not overlap. The authors should explain this discrepancy. What are these cells *TFAP2C* negative *HAND1* positive cells?

HAND1 expression is seen in both the amnion-like and extraembryonic mesenchyme-like differentiated populations (see Extended Data Fig. 7a). Co-expression with *TFAP2C* can be seen at both RNA and immunostaining levels (please note that this cell population is present within the central domain) at later timepoints in line with ongoing maturation of the amnion. *HAND1*-positive/*TFAP2C*-negative cells are likely extraembryonic mesenchyme (EXMC).

Response to Reviewers Figure 4.16: Expression of *TFAP2C* and *HAND1* in embryoids.

(A) Uniform manifold approximation projection (UMAP) colored by cell type assignment (left), *TFAP2C* expression (middle), and *HAND1* expression (right). *TFAP2C*-negative, *HAND1*-positive cells are likely extraembryonic mesenchyme (EXMC).

(27) In Ext. Data Fig. 6, *FOXF1* expression seems to be polarized. Has this been previously reported? What is the putative biological meaning of this expression pattern? What do the *FOXF1* high or low populations correspond to?

Unfortunately, the *FOXF1* antibody shows poor penetration within larger structures and the staining pattern is likely attributable to poor antibody quality. Given this, we have chosen to

remove the immunofluorescence data using this antibody from the manuscript.

(28) The statement regarding the segregation of hypoblast cells (GATA6+/FOXF1 or TBX20 low) compared to extraembryonic mesenchyme cells (GATA6+ FOXF1 or TBX20 high) is very hard to grasp as FOXF1 high cells seems to be localized to one pole of the structure whereas TBX20 is very diffuse. If possible, staining for FOXF1 and TBX20 should be done in parallel on the same structure. Moreover, other techniques such as in situ hybridization e.g. HCR could be used to rule out any non-specificity or cross-reactivity of used antibodies.

As noted above, we have removed immunofluorescence data for FOXF1 from the manuscript due to the poor penetration of the antibody. Nonetheless, we have now co-stained for FOXF1 and TBX20, presented below.

Response to Reviewers Figure 4.17: Expression of GATA6, TBX20, and FOXF1 in day 6 RUES2 hESC-derived embryoid.

(A) Co-expression of GATA6, TBX20, and FOXF1 in a representative embryoid at day 6. Unfortunately, the FOXF1 antibody appears to penetrate the embryoids poorly.

(29) Please add references that support the authors statement that FOXF1 and TBX20 levels of expression are used to segregate those two cell populations.

We have now provided references for these markers and others mentioned in the text (e.g. HAND1^{27,33}, VTCN1^{27,33}, ISL1²⁷, FOXF1^{3,27}, TBX20³⁴).

We have also included violin plots below from a non-human primate dataset²⁷ for canonical markers including HAND1, FOXF1, TBX20, VTCN1, ISL1, SOX2 below, where the enrichment of *FOXF1* and *TBX20* in the extraembryonic mesenchyme and absence in the endoderm is apparent.

Response to Reviewers Figure 4.18: Expression of canonical markers in Yang et al., 2021 cynomolgus macaque dataset²⁷.

(A) Violin plots showing clear expression patterns of HAND1, VTCN1, ISL1, FOXF1, and TBX20 in cynomolgus macaque embryos at post-implantation stages.

(30) Regarding the two embryoids shown for TBX20 staining in Ext. Data Fig. 6E, one seems to be TBX20 high whereas the other TBX20 low, but there seems to be no obvious variability in TBX20 staining inside same embryoid. Could the authors please elaborate and explain?

The difference in TBX20 levels is likely related to differences in size between day 6 and day 8 structures (approximately 2x). However, in both cases the central epiblast/amnion-like domain is negative, giving confidence that the antibody is able to detect TBX20 specifically. We have selected different representative images to make this clearer.

(31) The model system of the authors does not show any regionalization of amnion cells vs. pluripotent epiblast cells. It seems that the entire central epithelial population acquires either one or the other fate. The claim of the authors that proper amnion cell development can be recapitulated in their model system needs to be toned down. This is also one of the major discrepancies of this model with actual human/primate embryos. This should be clarified and reflected in the manuscript in a critical and appropriate manner.

We agree with the reviewer, as noted above the following statement is now included in the manuscript (L508-514):

<< Our post-implantation embryoids are able to self-organize and in rare cases show axis formation. Similar to other human embryo models such as blastoids, further optimization is needed to permit the maintenance of all major lineages of the post-implantation embryo with their full developmental potential and embryo-like morphology. As this model cannot implant

it does not have the capacity to develop toward fetal stages and does not mimic stages beyond primitive streak formation or contain all cell types of the gastrulation-stage embryo⁴⁶. >>

We do observe instances of dorsoventral organization and maintenance of both epiblast and amnion-like domains (please see below), which does support the capacity of cells to self-organize similarly to the embryo. However, further optimization is required to achieve this organization in a robust manner. We now presented this data in our manuscript but also state that only in a minority of our embryoids we can catch this situation that both epiblast and amnion maintain their specification on day 6 post cell seeding.

Response to Reviewers Figure 4.19: Example of day 6 embryoid with dorsoventrally organized amnion (ISL1-positive) and epiblast (SOX2-positive).

(A) In rare instances, embryoids can be observed that demonstrate clear axis formation. For example, here the central domain is organized dorsoventrally with an overlying, ISL1-positive amnion, a SOX2-positive epiblast-like domain, and several Brachyury/TBXT-positive cells at one end of the epiblast-like domain.

Next, the authors analyze the effect of BMP signaling on amnion specification by analyzing target genes ID1-4 and SMAD5 vs SMAD2-4 gene expression and the phosphorylation status of SMAD 1/5 vs 2/3 in the induced structures.

(32) PGCs cell development is known to be dependent on BMP4 signaling. Can the authors elaborate on whether PGCs are still present inside the structures when BMP is inhibited and how they might be affected by modulation of BMP signaling.

We thank the reviewer for their suggestion and have now carried out experiments where we modulate BMP and NODAL signaling and quantify PGC-like cell differentiation in embryoids. Inhibition of BMP blocks the formation of PGC-like cells in embryoids, in line with a crucial role of BMP signaling in PGC-like cell formation. This data is now included in Fig 4g-h and below.

Response to Reviewers Figure 4.20: Primordial germ cell-like cells differentiate in response to BMP and NODAL signaling in embryoids.

(A) Inhibition of BMP, using LDN193189, reduces primordial germ cell-like specification in embryoids. (B) Quantification of the number of SOX17/NANOG/AP2 γ triple-positive primordial germ cell-like cells (PGCLC) at day 4 (n=136 embryoids from 6 independent experiments). Scale bars = 100 μ m. Statistics: (B) One-way ANOVA with Holm-Sidak's multiple comparisons test. * p<0.05, ** p<0.01, ***p<0.001 and ****p<0.0001.

(33) In Figure 4F and G, structures treated with BMPs or BMP agonists should be added as a control.

We thank the reviewer for their suggestion and have now added BMP4-treated embryoids to the panels indicated by the reviewer and include this data below. Interestingly, the addition of BMP4 does not alter the composition of the epiblast-like domain significantly from control conditions. This contrasts with BMP inhibition and is likely due to high endogenous levels of BMP4 within embryoids.

Response to Reviewers Figure 4.21: Inhibition of BMP or activation of NODAL delays amnion differentiation.

(A) Inhibition of BMP or activation of NODAL signaling blocks exit from pluripotency and upregulation of amnion markers AP2 α and CDX2. (B) Quantification of the percentage of inner domains expressing SOX2 and CDX2 at day 4. Scale = 100 μ m. Statistics: (B) RM two-way ANOVA with Holm-Sidak's multiple comparisons test. * $p < 0.05$, ** $p < 0.01$, *** $p < 0.001$ and **** $p < 0.0001$.

(34) What is the source of BMPs in the structures? Is it the GATA3-AP γ cell population? Could the authors stain for BMP4s in their in vitro derived structures with or without CER1 population?

Our CellPhoneDB analysis presented in Extended Data Fig. 8c-d, suggests that the GATA3-AP2 γ inducible cells are the initial source of BMP (including BMP4) when cells are aggregated at day 0 (see Response to Reviewers Figure 4.12 above). At day 4, the EXMC expresses high levels of BMP4, though other cell types are also secreting BMPs, including BMP2 and 6 from the HYPO/VE and BMP5 and 7 from the amnion-like cells.

We have stained embryoids at day 4 and day 6 for BMP4 using a well-characterized antibody (<https://www.abcam.com/products/primary-antibodies/bmp4-antibody-ab39973.html>).

However, while BMP4 appears enriched in the GATA6-positive compartment and low in the epiblast-like central domain, we are not confident that the result adds meaningfully to the RNA expression data, given that this is a highly diffusible secreted protein. We include the images below for the reviewer below.

Response to Reviewers Figure 4.22: Immunostaining of BMP4 (Abcam ab39973) in day 4 and 6 embryoids. (A) Immunofluorescence staining for BMP4 in embryoids at days 4 and 6. BMP4 appears to be more highly expressed in the interstitial, GATA6+ domain.

(35) In the structures which express CER1, could the authors confirm using IF that these cells express Noggin?

Unfortunately, we were unable to identify an antibody that reliably detected Noggin. The only antibody we could identify that was validated for immunofluorescence or immunohistochemistry (https://www.rndsystems.com/products/mouse-noggin-antibody_af719) is specifically raised against the mouse Noggin protein.

Finally the authors use their system to analyze the effect of sustained SOX17 expression on the formation of the anterior hypoblast and its effect on embryo morphogenesis.

(36) GATA6 induction alone can generate structures with CER1 and increased TBXT expression at day 6. It seems that these cells could be used instead of and replace the GATA6-SOX17 cells used throughout the manuscript to generate embryoids. The authors should elaborate this point in detail and especially in relation to our previous comment on the relevance of long-term induction of GATA6 and SOX17 alone or in combination, especially as the literature indicates that short-term induction might be sufficient to obtain similar results.

While embryoids can be generated using GATA6 induction alone they form at relatively low efficiency, hindering the use of this system to study development. For this reason, we have utilized the GATA6-SOX17 inducible cell line to generate embryoids.

Please see the response to comment (37) below for further details on the impact of long-term induction.

(37) What would be the effect on CER1 and TBXT expression when doxycycline is removed earlier during the aggregation process for example on Day 1 or Day 2?

We agree with the reviewer that this is an important additional experiment. We have now tested shorter doxycycline inductions (until day 1 or day 3 post-aggregation) and assessed

the effect on CER1 and TBXT expression. This data is included in Fig. 5 and Extended Data Fig. 9, as well as below for convenience.

At day 4 post-aggregation, CER1 expression is rescued in structures generated using the GATA6-SOX17 inducible cells when doxycycline is removed at day 3, but not day 1 post-aggregation. This suggests two major properties of inducible hypoblast differentiation in our embryoid model:

1. Together with data showing that structures generated with GATA6 induction alone also rescue CER1 expression, it provides further support that prolonged SOX17 overexpression inhibits differentiation of CER1-positive hypoblast.
2. Given that this phenotype is only observable when doxycycline is removed at day 3, it suggests that the hypoblast-like cells are not competent to differentiate into anterior hypoblast-like population at day 1 post-aggregation.

Taken together, these experiments demonstrate that an extended induction time (approximately 6 days) is necessary to specify CER1-competent anterior hypoblast-like cells but that further induction inhibits the formation of this cell type. Importantly, removal of doxycycline at day 3 allows for high efficiency embryoid formation (utilizing the GATA6-SOX17 inducible cells), as well as efficient CER1 formation and downstream differentiation.

Response to Reviewers Figure 4.23: Prolonged SOX17 overexpression is antagonistic to anterior hypoblast-like differentiation.

(A) Efficiency of tri-lineage organized structure recovery from embryoids derived using distinct induction regimes for hypoblast-like cell induction or by removing doxycycline at day 1 or day 3 of embryoid formation. N=2 independent experiments for each. (B) Representative examples and quantification of inducible human embryoids and quantification showing CER1-positive cells surrounding the epiblast-like domain generated using induced GATA6 (iG6) but not induced GATA6-SOX17 (iG6-S17) dual induction of hypoblast-like cells. CER1 expression is also observed if doxycycline is withdrawn at day 3 post-aggregation but not at day 1. n=247 structures from 7 independent experiments. (C) Representative examples of inducible human embryoids and quantification at day 6 post-aggregation showing an increase in BRACHYURY/TBXT expression in conditions where CER1 was observable at day 4. n=101 structures from 6 independent experiments. Scalebars = 100µm. Statistics (B-C): Two-way ANOVA with Holm-Sidak's multiple comparisons test. * p<0.05, ** p<0.01, ***p<0.001 and ****p<0.0001.

(38) What are the TBXT cells in the structures? What is their origin? Are they derived from wild type PSCs or could they also emerge from the other induced cell populations including putative hypoblast cells? Do these TBXT cells population organize and form a primitive streak as suggested by the authors? The “standard model” of the authors appears to fail to induce TBXT positive cells. SOX17 is known to be involved in the differentiation of both mesoderm

and endoderm. Addition of SOX17 could bias the differentiation of cells into endoderm while lack of it might lead to the formation of TBXT mesodermal cells in the hypoblast population present in the model of the authors. The authors should elaborate and comment on this possibility.

We thank the reviewer for this thoughtful comment and agree that distinguishing the origin of streak-like cells in the embryoids is an important experiment. We have now generated structures with GFP-labelled GATA3-AP2γ inducible cells, unlabeled GATA6-SOX17 or GATA6 inducible cells, and mKate2 labeled wildtype cells, to allow us to track the initial population of TBXT-positive cells within the embryoids. These experiments demonstrate that the TBXT-positive cells are derived from the wildtype population (mKate2 positive; Extended Data Fig. 9c and below for convenience).

Response to Reviewers Figure 4.24: TBXT-positive population differentiates from wildtype ESCs.

(A) Representative day 6 embryoids generated with Shef6-mKate2 ESCs demonstrate that both the ISL1-positive and BRY-positive cell populations differentiate from the wildtype cells (i.e. are mKate2-positive) in structures generated with either GATA6 (iG6) or iG6-S17 hypoblast-like cells with removal of doxycycline at day 3 post-aggregation. Scale bars = 100μm.

(39) What are the patterns of expression of pSMADs in the GATA6 positive structures where CER1 is expressed? What is the effect of SOX17 only expression and can these cells generate equivalent structures?

We have now stained structures for CER1 and pSMAD1.5 (Fig. 5e-f). This analysis demonstrated that embryoids with CER1-positive cells exhibited lower pSMAD1.5 levels within the inner domain compared to embryoids lacking this population.

Response to Reviewers Figure 4.25: Presence of the CER1-positive cell population protects the epiblast from BMP.

(A) Representative images of post-implantation human embryoids without CER1-positive cells (CER1-) or with CER1-positive cells (CER1+). (B) Quantification of pSMAD1.5 expression levels in the SOX2-positive epiblast-like domain of day 4 embryoids with or without a CER1-positive population. Scale bars = 100 μ m. Statistics: (B) unpaired t-test. **** $p < 0.0001$.

We have also generated structures with SOX17-only overexpression. Embryoids can be successfully recovered, albeit at low efficiency. Embryoids generated using the SOX17 inducible line show low expression of CER1, in line with an inhibitory role for SOX17. This data is included in Fig. 5c-g and Extended Data Fig. 9a (included in Response to Reviewers Figure 4.23 above).

References

- 1 Kagawa, H. *et al.* Human blastoids model blastocyst development and implantation. *Nature* **601**, 600-605 (2022). <https://doi.org/10.1038/s41586-021-04267-8>
- 2 Zheng, Y. *et al.* Controlled modelling of human epiblast and amnion development using stem cells. *Nature* **573**, 421-425 (2019). <https://doi.org/10.1038/s41586-019-1535-2>
- 3 Pham, T. X. A. *et al.* Modeling human extraembryonic mesoderm cells using naive pluripotent stem cells. *Cell Stem Cell* **29**, 1346-1365.e1310 (2022). <https://doi.org/https://doi.org/10.1016/j.stem.2022.08.001>
- 4 Molè, M. A. *et al.* A single cell characterisation of human embryogenesis identifies pluripotency transitions and putative anterior hypoblast centre. *Nature Communications* **12** (2021). <https://doi.org/10.1038/s41467-021-23758-w>
- 5 Shahbazi, M. N. *et al.* Self-organization of the human embryo in the absence of maternal tissues. *Nature Cell Biology* **18**, 700-708 (2016). <https://doi.org/10.1038/ncb3347>
- 6 Deglincerti, A. *et al.* Self-organization of the in vitro attached human embryo. *Nature* **533**, 251-254 (2016). <https://doi.org/10.1038/nature17948>
- 7 Xiang, L. *et al.* A developmental landscape of 3D-cultured human pre-gastrulation embryos. *Nature* **577**, 537-542 (2020). <https://doi.org/10.1038/s41586-019-1875-y>
- 8 Chhabra, S. & Warmflash, A. BMP-treated human embryonic stem cells transcriptionally resemble amnion cells in the monkey embryo. *Biology Open* **10** (2021). <https://doi.org/10.1242/bio.058617>
- 9 Lacoste, A., Berenshteyn, F. & Brivanlou, A. H. An efficient and reversible transposable system for gene delivery and lineage-specific differentiation in human embryonic stem cells. *Cell Stem Cell* **5**, 332-342 (2009). <https://doi.org/10.1016/j.stem.2009.07.011>
- 10 Liu, X. *et al.* Reprogramming roadmap reveals route to human induced trophoblast stem cells. *Nature* **586**, 101-107 (2020). <https://doi.org/10.1038/s41586-020-2734-6>
- 11 Liu, X. *et al.* Comprehensive characterization of distinct states of human naive pluripotency generated by reprogramming. *Nature Methods* **14**, 1055-1062 (2017). <https://doi.org/10.1038/nmeth.4436>
- 12 Alves-Lopes, J. P. *et al.* Specification of human germ cell fate with enhanced progression capability supported by hindgut organoids. *Cell Rep* **42**, 111907 (2023). <https://doi.org/10.1016/j.celrep.2022.111907>
- 13 Mackinlay, K. M. L. *et al.* An in vitro stem cell model of human epiblast and yolk sac interaction. *eLife* **10** (2021). <https://doi.org/10.7554/eLife.63930>
- 14 Abe, K. *et al.* Endoderm-specific gene expression in embryonic stem cells differentiated to embryoid bodies. *Exp Cell Res* **229**, 27-34 (1996). <https://doi.org/10.1006/excr.1996.0340>
- 15 Zeevaert, K., Elsafi Mabrouk, M. H., Wagner, W. & Goetzke, R. Cell Mechanics in Embryoid Bodies. *Cells* **9** (2020). <https://doi.org/10.3390/cells9102270>
- 16 Rostovskaya, M., Stirparo, G. G. & Smith, A. Capacitation of human naïve pluripotent stem cells for multi-lineage differentiation. *Development (Cambridge)* **146** (2019). <https://doi.org/10.1242/dev.172916>
- 17 Bray, N. L., Pimentel, H., Melsted, P. & Pachter, L. Near-optimal probabilistic RNA-seq quantification. *Nature Biotechnology* **34**, 525-527 (2016). <https://doi.org/10.1038/nbt.3519>
- 18 Melsted, P. *et al.* Modular, efficient and constant-memory single-cell RNA-seq preprocessing. *Nature Biotechnology* **39**, 813-818 (2021). <https://doi.org/10.1038/s41587-021-00870-2>
- 19 Townes, F. W. & Irizarry, R. A. Quantile normalization of single-cell RNA-seq read counts without unique molecular identifiers. *Genome Biology* **21** (2020). <https://doi.org/10.1186/s13059-020-02078-0>
- 20 Yan, L. *et al.* Single-cell RNA-Seq profiling of human preimplantation embryos and embryonic stem cells. *Nature Structural and Molecular Biology* **20**, 1131-1139 (2013). <https://doi.org/10.1038/nsmb.2660>
- 21 Blakeley, P. *et al.* Defining the three cell lineages of the human blastocyst by single-cell RNA-seq. *Development (Cambridge)* **142**, 3613-3613 (2015). <https://doi.org/10.1242/dev.131235>
- 22 Petropoulos, S. *et al.* Single-Cell RNA-Seq Reveals Lineage and X Chromosome Dynamics in Human Preimplantation Embryos. *Cell* **165**, 1012-1026 (2016). <https://doi.org/10.1016/j.cell.2016.03.023>
- 23 Zhou, F. *et al.* Reconstituting the transcriptome and DNA methylome landscapes of human implantation. *Nature* **572**, 660-664 (2019). <https://doi.org/10.1038/s41586-019-1500-0>
- 24 Young, M. D. *et al.* Single-cell transcriptomes from human kidneys reveal the cellular identity of renal tumors. *Science* **361**, 594-599 (2018). <https://doi.org/10.1126/science.aat1699>
- 25 Ma, H. *et al.* In vitro culture of cynomolgus monkey embryos beyond early gastrulation. *Science* **366** (2019). <https://doi.org/10.1126/science.aax7890>
- 26 Nakamura, T. *et al.* A developmental coordinate of pluripotency among mice, monkeys and humans. *Nature* **537**, 57-62 (2016). <https://doi.org/10.1038/nature19096>
- 27 Yang, R. *et al.* Amnion signals are essential for mesoderm formation in primates. *Nature Communications* **12** (2021). <https://doi.org/10.1038/s41467-021-25186-2>
- 28 Tyser, R. C. V. *et al.* Single-cell transcriptomic characterization of a gastrulating human embryo. *Nature* **600**, 285-289 (2021). <https://doi.org/10.1038/s41586-021-04158-y>
- 29 Amadei, G. *et al.* Inducible Stem-Cell-Derived Embryos Capture Mouse Morphogenetic Events In Vitro. *Developmental Cell* **56**, 366-382.e369 (2021). <https://doi.org/10.1016/j.devcel.2020.12.004>
- 30 Lau, K. Y. C. *et al.* Mouse embryo model derived exclusively from embryonic stem cells undergoes neurulation and heart development. *Cell Stem Cell* **29**, 1445-1458 e1448 (2022). <https://doi.org/10.1016/j.stem.2022.08.013>
- 31 Jo, K. *et al.* Efficient differentiation of human primordial germ cells through geometric control reveals a key role for Nodal signaling. *Elife* **11** (2022). <https://doi.org/10.7554/eLife.72811>

- 32 Hollnagel, A., Oehlmann, V., Heymer, J., Rüther, U. & Nordheim, A. Id genes are direct targets of bone morphogenetic protein induction in embryonic stem cells. *Journal of Biological Chemistry* **274**, 19838-19845 (1999). <https://doi.org/10.1074/jbc.274.28.19838>
- 33 Munger, C. *et al.* Microgel culture and spatial identity mapping elucidate the signalling requirements for primate epiblast and amnion formation. *Development* **149** (2022). <https://doi.org/10.1242/dev.200263>
- 34 Saykali, B. *et al.* Distinct mesoderm migration phenotypes in extra-embryonic and embryonic regions of the early mouse embryo. *Elife* **8** (2019). <https://doi.org/10.7554/eLife.42434>
- 35 Yanagida, A. *et al.* Naive stem cell blastocyst model captures human embryo lineage segregation. *Cell Stem Cell* **28**, 1016-1022.e1014 (2021). <https://doi.org/10.1016/j.stem.2021.04.031>
- 36 Yu, L. *et al.* Blastocyst-like structures generated from human pluripotent stem cells. *Nature* **591**, 620-626 (2021). <https://doi.org/10.1038/s41586-021-03356-y>
- 37 Dong, C. *et al.* Derivation of trophoblast stem cells from naïve human pluripotent stem cells. *eLife* **9** (2020). <https://doi.org/10.7554/eLife.52504>
- 38 Guo, G. *et al.* Human naive epiblast cells possess unrestricted lineage potential. *Cell Stem Cell* **28**, 1040-1056.e1046 (2021). <https://doi.org/10.1016/j.stem.2021.02.025>
- 39 Chen, D. *et al.* Human Primordial Germ Cells Are Specified from Lineage-Primed Progenitors. *Cell Rep* **29**, 4568-4582 e4565 (2019). <https://doi.org/10.1016/j.celrep.2019.11.083>
- 40 Zheng, Y. *et al.* Single-cell analysis of embryoids reveals lineage diversification roadmaps of early human development. *Cell Stem Cell* **29**, 1402-1419 e1408 (2022). <https://doi.org/10.1016/j.stem.2022.08.009>
- 41 Krendl, C. *et al.* GATA2/3-TFAP2A/C transcription factor network couples human pluripotent stem cell differentiation to trophectoderm with repression of pluripotency. *Proceedings of the National Academy of Sciences of the United States of America* **114**, E9579-E9588 (2017). <https://doi.org/10.1073/pnas.1708341114>
- 42 Gerri, C. *et al.* Initiation of a conserved trophectoderm program in human, cow and mouse embryos. *Nature* **587**, 443-447 (2020). <https://doi.org/10.1038/s41586-020-2759-x>
- 43 Pastor, W. A. *et al.* TFAP2C regulates transcription in human naive pluripotency by opening enhancers. *Nature Cell Biology* **20**, 553-564 (2018). <https://doi.org/10.1038/s41556-018-0089-0>
- 44 Séguin, C. A., Draper, J. S., Nagy, A. & Rossant, J. Establishment of Endoderm Progenitors by SOX Transcription Factor Expression in Human Embryonic Stem Cells. *Cell Stem Cell* **3**, 182-195 (2008). <https://doi.org/10.1016/j.stem.2008.06.018>
- 45 Wamaitha, S. E. *et al.* Gata6 potently initiates reprogramming of pluripotent and differentiated cells to extraembryonic endoderm stem cells. *Genes & Development* **29**, 1239-1255 (2015). <https://doi.org/10.1101/gad.257071.114>
- 46 Clark, A. T. *et al.* in *Stem Cell Reports* Vol. 16 1416-1424 (Cell Press, 2021).

Reviewer Reports on the First Revision:

Referees' comments:

Referee #1 (Remarks to the Author):

In their revision the authors clarify a number of important points raised in the first round of review through additional experimentation, analysis, and revision of the text. The strengths as well as the limitations and complexities of their model are now expressed much more clearly. Though the model has its limitations, the study represents a timely contribution to a rapidly evolving and exciting field, and one that will spur further advances towards more refined embryo model systems.

Referee #3 (Remarks to the Author):

The extensive revisions and additions to the manuscript have in large-part clarified the comments raised in my initial review. In particular, the details of the computational analyses have been clarified, the essentiality of G3-Ap cells has been demonstrated.

I do have one suggestion / clarification with regards to the response to inferring dynamics of BMP signaling along trajectories. The approach used by authors of multivelo latent time and SwitchDE is reasonable but I am not sure if it utilized correctly. Trends should be computed separately for each lineage to demonstrate that the highlighted dynamics are correct. The trends are currently fit across both lineages which makes it near impossible to interpret the results.

Referee #4 (Remarks to the Author):

The authors have addressed the majority of the points and concerns which I had raised previously in a largely satisfactory manner. They provide overall appropriate responses and supporting data in a point-by-point manner. I furthermore believe that the new title of the manuscript describes more accurately the results shown in this study. Of notice, I find that the functional validation of the generated inducible cell lines using interspecies chimeras is an elegant and powerful approach, which taken together with the added new data provides sufficient support to the claims made by the authors. I furthermore agree with the statement of the authors that “investigating other combinations of transgenes may be a fruitful avenue to better generate bona fide extraembryonic cells in vitro but believe this is outside the scope of this model and might lead, in the future, to another human embryo model.”

Minor remaining points:

1. The figure legend of Extended Data Figure 1h should include detailed information about the experimental conditions, such as the timing and concentration of Dox treatment, as illustrated in Response to Reviewers Figure 4.1 in the rebuttal.
2. In Figure 3c the legends are overlapping (for example percent active is on top of “40”)

Author Rebuttals to First Revision:

We thank the reviewers for their consideration of our work, and for their input which allowed us to substantially improve the manuscript. We have responded to the points below in blue.

Referee #1 (Remarks to the Author):

In their revision the authors clarify a number of important points raised in the first round of review through additional experimentation, analysis, and revision of the text. The strengths as well as the limitations and complexities of their model are now expressed much more clearly. Though the model has its limitations, the study represents a timely contribution to a rapidly evolving and exciting field, and one that will spur further advances towards more refined embryo model systems.

We thank the reviewer for their consideration of our work.

Referee #3 (Remarks to the Author):

The extensive revisions and additions to the manuscript have in large-part clarified the comments raised in my initial review. In particular, the details of the computational analyses have been clarified, the essentiality of G3-Ap cells has been demonstrated.

We are grateful to the reviewer for their considered comments on our work.

I do have one suggestion / clarification with regards to the response to inferring dynamics of BMP signaling along trajectories. The approach used by authors of multivelo latent time and SwitchDE is reasonable but I am not sure if it utilized correctly. Trends should be computed separately for each lineage to demonstrate that the highlighted dynamics are correct. The trends are currently fit across both lineages which makes it near impossible to interpret the results.

We thank the reviewer for the suggestion to split the latent time and SwitchDE plots by trajectory. We have now updated the plots in Extended Data Fig. 8a-b to reflect this. We have shown through both the multivelo velocity inference and experimentation with mKate2 fluorescently labeled wildtype cells that our epiblast-like domain gives rise to both amnion and extraembryonic mesenchyme cell types in embryoids. Therefore, we have plotted either epiblast to amnion or epiblast to extraembryonic mesenchyme trajectories. To achieve this, we subset our sequencing object by the relevant cell types, and ran SwitchDE to fit trends over latent time for each trajectory, individually. We agree with the reviewer that the similarities and differences between these trajectories are clearer with this approach.

Referee #4 (Remarks to the Author):

The authors have addressed the majority of the points and concerns which I had raised previously in a largely satisfactory manner. They provide overall appropriate responses and

supporting data in a point-by-point manner. I furthermore believe that the new title of the manuscript describes more accurately the results shown in this study. Of notice, I find that the functional validation of the generated inducible cell lines using interspecies chimeras is an elegant and powerful approach, which taken together with the added new data provides sufficient support to the claims made by the authors. I furthermore agree with the statement of the authors that “investigating other combinations of transgenes may be a fruitful avenue to better generate bona fide extraembryonic cells in vitro but believe this is outside the scope of this model and might lead, in the future, to another human embryo model.”

We thank the reviewer for their consideration and comments.

Minor remaining points:

1. The figure legend of Extended Data Figure 1h should include detailed information about the experimental conditions, such as the timing and concentration of Dox treatment, as illustrated in Response to Reviewers Figure 4.1 in the rebuttal.

We have updated the figure legend of Extended Data Fig. 1h to reflect the additional information provided in the previous rebuttal.

2. In Figure 3c the legends are overlapping (for example percent active is on top of “40”)

We have adjusted the figure accordingly, and thank the reviewer for pointing out this oversight.